# A synergistic mindsets intervention protects adolescents from stress

David S. Yeager[1✉], Christopher J. Bryan[2✉], James J. Gross[3], Jared S. Murray[4,5], Danielle Krettek Cobb[6], Pedro H. F. Santos[4], Hannah Gravelding[7], Meghann Johnson[1] & Jeremy P. Jamieson[7✉]

Social-evaluative stressors—experiences in which people feel they could be judged negatively—pose a major threat to adolescent mental health[1–3] and can cause young people to disengage from stressful pursuits, resulting in missed opportunities to acquire valuable skills. Here we show that replicable benefits for the stress responses of adolescents can be achieved with a short (around 30-min), scalable 'synergistic mindsets' intervention. This intervention, which is a self-administered online training module, synergistically targets both growth mindsets[4] (the idea that intelligence can be developed) and stress-can-be-enhancing mindsets[5] (the idea that one's physiological stress response can fuel optimal performance). In six double-blind, randomized, controlled experiments that were conducted with secondary and post-secondary students in the United States, the synergistic mindsets intervention improved stress-related cognitions (study 1, $n = 2,717$; study 2, $n = 755$), cardiovascular reactivity (study 3, $n = 160$; study 4, $n = 200$), daily cortisol levels (study 5, $n = 118$ students, $n = 1,213$ observations), psychological well-being (studies 4 and 5), academic success (study 5) and anxiety symptoms during the 2020 COVID-19 lockdowns (study 6, $n = 341$). Heterogeneity analyses (studies 3, 5 and 6) and a four-cell experiment (study 4) showed that the benefits of the intervention depended on addressing both mindsets—growth and stress—synergistically. Confidence in these conclusions comes from a conservative, Bayesian machine-learning statistical method for detecting heterogeneous effects[6]. Thus, our research has identified a treatment for adolescent stress that could, in principle, be scaled nationally at low cost.

Adolescents today are suffering record levels of stress-related anxiety and depressive symptoms[1–3]. This has prompted public health experts to call for urgent action to mitigate the forthcoming 'mental health pandemic'[7] by understanding and addressing adolescent stress[8,9].

Conventional thinking portrays stress as mostly a bad thing to be avoided or kept at bay[10]. But this 'stress avoidance' mentality ignores the reality that elevated levels of stress are a normal and, in many ways, even a desirable feature of adolescence[11]. Adolescents must acquire a wide and varied array of complicated social and intellectual skills as they transition to adult social roles and prepare for economic independence. This developmental process is inherently stressful, but it is also essential to the task of becoming an adult[11]. The conventional view that high levels of stress are toxic is likely to lead many adolescents simply to disengage from stressors such as demanding coursework, putting them at a serious disadvantage in the future. Technology has displaced many low-skilled jobs and created more well-compensated but highly technical ones[12]. As a result, adolescents must complete more advanced coursework in mathematics and science than ever before to be competitive for many of the most attractive careers[13].

The demands of this advanced technical coursework are experienced by many adolescents as highly stressful[14]. Moreover, in recent years, the COVID-19 pandemic has created intense and unrelenting stress in the form of social isolation, uncertainty about the future and, for many families, financial distress[1–3]. To protect adolescents from negative mental health effects and help them to prepare for a competitive and technically demanding labour market, we must find a way to help young people to embrace and overcome the challenges that characterize this life stage.

In consequence, affective scientists have increasingly advocated for a stress optimization approach, defined as learning to engage positively with rigorous but useful social and academic stressors, rather than seeking indiscriminately to minimize or avoid stress[5]. To date, however, the search for an intervention that effectively equips adolescents with stress optimization skills has been largely unsuccessful. Although therapies can sometimes provide relief to those already suffering from stress-related clinical symptoms, interventions aimed at the broader non-clinical population have been found to produce short-lived, mostly negligible protection, at best, from

[1]Department of Psychology and Behavioral Science and Policy Institute, University of Texas at Austin, Austin, TX, USA. [2]Department of Business, Government, and Society and Behavioral Science and Policy Institute, University of Texas at Austin, Austin, TX, USA. [3]Department of Psychology, Stanford University, Stanford, CA, USA. [4]Department of Information, Risk and Operations Management, University of Texas at Austin, Austin, TX, USA. [5]Department of Statistics and Data Sciences, University of Texas at Austin, Austin, TX, USA. [6]Empathy Lab, Google, Mountain View, CA, USA. [7]Department of Psychology, University of Rochester, Rochester, NY, USA. ✉e-mail: dyeager@utexas.edu; Christopher.Bryan@mccombs.utexas.edu; jeremy.jamieson@rochester.edu

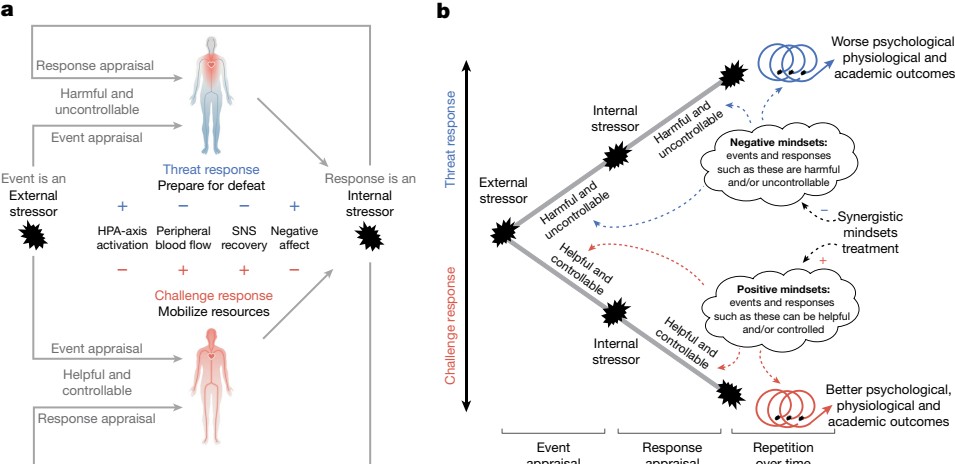

**Fig. 1 | How young people's social-evaluative stressors accumulate consequences for healthy development. a,b,** First, the individuals appraise both acute stressful events and their stress responses (**a**); and second, their mindset beliefs shape their appraisals and responses, which leads to differences in internalizing symptoms over time (**b**). This integrated model is rooted in established process models in affective science[16,26], recursive process models in psychology[44,47] and mindset models[4,23,72]. **a,** Stressful events, such as a challenging exam or an argument with a friend, are appraised as more harmful and uncontrollable or more helpful and controllable, cultivating threat or challenge response tendencies, respectively. Then, the meaning of the stress response is appraised as either distressing and non-functional (harmful and uncontrollable) or as a resource that helps one address situational demands (helpful and controllable), which results in further threat- or challenge-type stress responses, respectively[25,26]. Threat stimulates the hypothalamic–pituitary–adrenal (HPA) axis in the brain, the end-product of

which is the catabolic adrenal hormone cortisol, in anticipation of damage or social defeat[31]. Challenge is characterized by increased peripheral blood flow (hence the red depiction), and a faster return to homeostasis after stress offset. Threat, however, results in increased vascular resistance and less oxygenated blood flow to the periphery (hence the blue depiction) as HPA activation tempers sympathomedullary effects and produces a more prolonged stress response[25,26,29]. Threat leads to avoidance motivation and negative affect, whereas challenge elicits approach motivation and more positive affect relative to threat. SNS, sympathetic nervous system. **b,** Mindsets are situation-general beliefs about categories of events (for example, academic stressors) and responses (for example, feelings of worry) that shape appraisals at the event stage and next at the response stage[5,21,23,29]. Individuals who respond with an optimized challenge-type stress response engage with and respond to future stressors more adaptively in a self-reinforcing, positive feedback cycle that results in better coping and performance.

the mental health risks that are associated with non-optimal stress management[15].

In past laboratory experiments, teaching people to reappraise a specific stressful experience (that is, to reinterpret its meaning[16]), such that they see it as helpful and controllable (versus unhelpful and uncontrollable), has been shown to improve immediate cognitive, physiological and behavioural stress responses[17]. This reappraisal approach, however, suffers from the 'transfer problem': people typically fail to extrapolate from the specific instance of reappraising a single stressful experience to the general lesson that they can reappraise other stressful experiences in a similar manner[18,19]. In the present research, we build on the reappraisal approach by targeting mindsets—cognitive processes that operate at a more general level than situation-specific appraisals and can shape how people interpret the meaning of broad categories of situations (for example, struggling to master a skill or negative emotions in general)[20–22]. Mindsets, therefore, can guide people's appraisals of a wide range of situations within the relevant category, including completely novel situations like the need to keep up with academic work through remote learning during pandemic-related school closures.

Here we show that it is possible to achieve stress optimization by targeting adolescents' mindsets about their stressful experiences. We demonstrate that a short (around 30-min) intervention that could, in principle, be administered at low cost to entire populations of adolescents[4] successfully optimized adolescents' stress responses. We document these improvements using an array of complementary indicators at multiple levels of analysis, including adolescents' cognitive appraisals of a stressful demand on them, their cardiovascular and neuroendocrine responses to such stressors, and the emergence of downstream mental health symptoms from exposure to chronic daily stress (Fig. 1a,b).

## The synergistic mindsets approach

We designed the intervention that we evaluate here to harness the complementarity that we identified between two existing mindset interventions, each of which targets a different aspect of people's experience of stress. The first of these, the growth mindset[4,20,23], centres on the belief that ability (for example, intellectual, athletic or musical) is not fixed but can be developed with effort, effective strategies and support from others. This mindset casts normal but challenging stressors (for example, rigorous, advanced coursework) as both helpful (because they provide opportunities for valuable learning and skill development) and controllable (because the abilities needed to overcome them can be developed). The second, known as the stress-can-be-enhancing mindset[5,21], centres on the understanding that our psychophysiological stress response (for example, sweaty palms, racing heart, deeper breathing and feeling anxious) can be positive (because these changes mobilize energy and deliver oxygenated blood to the brain and tissues) and can be controlled once you understand its purpose (because you can choose to take advantage of the enhanced capacity for performance it fuels rather than being worried and distracted by it).

These two mindsets were not presented as separate ideas, but rather as intertwined and complementary elements of a coherent whole. The growth mindset messaging was designed to shape adolescents' appraisals of the stressful demands on them—encouraging them to think of difficult challenges not as hazards to be avoided but as valuable opportunities for self-improvement. The stress-can-be-enhancing mindset messaging encouraged adolescents to see the activation of their psychophysiological stress response, which often follows engagement with challenging stressors, as a helpful resource that energizes their pursuit of valued goals, rather than as a problem.

We argue that these two mindsets need to be integrated to reliably optimize stress management in real-world settings (Fig. 1a,b). For

example, if an adolescent believes that struggle can promote learning (an event-focused growth mindset), but also believes that their psychophysiological stress response is harmful and uncontrollable (a response-focused stress-is-debilitating mindset[5]) the activation of that stress response might deter them from pursuing stressful but valuable learning experiences. Likewise, an adolescent who understands that their psychophysiological stress response can be used as a resource (a response-focused stress-can-be-enhancing mindset) but sees difficulty and struggle as hazards to be avoided (an event-focused fixed mindset) is still at risk of disengaging from stressful demands any time that they encounter difficulty or failure. By targeting both mindsets simultaneously, the synergistic mindsets intervention can convey the empowering message that both stressful events and stress responses can be harnessed in support of valued goals.

## Overview of six experiments

We assessed the effects of the synergistic mindsets intervention in six experiments. Approvals for these studies were obtained from the Institutional Review Boards at the University of Rochester or the University of Texas at Austin. Participants in all studies provided informed consent or assent. The studies all focused on the kinds of stressors that are common in educational contexts (for example, taking a timed quiz, giving a speech to classmates, transitioning to high school or keeping up with academic work during the social isolation of pandemic-related school closures) and that constitute a primary source of adolescents' evaluative stress as they navigate a sometimes-volatile social world while also acquiring the technical and intellectual skills that they need for adulthood[24]. Adolescents completed the online intervention module on their own, in a naturalistic school setting, without assistance and without discussing the content with each other or with instructors. Hence, the study procedures mirrored the routine conditions under which scale-up could occur.

Our aim, in every study, was to reduce threat-type stress responses. Threat-type stress responses begin with the appraisal that a stressor is harmful (that is, 'bad for me') and uncontrollable, which leads to the conclusion that one cannot handle the demands of the stressor (that is, a threat appraisal)[25]. Threat appraisals lead to a cascade of physiological and psychological responses that follow from the expectation that one is about to experience potentially catastrophic damage and defeat[25,26] (Fig. 1a,b). The order of the six experiments corresponds to the typical sequence that threat-type stress responses follow, from cognitive appraisals to physiological (cardiovascular and neuroendocrine) responses to internalizing symptoms[27] (Fig. 1 and Table 1).

We used a Bayesian statistical analysis approach that uses machine-learning tools to model covariates (and their complex interactions), and to model heterogeneous effects. It uses Bayesian additive regression tree (BART) priors to make these models conservative. This mitigates the problem of arbitrary covariate or moderator specifications leading to spurious or overstated results. We focus on effect sizes and uncertainty intervals rather than on 'all-or-none' null hypothesis significance testing. All findings also met conventional frequentist standards for statistical significance (Extended Data Table 2 and Supplementary Fig. 5).

## Effects on cognitive appraisals

In two large, pre-registered experiments, we examined the effects of the intervention on the cognitive appraisal processes that comprise the first step in the threat-type stress response. Participants in study 1 were 2,717 secondary school students in 35 public schools in the United States who, after completing the synergistic mindsets (or a control) intervention, were asked to imagine that the instructor of their most difficult course had just assigned a very demanding project with very little time to complete it and that they would be expected to present

**Table 1 | Overview of studies**

| Studies (Sample size) | Population | Stressor | Measures of threat-type stress response |
|---|---|---|---|
| 1 (*n*=2,717) | 13–18-year-old public school students in the USA during the COVID-19 pandemic | Anticipated timed assignment | Event- and response-focused appraisals |
| 2 (*n*=755) | Diverse undergraduate students attending a public university | Experienced timed assignment | Cognitive appraisals at 1–3 days and 3 weeks after test |
| 3 and 4 (3, *n*=160; 4, *n*=200) | Undergraduate students at a private university | Trier Social Stress Test (TSST) | Peripheral blood flow |
| 5 (*n*=118 individuals; *n*=1,213 observations) | 14–16-year-old adolescents from racial or ethnic minority groups, facing economic disadvantages | Daily stressors in high school | Daily negative self-regard and activation of the HPA axis |
| 6 (*n*=341) | Same as study 2 but during the onset of the COVID-19 pandemic in spring 2020 | Ongoing academic demands during COVID-19 quarantines | Generalized internalizing symptoms |

All experiments were conducted in the United States. Across the six experiments, the synergistic mindsets intervention reduced maladaptive beliefs compared to the control condition by 0.25 s.d. or more, which means that each experiment passed the manipulation check (see Methods, 'Manipulation checks (all studies)' for more detail).

their work in front of their classmates. As expected, the intervention reduced negative event-focused appraisals of this hypothetical academic stressor relative to controls (for example, "How likely would you be to think that the very hard assignment in [your most stressful class] is a negative threat to you?"); average treatment effect (ATE) = −0.11 s.d. [−0.03, −0.20] (numbers in square brackets are the 10th and 90th percentiles). The intervention also reduced negative response-focused appraisals (for example, "I think my body's stress responses would hurt my performance"); ATE = −0.19 s.d. [−0.08, −0.30]. These outcomes correspond to the first two steps depicted in Fig 1b.

Study 2 examined the effects of the intervention on appraisals of a real, acute stressor (Fig. 2). Participants were 755 students in a large, undergraduate introductory social science course at a selective public university in the United States. Immediately after a timed, challenging quiz (which occurred one to three days after intervention and was not mentioned in the intervention content), treated participants made less-negative stress appraisals; ATE = −0.39 s.d. [−0.28, −0.51]. This effect persisted but was attenuated by around 50% when participants completed a subsequent timed quiz three weeks after the first; ATE = −0.18 s.d. [−0.05, −0.31]. Note that even the attenuated effect size at the three-week follow-up was indistinguishable in size from the effect on immediate appraisals of a hypothetical stressor in study 1. Study 2 showed that participants transfer the lessons of a one-time, short, self-guided intervention, with no boosters, to the naturalistic stressors that they encounter in their daily lives, and that this protection endured for at least three weeks after treatment.

## Effects on physiological responses

Study 3 used a well-validated, standardized acute stress induction paradigm (the Trier Social Stress Test[28 (TSST)], see also ref. [29]) to assess whether the stress-optimizing effects of the intervention extend to people's cardiovascular stress responses. Participants were 166 university students who completed the study for course credit. Consistent with standard TSST protocols, participants were informed

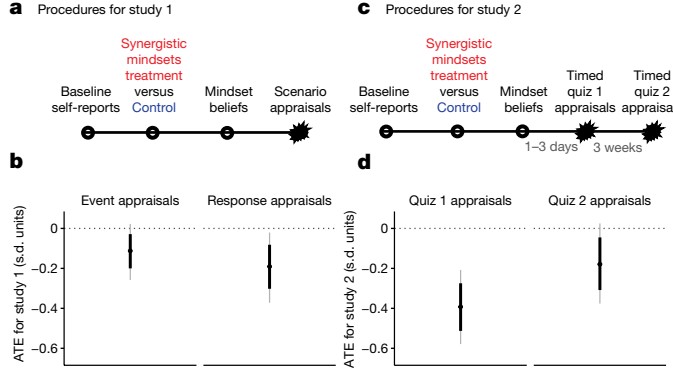

**Fig. 2 | Procedures and results of studies 1 and 2. a–d**, Studies 1 and 2 (*n* = 2,534 and *n* = 790, respectively) showed that relative to the neutral control condition the synergistic mindsets intervention reduced negative appraisals of an immediate, hypothetical stressor (**a**,**b**), and an acute naturalistic stressor up to 3 weeks after the intervention (**c**,**d**). Participants were secondary school students (study 1) or undergraduates (study 2) attending public schools in the United States. Starbursts represent stressor onset. Dots correspond to the ATEs estimated with the Bayesian model. Thick lines represent the 10th to 90th percentiles; grey lines represent the 2.5th to 97.5th percentiles. The appraisals for each study were coded so that higher values corresponded to more negative appraisals, so negative treatment effects are consistent with a beneficial stress optimization effect. Average effect sizes appear in the text. Study 1, control *n* = 1,326; treatment *n* = 1,208. Study 2, control *n* = 403; treatment *n* = 387.

that they would be asked to deliver an impromptu speech about their personal strengths and weaknesses in front of an audience of peer evaluators. Evaluators were trained to provide negative nonverbal feedback (for example, furrowing brow, sighing, crossing arms and so on) and no positive feedback—either verbal or nonverbal—during the speech[28]. When the speech was complete—and without prior warning—participants were asked to do mental mathematics (counting backwards from 996 in increments of 7) as quickly as possible in front of the same unsupportive evaluators. Evaluators immediately called attention to any errors participants made in the mental mathematics task and instructed them to begin again. Figure 3a depicts the five TSST epochs during which electrocardiography (ECG), impedance cardiography (ICG) and blood pressure signals were monitored to assess stress responses, with the speech epoch expected to elicit the most distress. The focal outcome was total peripheral resistance (TPR), a measure of vasoconstriction in the body's periphery (that is, the limbs) and a primary indicator of threat-type stress responses[26,30] (Fig. 1a). Therefore, we expected the intervention to reduce the levels of TPR.

### Average effects

Control group participants exhibited an increase in TPR from the baseline to the active epochs (Fig. 3b). Consistent with existing literature[31], increases in TPR were most pronounced during the epoch in which participants delivered the impromptu speech. Analyses, therefore, focus primarily on the effects of the intervention during the speech epoch.

The synergistic mindsets intervention reduced participants' TPR, relative to controls, in every epoch of the TSST, and especially during the speech epoch—the most intense period of social-evaluative stress (Fig. 3b). The estimated conditional average treatment effect (CATE) was less than zero in every epoch (Fig. 3c). Analyses of other cardiovascular indicators of threat- versus challenge-type stress responses (stroke volume during active epochs, and pre-ejection period (PEP) during the post-stressor recovery epoch) revealed treatment effects consistent with those on TPR (Extended Data Figs. 2 and 3).

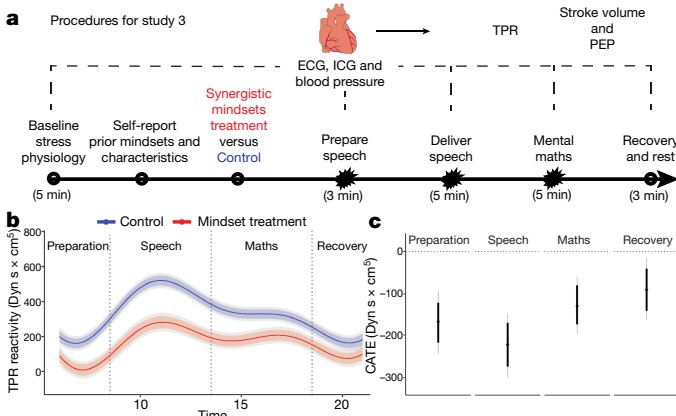

**Fig. 3 | In study 3, the synergistic mindsets intervention improved cardiovascular responses to the TSST. a–c**, Participants in study 3 (*n* = 160) were undergraduate students in a laboratory experiment. **a**, Procedures for study 3. **b**,**c**, Coloured lines (**b**) and dots (**c**) correspond to the expected value of the outcome (**b**) or the ATE (**c**), estimated with the Bayesian model. The thick bands represent the 10th to 90th percentiles of the posterior distributions; grey lines represent the 2.5th to 97.5th percentiles. TPR (**b**) is measured in Dyn s × cm⁵, where. Time indicates the elapsed, cumulative physiological recording. Starbursts indicate TSST epochs that presented acute demands (that is, the stressful epochs). Baseline measurements were taken before the stress induction and random assignment to condition. Baseline scores were subtracted from all active epochs to compute reactivity scores for each minute. Preparation measurements were taken after intervention materials when participants planned their speech; speech delivery and mental mathematics measurements were taken during the speech and maths tasks, respectively; and finally, measurements were taken during a recovery period in which evaluative pressure (stress) was removed. The differences in TPR for the two groups were similar at baseline (see propensity score comparisons in the Supplementary Information). In **c**, ATEs and 10th to 90th percentiles are: preparation = −168 Dyn s × cm⁵ [−217, −121], speech = −223 [−274, −172], maths = −128 [−175, −80], recovery = −90 [−139, −41]. Control, *n* = 86; treatment, *n* = 74.

### Heterogeneous effects

We assessed participants' event- and response-focused mindsets by self-reporting before randomization, and tested for moderation by these variables. We expected negative prior mindsets to predict worse stress responses in the control condition, and this was confirmed (Extended Data Table 1). We also hypothesized that the synergistic mindsets intervention would provide the greatest benefit to participants who did not already endorse both positive mindsets (that is, growth and stress-can-be-enhancing), and who were therefore at greater risk of a threat-type response to the TSST. This is what we found (Extended Data Fig. 1). Indeed, participants with dual negative mindsets before the intervention who received the synergistic mindsets treatment exhibited levels of TPR that were indistinguishable from controls with dual positive mindsets before intervention (Fig. 3c). Analyses of other, complementary cardiovascular indicators (for example, stroke volume) yielded the same pattern (Extended Data Fig. 2).

## Replication of physiological effects

Study 4 was a pre-registered replication and extension of study 3. Participants were 200 university students who completed the study for course credit.

### Replication of effects on TPR

Directly replicating the findings in study 3, the synergistic mindset intervention again reduced TPR during the speech epoch of the TSST, relative to the control condition; ATE = −0.44 s.d. [−0.67, −0.20]; posterior probability of a reduction in TPR = 0.994.

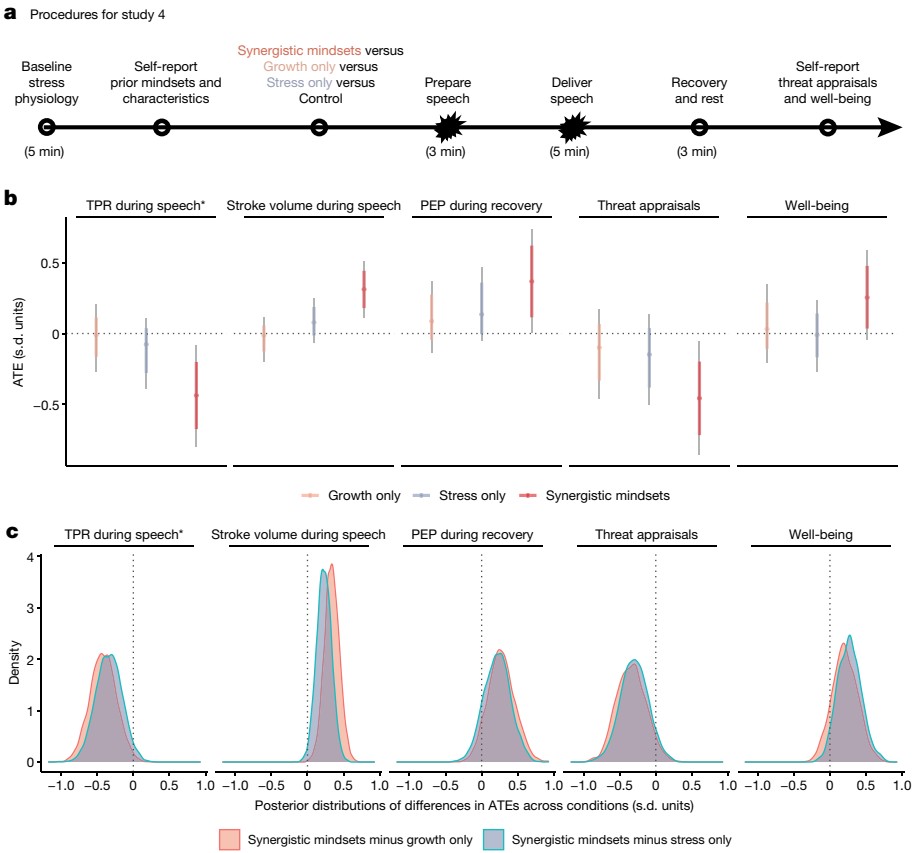

**Fig. 4 | In study 4, the synergistic mindsets intervention improved cardiovascular responses to the TSST, and this effect was larger than the effects of single-mindset interventions. a**–**c**, Participants in study 4 (*n* = 200) were undergraduate students in a laboratory experiment. **a**, Procedures for study 4. **b**, ATEs across outcomes. Dots correspond to the ATEs estimated with the Bayesian algorithm. Thick lines represent the 10th to 90th percentiles; grey lines represent the 2.5th to 97.5th percentiles. **c**, The entire posterior distributions of a difference between the treatment effects of the conditions (synergistic mindset versus single mindset) (that is, a test of the interaction

effect hypothesis), estimated in the Bayesian model. Study 4 streamlined the TSST procedures to allow for more efficient data collection, so the maths epoch was removed. The pre-registration stated that the primary outcome would be TPR during the speech delivery epoch. All results were estimated with the multi-arm implementation of the BCF algorithm; cardiovascular outcomes (TPR, stroke volume, PEP) used targeted smoothing. Additional details for the study procedures are provided in the legend of Fig. 3. In **a**, starbursts represent stressor onset. Asterisks in **b**,**c** indicate a pre-registered outcome. Control, *n* = 44; growth only, *n* = 52; stress only, *n* = 65; synergistic, *n* = 39.

## Comparison to single-mindset conditions

In addition to replicating the findings of study 3, study 4 included two additional conditions—a growth-mindset-only treatment and a stress-mindset-only treatment—to test whether the synergistic combination of positive event- and response-focused mindsets is truly essential to prevent threat-type responses, as our theoretical model predicts (Fig. 1), or whether one or the other of these component mindsets might be equally effective on its own. This four-cell experiment was analysed using a multi-arm implementation of the Bayesian causal forest (BCF) model, which was developed for the present research. Figure 4 shows that neither of the single-mindset treatments reliably reduced TPR relative to the neutral control condition: stress (but not growth) mindset, posterior probability of a reduction in TPR = 0.785; growth (but not stress) mindset, posterior probability = 0.578). As predicted, the ATE of the synergistic mindsets intervention was larger than the stress-mindset-only ATE by an average of −0.34 s.d. [−0.57, −0.10] (posterior probability of a negative difference = 0.971), and was −0.42 s.d. [−0.66, −0.18] larger than the growth-mindset-only ATE (posterior probability = 0.990; see Fig. 4c for a summary plot of the posterior distributions).

## Extension to secondary cardiovascular outcomes

The conclusion that the synergistic combination of the two mindsets is more powerful than either of its component mindsets alone is further

supported by an analysis of stroke volume during the speech epoch, and PEP during the recovery epoch—both of which are positive indicators of a challenge-type stress response. The synergistic mindsets ATEs for stroke volume and PEP were 0.31 s.d. [0.18, 0.44] and 0.37 s.d. [0.11, 0.62], respectively (Fig 4b). Consistent with the TPR findings, these ATEs were both meaningfully larger than the ATEs for either the stress-mindset-only or the growth-mindset-only condition (posterior probabilities of a difference in ATEs for stroke volume = 0.999 and 0.989, respectively; for PEP: 0.876 and 0.923 respectively; Fig 4c).

## Understanding mechanisms

Study 4 also included, on an exploratory basis, two self-report measures that extended the model in Fig. 1. The first was a more direct measure of threat (versus challenge) appraisals (for example, ratings of the statements "I felt threatened by the task" and "I felt that the task challenged me in a positive way"). The second was a measure of psychological well-being (for example, feeling more liked, powerful and high in self-esteem, and less rejected, insecure or disconnected). For each outcome, the synergistic mindsets condition showed the predicted effects relative to the control condition (appraisals ATE = −0.46 s.d. [−0.72, −0.20]; well-being ATE = 0.25 s.d. [0.04, 0.48]). The ATE of the synergistic mindsets intervention was also meaningfully larger than those of either single-mindset treatment for both outcomes (Fig. 4c; all posterior probabilities of a difference in the direction of the point estimate > 0.884).

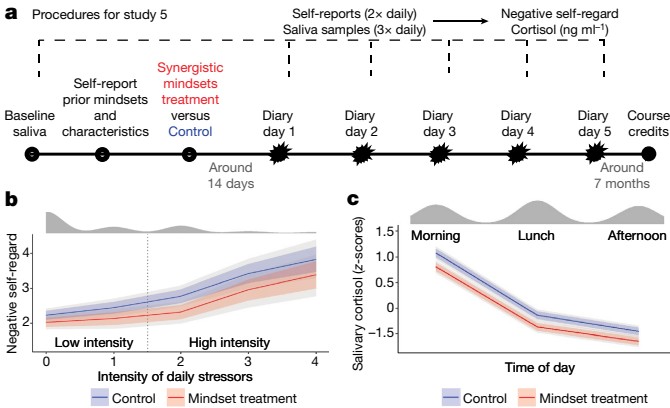

**Fig. 5 | In study 5, the synergistic mindsets intervention reduced negative self-regard. a**, Procedures for study 5. **b,c**, The synergistic mindsets intervention reduced negative self-regard (see Methods for the scoring of this measure) relative to controls overall and especially on intensely stressful days (**b**). The intervention also reduced (**c**) daily salivary cortisol levels overall relative to controls. Participants (*n* = 119 individuals; *n* ≤ 1,213 observations (total number of daily diary or cortisol observations for all participants)) were students from low-income families attending a public high school in the United States. Starbursts represent stressor measurements. Univariate marginal distribution plots are shown at the top in **b,c**. Thick coloured lines represent the 10th to 90th percentiles; grey lines represent the 2.5th to 97.5th percentiles. The vertical dashed line in **b** represents the cut-off point for high versus low daily stress intensity that was used to estimate subgroup CATEs. The unstandardized CATE for negative self-regard for high daily stress intensity was −0.48 scale points [−0.81, −0.14]; for low daily stress intensity days it was −0.23 scale points [−0.44, −0.02]. The ATEs for academic course credits and cortisol are presented in the text. Control, *n* = 58; treatment, *n* = 61.

## Effects on daily stress responses

Study 5 assessed the longer-term protective effects of the synergistic mindsets intervention using psychological and hormonal indicators of repeated unhealthy responses to stress over time. Participants were 118 adolescents who attended a rigorous, urban public charter high school in a low-income neighbourhood; 95% identified as Black/African-American or Hispanic/Latinx, and 99% were from economically disadvantaged families. We chose this population because students facing the combination of socioeconomic disadvantage and demanding academic standards are especially likely to experience increased levels of chronic, daily stress[32–34]. In addition, because this sample is quite different demographically from the samples in our other studies, study 5 helps us to gauge the generalizability of the synergistic mindsets intervention to other population subgroups that might stand to benefit from it.

The study procedures are shown in Fig. 5a. Participants first completed a pre-intervention survey assessment of negative event- and response-focused mindsets, and then completed the synergistic mindsets (or control) intervention in a private room at school, with random assignment occurring at the individual level. Then, an average of 14 days later, students completed brief (5-min) stress surveys twice daily over the course of one school week (4–5 consecutive days), yielding up to 10 daily stress reports per individual. The daily surveys measured the intensity of evaluative stress that participants were experiencing, and their global feelings of self-regard ("Overall, how good or bad did you feel about yourself today?"). Negative self-regard is a precursor of clinical anxiety and depression and a central symptom of clinical depression[35]. On the same days on which daily stress assessments were taken, students also provided up to three saliva samples (in the morning after arrival at school; during the lunch period; and after school ended) that were later assayed for cortisol levels using liquid chromatography–tandem mass spectrometry (LC–MS/MS)[36].

When individuals undergo a threat-type response to stress, cortisol levels rise immediately and remain increased after stress offset, as the hormone lingers in the body for approximately 1 h (refs. [25,31]). Persistently elevated cortisol levels across samples taken multiple times each day over multiple days, therefore, reflect chronic activation of the hypothalamic–pituitary–adrenal (HPA) axis, a clear indication of threat-type responses to daily stressors. Affective states, by contrast, were assessed in reference to specific stressors that occurred in each survey period. Thus, these two indicators—self-reported daily stress intensity paired with negative self-regard, and overall cortisol levels across all days and times—can provide complementary information about daily stress responses.

### Average effects: negative self-regard

The synergistic mindsets intervention reduced daily negative self-regard compared to controls overall by −0.19 s.d. [−0.33, −0.05]. This effect was more than twice as large on high-stress days, −0.32 s.d. [−0.54, −0.09] than on low-stress days, −0.15 s.d. [−0.37, −0.01], as one would expect of an intervention designed to optimize people's responses to stress (Fig. 5b). Daily stress intensity was positively associated with negative self-regard in the control condition, *r*(532) = 0.38, but this association was attenuated by 50% in the treatment condition, *r*(521) = 0.19 (Fig. 5b). In sum, the synergistic mindsets intervention protected against the negative mental health effects of the most intense, negative stressors.

### Heterogeneous effects: daily negative self-regard

The intervention's buffering effect against negative self-regard on high-stress days was 40% larger (−0.38 s.d.), on average, among individuals who held negative event- and response-focused mindsets before the intervention, than among participants who held positive prior mindsets (−0.27 s.d.; Extended Data Fig. 4).

### Average and heterogeneous effects: cortisol

The synergistic mindsets intervention reduced the chronic HPA-axis activation of participants, relative to controls, as assessed using the average cortisol levels of participants across all measurement days and times; ATE = −0.23 s.d. [−0.34, −0.12]. Self-reported daily stress intensity was unrelated to cortisol levels (*r*(1182) = 0.01), consistent with the interpretation of average cortisol levels across measurement days and as a global indicator of the functioning of the HPA system, not as an index of responses to specific stressors. No meaningful heterogeneity (across time, stress intensity or prior mindsets) was observed in the cortisol effects.

### Academic achievement

As we explained above, the synergistic mindsets intervention is designed not only to prevent negative mental health effects of normal stress but also to help adolescents to engage with (rather than disengaging from) useful but stressful learning opportunities such as rigorous academic coursework. Therefore, we sought to assess, on an exploratory basis, whether the synergistic mindsets intervention had a positive effect on students' academic outcomes in study 5. We obtained data on the rate at which participants passed their core classes from official school transcripts. Notably, in the six to seven months from the end of the post-intervention daily diary measurement until the end of the school year when final grades were recorded, we had no contact with participants and they received no reminders of the intervention or its content. The school year in question was the one that ended in the spring of 2020 during the COVID-19 lockdowns. Using the highly conservative BCF method, we found that the synergistic mindsets intervention increased the overall rate at which students passed their core classes by 14.4 percentage points (pp) [0.4, 29.4]. These treatment effects were driven by improvements in the most demanding and technical courses (mathematics and science), which students in

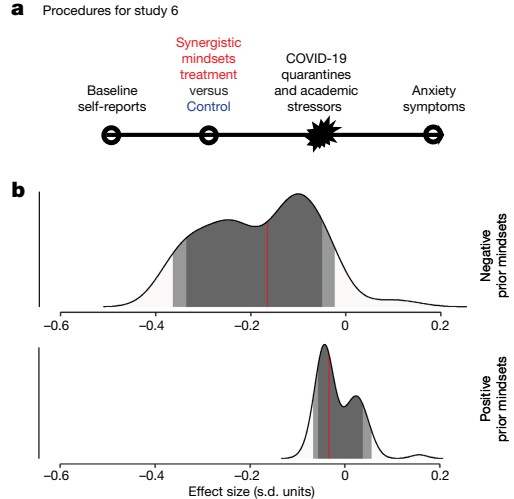

**a** Procedures for study 6

Baseline self-reports — Synergistic mindsets treatment versus Control — COVID-19 quarantines and academic stressors — Anxiety symptoms

**Fig. 6 | In study 6, the synergistic mindsets intervention reduced symptoms of general anxiety relative to controls.** Participants in study 6 (*n* = 351) were undergraduate students attending a public university in the United States. **a**, Procedures for study 6. **b**, The posterior distribution of treatment effects estimated by the Bayesian model, the red line is the CATE for each subgroup, the dark shading marks the interquartile range and the lighter shading marks the 10th to 90th percentiles. Although there was a small posterior probability of a null treatment effect among prior negative mindset participants, there was a higher probability of effects > 0.30 s.d. The prior mindset subgroups used to display treatment effects in **b** were generated by implementing a hands-off Bayesian decision-making algorithm that maximized the outcome differences among the mindset groups, without using information about magnitudes of treatment effects (see Supplementary Information). Control, *n* = 172; treatment, *n* = 179.

the control condition passed at a rate of only 47%. By contrast, 63% of participants in the synergistic mindsets condition passed these courses; ATE = 14.5 pp [0.4, 31.7]. Smaller and less reliable effects were observed in non-STEM courses (English, language arts and social studies), which had a much higher overall pass rate and tend to be less stressful on average (control = 67%; mindset treatment = 73%; ATE = 5.3 pp [−4.8, 17.2]). Treatment effects on course pass rates were not moderated by prior negative mindsets. This exploratory analysis provides direct evidence that, in addition to providing robust and enduring protection of adolescents' mental (and physical) health during periods of high stress, the intervention also helps adolescents to take fuller advantage of stressful but valuable opportunities for learning and skill development. Second, this analysis helps to allay any concerns that the findings of the study in the cortisol and daily diary data were inflated because the act of completing the daily diaries artificially boosted the salience of the intervention's key ideas in participants.

## Effects on overall anxiety symptoms

The results in studies 4 and 5 suggest the possibility for cumulative consequences of mindsets for mental health during times of negative stress[37] (Fig. 1). This possibility was explored with a final experiment. In study 6, the environmental stressor was continued academic pressure and social isolation during the early stages of the COVID-19 pandemic in the United States in the spring of 2020, as students were forced to leave university housing and abstain from most normal, in-person social interaction (see study procedure in Fig. 6). Thus we thought that reshaping adolescents' appraisals of the normal social-evaluative demands of student life, which did not abate during the pandemic, might have had substantial protective effects on the mental health of participants during this period. The outcome of interest was participants' levels of generalized anxiety symptoms, measured with the same standardized,

widely used screening tool[38] used in past representative sample surveys that have contributed to public concern about a mental health crisis in the wake of the COVID-19 pandemic[3].

Participants were 341 students in a section, offered during the spring semester of 2020, of the same large, undergraduate introductory social science course from which we sampled in study 2, but in the next semester. Participants completed either the synergistic mindsets or the control interventions—framed as a course activity—at the end of January 2020, and participants completed the survey of generalized anxiety symptoms as part of a course activity on psychological disorders in mid-April—approximately one month after the university suspended all in-person teaching in response to the COVID-19 pandemic. Participants were not made aware of any link between the intervention and the anxiety survey—both of which they saw as regular components of the course—thus providing a strong test of the transfer hypothesis.

Because studies 3 and 5 found stronger salutary effects of the synergistic mindsets intervention among those with negative event- and response-focused mindsets pre-intervention—and because those mindsets were positively associated with anxiety symptoms in the control condition (Extended Data Table 1)—we expected the Bayesian algorithm to again find stronger effects for this group in Study 6.

Among participants who had negative prior mindsets, those who received the synergistic mindsets (versus the control) intervention in January exhibited lower levels of generalized anxiety symptoms in April; CATE = −0.17 s.d. [−0.37, 0.00] (see Fig. 6b). Although the BCF model identified a small probability of a near-null effect in this subgroup (Fig. 6b)—unsurprising because BCF uses a highly conservative prior distribution—that probability was considerably smaller than the probability that the treatment effect exceeded 0.30 s.d., which would be a large effect for real-world symptom reductions[39]. There was no discernible effect among adolescents with positive pre-intervention mindsets who, as noted, were less likely to show anxiety symptoms overall; CATE = −0.03 *SD* [−0.17, 0.12] (see also Extended Data Fig. 5).

## Discussion

Across six randomized experiments using a range of outcome measures, levels of analysis and timescales, we found replicable evidence that a single-session, self-administered, synergistic mindsets intervention can protect vulnerable adolescents against unhealthy threat-type responses to normal social-evaluative stress and the negative mental health outcomes associated with such stress responses. Although our focus has primarily been on the protective effects of this intervention against the negative mental health effects of treat-type responses, it is worth noting that the profile of cardiovascular responses that are characteristic of threat-type stress responses (increased TPR and reduced stroke volume during active stress response, and a slower return to baseline PEP after stress offset)—and which the synergistic mindsets intervention protected vulnerable participants against—is known to increase the risk of cardiovascular disease and premature death. Future studies should assess more directly whether this intervention might provide significant protection against the negative physical health effects of chronically elevated stress.

Because mindset interventions similar to the one tested here can be delivered in a cost-effective manner in national scale-up studies[4], the present research represents a critical theoretical step from basic insights about affect regulation towards the discovery of actionable intervention methods that might be able to produce real, lasting change at scale. Although our evidence indicates that many of the intervention's benefits were specific to participants with negative pre-intervention event- and response-focused mindsets, it makes the most sense to think of the synergistic mindsets intervention as a tool for universal prevention rather than targeted 'high risk' prevention. We found no evidence that the intervention caused harm to any group, and we did find some evidence that it can have key benefits (for

example, reduced global cortisol levels, improved academic achievement) to participants irrespective of their prior mindsets. For these reasons and because it would be prohibitively difficult and costly to accurately identify all those at increased risk of negative stress-related outcomes, interventions like this one, which aim to protect people against population-level risk factors, typically produce much larger improvements in public health when they are administered to entire populations[40–42].

An important next step, however, will be to more fully assess the generalizability and heterogeneity of these effects with new large-scale trials in diverse populations and contexts[43]. These trials might reveal previously undiscovered context-, population- or individual-level moderators of the intervention's effects that inform decisions about how best to scale the intervention; for example, by identifying environmental conditions known as 'affordances' on which the beneficial effects of the intervention depend[44,45]. Doing so can also contribute to theory by shedding light on the psychological mechanisms by which the intervention has its effects[43,46]. The finding, in the present research, that many of the intervention's effects were moderated by participants' prior mindsets, for example, suggests that it works by interrupting the negative recursive process[47] of appraisals stemming from negative mindsets that, if left unchecked, can have accumulating negative psychological consequences (Fig. 1b).

We emphasize that our claims about the benefits of synergistic mindsets are limited to how adolescents respond to the inevitable stress that comes from engaging with challenging opportunities for learning and skill development, such as formal education. The intervention is not designed to change people's appraisals of serious, negative and uncontrollable stressors, such as trauma or abuse. With that said, we did find evidence that the synergistic mindsets intervention can help people cope better with the normal stress of preparing for adulthood in the modern economy, even when they are also facing harmful and uncontrollable stressors, such as economic disadvantage (study 5) or pandemic-related lockdowns (study 6). We are furthermore optimistic that synergistic mindsets could have protective effects in the face of a wide range of normal stressors (for example, in workplace, athletic or romantic contexts). To work effectively in such contexts, however, the details of the intervention content would probably need to be adapted to convey the relevance of synergistic mindsets to the stressors that people face in those settings.

Finally, our research suggests that the public discourse is at present operating under a flawed narrative about young people and what they are capable of. As we noted in the opening of this article, the predominant societal reaction to alarming levels of anxiety and stress has been to argue that we should expect less of young people. But, in a time characterized by political division and social unrest, climate change, rising inequality and geopolitical conflict, it is critical that young people gain the knowledge and skills that they will need to solve humanity's challenges when they take over society's important institutions. Adolescence, after all, is a developmental stage that is uniquely suited to reshaping the future. Therefore, we propose an alternative narrative that emphasizes the role of young people in taking on the formidable challenges of the future. Our studies suggest that we might not teach adolescents that they are too fragile to overcome difficult struggles, but that we might, instead, provide them with the resources and guidance that they need to unleash their skills and creativity in addressing big problems.

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

# Methods

## Ethics approval

Approvals for these studies were obtained from the Institutional Review Boards at the University of Rochester or the University of Texas at Austin. Participants in all studies provided informed consent or assent.

## Study registration and efforts to curb researcher degrees of freedom

All studies are registered on the Open Science Framework (study 1: https://osf.io/tgysd; study 2: https://osf.io/hb6vs, study 3: https://osf.io/x4a63; study 4: https://osf.io/fkgru; study 5: https://osf.io/9pfha; study 6: https://osf.io/mkqgf). Detailed descriptions of open science disclosures, links to study materials, analysis plans and deviations from analysis plans appear in the Supplementary Information. Studies 1, 2 and 4 were registered before analysing the data. Studies 3, 5 and 6 were registered after analysing the data. As explained in greater detail in the Supplementary Information, researcher degrees of freedom for Studies 3, 5 and 6 were constrained by following published and previously pre-registered standard operating procedures for TSST and daily diary studies[29] (the focus on TPR, stroke volume and PEP in study 3 and the focus on the stressor intensity × treatment interaction in study 5), and by following the same analysis steps as the pre-registered studies (for example, the same core covariates and moderators whenever measured and the same conservative BCF modelling approach).

## Intervention overview

The intervention consisted of a single self-administered online session lasting approximately 30 min. Random assignment to the intervention or control condition occurred in real time via the web-based software Qualtrics, as participants were completing the online intervention materials. Simple random assignment was used, with equal probabilities of selection, but the actual observed proportions in treatment or control groups varied randomly across the six studies. Participants were blinded to the presence of different conditions, and teachers or others interacting with participants were blind to the intervention content and to condition assignment. Thus, the intervention experiments used a double-blind design throughout.

## Synergistic mindsets intervention

The intervention used methods for mindset interventions that are well-established in the literature and have been used successfully in national scale-up studies[4]. The intervention first aimed to convey the message that stressful events are controllable and potentially helpful. It did so by targeting negative fixed mindset beliefs, or the belief that intellectual ability is fixed and cannot change, which can lead to the appraisal that negative events are uncontrollable and harmful. In particular, the fixed mindset leads to a pattern of appraisals about effort (that having to try hard or ask for help means you lack ability), about causes of failures (the attribution that failure stems from low ability) and about the desired goal in a setting (the goal of not looking stupid in front of others)[20,48]. The intervention overcame these negative patterns of appraisals by conveying the growth mindset. The growth mindset promotes the appraisal that difficulties can be controlled and helpful. It argues that most people who became good at something important had to face and overcome struggles, and therefore, your own struggles should not be viewed as signs of deficient abilities but instead should be viewed as part of your path toward important skill development. To justify the controllable and helpful stressor appraisal, the intervention drew on neuroscientific information about the brain's potential to develop more efficient ('stronger') connections when it faces and overcomes challenges, using the analogy of muscles growing stronger when they are subjected to rigorous exercise[49].

Second, the intervention targeted the stress-is-debilitating mindset[50], which is the belief that stress is inherently negative and compromises performance, health and well-being; this mindset leads to the appraisal that a given stressor is uncontrollable and harmful. Counter to the stress-is-debilitating mindset, the intervention developed here introduced the stress-can-be-enhancing mindset[50], which is the belief that stress can have beneficial effects on performance, health and well-being; this more adaptive belief system leads to the appraisal that stressors can be potentially helpful and controlled. The intervention explained that when people undergo challenges, they inevitably begin to experience stress, which can manifest in a racing heart, sweaty palms or possibly feelings of anxiety or worry. The intervention leads people to perceive those signals as information that the body is preparing to overcome the challenge; for instance, by providing more oxygenated blood to the brain and the muscles[17]. Thus, the stress response is framed as helpful for goal pursuit, not necessarily harmful. The intervention also argued that feelings of anxiety can be a sign that you have chosen a meaningful and ambitious set of goals to work on, and therefore can indicate a positive trajectory, not a negative one.

Notably, these two mindsets were conveyed synergistically, not independently, so that they built on one another. Participants were encouraged to view struggles as potentially positive and worth engaging with, and then they were invited to view inevitable stress coming from this engagement as a part of the body's natural way to help them overcome the stressor.

These mindset messages were couched within a summary of scientific research on human performance and stress. Participants were not simply informed of these facts, but they were instead invited to engage with them, make them their own and plan how they could use them in the present and future. Participants heard stories from prior participants (older students in this case) who used these ideas to have success in important performance situations, and they also completed open-ended and expressive writing exercises. For instance, participants wrote about a time when they were worried about an upcoming stressor, and then later on they wrote advice for how someone else who might be undergoing a similar experience could use the two mindsets they learned about, which has been called a 'saying-is-believing' writing exercise[51].

We defined adherence as completion of the last page of the intervention. In the studies in which participants were closely supervised by researchers (studies 3, 4 and 5), adherence was high (97% to 99%). In the studies in which the intervention was self-administered with no supervision, adherence was lower but still acceptable: 85%, 88% and 82% for studies 1, 2 and 6, respectively. Because we conducted intent-to-treat analyses, participants were retained in the analytic sample regardless of intervention completion status.

## Control group content

The control group intervention was also an online, self-administered activity lasting around 30 min. It was designed to be relatively indistinguishable from the intervention group by using similar visual layout, fonts, colours and images. The content was predominately from the control condition from a prior national growth mindset experiment[4], which included basic information about the brain and human memory. It also involved open-ended writing activities and stories from older students. However, the control condition did not make any claims about the malleability of intelligence. To this standard content, we added basic information about the body's stress response system (for example, the sympathetic and parasympathetic nervous system and the HPA axis) to control for the possibility that simply reflecting on stress and stress responses could account for the results. The latter content did not include any evaluations of whether stress responses are good or bad, or controllable or uncontrollable.

## Negative prior mindsets

At baseline, participants in all experiments except study 2 completed standard measures of negative event-focused mindsets (fixed mindset of intelligence; that is, "Your intelligence is something about you that

you can't change very much")[4] and response-focused mindsets (the stress-is-debilitating mindset[21]; that is, "The overall effect of stress on my life is negative") (for both, 1 = strongly disagree, 6 = strongly agree). The items for each construct were combined into indices by taking their unweighted averages. Measures of internal consistency were all in the acceptable range (between 0.70 and 0.85). Means and standard deviations for each of the six studies are presented in Supplementary Table 6. In the primary Bayesian analyses for studies 3, 5, and 6, the two measures and their product were entered into the covariate and moderator function, and the machine-learning algorithm decided how best to use the mindset measures to optimize prediction or moderation. In the preliminary correlational analyses (Extended Data Table 1), we analysed the multiplicative term of the two, for simplicity.

## Analysis strategy

For all experimental analyses, we used intention-to-treat analyses, which means that data were analysed for all individuals who were randomized to condition and who provided outcome data, regardless of their fidelity to the intervention protocol. If participants were missing data on covariates, those data were imputed. This analysis is more conservative than analyses that drop participants with low fidelity, but it also better reflects real-world effect sizes.

Our research advanced a fully Bayesian regression approach called Bayesian causal forests and its extension targeted smooth Bayesian causal forests (BCF and tsBCF)[6,52,53] to calculate treatment effects and understand moderators of the treatment effects. A previous version of the BCF algorithm has won several open competitions for yielding honest and informative answers to questions about the complex, but systematic, ways in which a treatment's effects are—or are not—heterogeneous, and it is designed to be quite conservative[6]. We used the existing single-level BCF method for studies 1, 2, and 6. The model is specified in equation (1):

$$y_{ij} = \alpha_i + \beta(x_{ij}) + \tau(w_{ij})z_i + \epsilon_{ij} \tag{1}$$

In studies 3 and 4, we updated the BCF method to apply to time-series data. See equation (2):

$$y_{ij} = \alpha_j + \beta(x_j, t_{ij}) + \tau(w_{ij}, t_{ij})z_j + \epsilon_{ij} \tag{2}$$

In equations (1) and (2), $y_{ij}$ is the outcome for adolescent $i$ at time $j$, $\alpha_j$ is the random intercept for each individual, $x_j$ is the vector of covariates that predict the outcome and could control for chance imbalances in random assignment, $w_{ij}$ is the vector of potential treatment effect moderators, $t$ is time (the $t_{ij}$ term is omitted in all studies except studies 3 and 4), $z_j$ is the dichotomous treatment effect indicator for each individual, and $\epsilon_{ij}$ is the error term. (Study 4 involved additional updates to allow for multi-arm comparisons that accommodate the four-cell design; see the Supplementary Information).

What makes BCF unique, and well-suited for this application, is that both $\beta(.)$ and $\tau(.)$ are non-linear functions that take a 'sum-of-trees' representation, and which are estimated using standard BART machine-learning tools[6,54,55]. This frees researchers from making arbitrary decisions about which covariates to include, what their functional form should be and how or whether covariates should interact. Notably, BCF uses conservative prior distributions, especially for the moderator function, to shrink towards homogeneity and to simpler functions, avoiding over-fitting. The data are used once—to move from the prior to the posterior distribution—and all analyses then summarize draws from the posterior.

The BCF approach contrasts with the classical method, which involves re-fitting the model many times to estimate simple effects or to conduct robustness analyses with different specifications. The BCF approach, therefore, reduces researcher degrees of freedom, mitigating the risk of false discoveries and other spurious findings. In this research we focused on estimation of treatment effects (that is, how large the effect is) and not

null hypothesis testing (that is, whether it is 'significant' or not) because of well-known problems with the all-or-nothing thinking inherent in the null hypothesis significance test[56]. Following convention[57], we reported the ATEs and the CATEs with the associated 10th and 90th percentiles from the posterior distributions (see the Figures for the 2.5th and 97.5th percentiles). When the pre-analysis plan called for it (in study 4), we report the exact posterior probabilities of a difference in effects.

The covariates included in each study are listed in Supplementary Table 5. The core covariates and moderators were: the prior mindset measures (fixed mindset and stress-is-debilitating mindsets), sex and perceived social stress, as pre-registered (https://osf.io/tgysd). When available, other covariates were added as well: age, race or ethnicity, self-esteem, test anxiety, social class and personality. Justifications for each covariate appear in Supplementary Table 5.

## Effect size calculations

Unless otherwise noted, effects are standardized by the pooled s.d.

## Manipulation checks (all studies)

The intervention reduced negative mindset beliefs relative to controls (four items, including "Stress stops me from learning and growing" and "The effects of stress are bad and I should avoid them"; 1 = strongly disagree, 6 = strongly agree). BCF analyses revealed lower levels of negative mindsets in the synergistic mindsets intervention condition at post-test compared to the neutral control condition, signifying a successful manipulation check: study 1: ATE = −0.28 s.d. [10th percentile: −0.43, 90th percentile: −0.16]; study 2: −0.49 s.d. [−0.73, −0.24]; study 3: −0.50 s.d. [−0.89, −0.14]; study 4: −0.54 s.d. [−0.75, −0.33]; study 5: −0.26 s.d. [−0.61, 0.03]; study 6: −0.56 s.d. [−0.71, −0.40]. The two field experiments with high schoolers (studies 1 and 5) had smaller manipulation check effects that were more imprecise than the others (studies 2, 3, 4 and 6). This was expected because the former studies were conducted in naturalistic school settings that tend to produce noisier data.

## Study 1

**Sample size determination.** Sample size was planned to have sufficient power to detect a treatment effect in a field experiment of 0.10 s.d. or greater, with 0.10 s.d. being the minimum effect size that we would interpret as meaningful for a study focused on immediate post-test self-reports. We worked with our data collection partner, the Character Lab Research Network (CLRN) (https://characterlab.org/research-network/), to recruit as close to 3,000 participants as possible in a single semester. The final sample size was determined by the logistical constraints of data collection during the COVID-19 pandemic and by CLRN's data availability.

**Participants.** Participants were from a large, heterogeneous sample of adolescents who were evenly distributed across grades 8 to 12 in 35 public schools in the United States (13 years old: 16%; 14 years old: 20%; 15 years old: 20%; 16 years old: 21%; 17 years old: 18%; 18 years old: 5%). The schools were sampled from a stratum of large, diverse, suburban and urban public schools in the southeast United States. Forty-nine per cent of adolescents identified as male, 49% as female and 2% as gender non-binary. Participants were racially and ethnically diverse (participants could indicate multiple racial or ethnic identities so numbers exceed 100%): Black: 20%; Latinx: 39%; white: 68%; Asian: 7%. Participants were also socioeconomically diverse: 40% received free or reduced-price lunch, an indicator of low family income. Therefore, study 1 provided a test of the hypothesis that the intervention could be widely disseminated and effectively change beliefs and appraisals in a large and diverse sample of adolescents. Even so, the sample was not strictly representative because random sampling was not used to recruit the CLRN sample.

**Procedure.** Participants were recruited by CLRN (https://characterlab.org/research-network/), which administers roughly 45-min online survey

experiments three times per year to a large panel of adolescents in the 6th to the 12th grade. Researchers program their studies using the Qualtrics platform and students self-administer the materials at an appointed time. Data collection continued during the modified instructional settings of autumn 2020. We note that all measures had to be short so as to keep the respondent burden low and fit within the required time limit for CLRN studies. Thus, the trade-off in study 1, when achieving scale and reaching a large adolescent population during the COVID-19 pandemic, was estimating potentially weaker effect sizes owing to greater statistical noise.

**Measures.** At the beginning of the survey, participants indicated their most stressful class (for example, mathematics, science, English or language arts). Then, after the intervention (or control) experience they were asked to imagine that "later today or tomorrow your teacher [in your most stressful class] asked you to do a very hard and stressful assignment. Imagine this is the kind of assignment that will take a lot of time to finish but you only have two days to turn it in. Also pretend that you will soon have to present your work in front of the other students in your class." Participants then reported their event-focused appraisals on three items (for example, "How likely would you be to think that the very hard assignment is a negative threat to you?"; 5 = not at all likely to think this, 1 = extremely likely to think this). Next, participants reported their response-focused appraisals ("Do you think your body's stress responses (your heart, your sweat, your brain) would help you do well on the assignment, hurt your performance on the assignment, or not have any effect on your performance either way?"; 5 = definitely hurt my performance, 1 = definitely help my performance). The items were aggregated by taking their unweighted averages.

The end of the study also included an additional behavioural intention measure: a choice between an 'easy review' extra credit assignment and a 'hard challenge' assignment[58,59]. The intervention increased the rate of choosing the challenging assignment by 0.11 s.d. [0.028, 0.200]. We expected the treatment to increase engagement with stressors because it leads to the appraisal that they are opportunities for learning and growth.

## Study 2
**Sample size determination.** All students in an introductory social science course in autumn 2019 were invited to complete the intervention or control materials in return for a small amount of course credit. Sample size was set by the response rate.

**Participants.** Participants were predominately first-year college students attending a selective public university in the United States that drew from a wide range of socioeconomic status groups: 17 years old: 3%; 18 years old: 49%; 19 years old: 29%; 20 years old: 11%: 21 or older: 8%. Sixty-four per cent identified as female and the rest as male; 39% had mothers who did not have a four-year college degree or higher (an indicator of lower socioeconomic status), and 59% identified as lower class, lower middle class or middle class (versus upper middle or upper class).

**Procedure.** This experiment was conducted in a social science course in which students completed timed, challenging quizzes at the beginning of each class meeting, twice per week. In the second week of the semester, soon before the first graded quiz, students were invited to complete the intervention (or control) materials on their own time using their own computer in return for course credit, and 83% of invited students did so. The effects of the intervention were assessed through students' appraisals of the first graded quiz of the semester one to three days later. The appraisal items were necessarily short because they were embedded at the end of the assignment and students completed them during class before the lecture. The appraisal items were then administered a second time after another quiz, which occurred three to four weeks after intervention.

**Measures.** Participants rated their agreement or disagreement with the statements "I felt like my body's stress responses hurt my performance on today's benchmark" (1 = strongly disagree, 5 = strongly agree) and "I felt like my body's stress responses helped my performance on today's benchmark" (5 = strongly disagree, 1 = =strongly agree). The two ratings were averaged to provide an appraisal index, with higher values corresponding to more negative appraisals[60].

## Study 3
**Sample size determination.** An a priori power analysis was used to determine sample size. Previous stress research that assessed cardiovascular responses in laboratory-based stress induction paradigms produced medium to large effect sizes (for example, range: $d = 0.59$ to $d = 1.44$. Based on a standard medium effect size, at the low end of this range ($d = 0.50$), with a two-tailed hypothesis, G*Power indicated that 64 participants per condition (that is, 128 total participants) would be necessary to achieve a target power level of 0.80 to test for basic effects of the treatment using frequentist methods. In anticipation of potential data loss, we determined a priori that we would oversample by 20%. Data collection was terminated the week after more than 150 participants had been enrolled in the study and provided valid data.

**Participants.** Participants were prescreened and excluded for physician-diagnosed hypertension, a cardiac pacemaker, body mass index (BMI) > 30 and medications with cardiac side effects. A total of 166 students were recruited from a university social science subject pool (120 females, 46 males; 76 white/Caucasian, 12 Black/African-American, 17 Latinx, 65 Asian/Asian-American, 2 Pacific Islander, 4 mixed ethnicity, 7 other; mean age = 19.81, s.d. = 1.16, range = 18–26; 32% reported that their mothers did not have a college degree). After data collection, two participants were excluded owing to experimenter errors. In addition, impedance cardiography data for four participants could not be analysed owing to technical issues (prevalence of noise and artefacts in the signals). Decisions about the inclusion of participants were made blind to condition assignment and to levels of the outcome. Participants were compensated US$20 or 2 h of course credit for their participation.

**Procedure.** After intake questions, application of sensors and acclimation to the laboratory environment, participants rested for a 5-min baseline cardiovascular recording that occurred approximately 25 min after arrival at the laboratory. They were then randomly assigned to an intervention condition by the computer software in real time and completed either intervention or control materials, which took approximately 20 min in this sample. Participants then completed the TSST[28]. The TSST asks participants to give an impromptu speech about their personal strengths and weaknesses in front of two evaluators. Evaluators are presented as members of the research team who are experts in nonverbal communication and will be monitoring and assessing the participant's speech quality, ability to clearly communicate ideas and nonverbal signalling. Throughout the speech (and mathematics) epochs of the TSST, evaluators provide negative nonverbal feedback (for example, furrowing brow, sighing, crossing arms and so on) and no positive feedback, either nonverbal or verbal[28]. At the conclusion of speeches, and without prior warning, participants are asked to do mental mathematics (counting backwards from 996 in increments of 7) as quickly as possible in front of the same unsupportive evaluators. Incorrect answers were identified by evaluators, and participants were instructed to begin back at the start. This stress induction procedure is widely used to induce the experience of negative, threat-type stress responses[29,31]. After completion of the TSST task, participants rested quietly for a three-minute recovery recording. Before leaving the laboratory, all participants were debriefed and comforted.

**Physiological measures.** The following measures were collected during baseline and throughout the TSST: ECG, ICG and blood pressure. ECG and ICG signals were sampled at 1,000 Hz, and integrated with a Biopac MP150 system. ECG sensors were affixed in a Lead II configuration. Biopac NICOO100C cardiac impedance hardware with band sensors (mylar tapes wrapped around participants' necks and torsos) were used to measure impedance magnitude ($Z_0$) and its derivative ($dZ/dt$). Blood pressure readings were obtained using Colin7000 systems. Cuffs were placed on participants' non-dominant arm to measure pressure from the brachial artery. Blood pressure recordings were taken at two-minute intervals during baseline, throughout the stress task and during recovery. Blood pressure recordings were initiated from a separate control room. ECG and ICG signals were scored offline by trained personnel. First, one-minute ensemble averages were analysed using MindWare software IMP v.3.0.21. Stroke volume was calculated using the Kubicek method[61]. B- and X-points in the $dZ/dt$ wave, as well as Q- and R-points in the ECG wave, were automatically detected using the maximum slope change method. Then, trained coders blind to condition examined all placements and corrected erroneous placements when necessary.

Analyses targeted three physiological measures: PEP, stroke volume and TPR. This suite is commonly used to analyse threat- versus challenge-type stress responses (for a review, see ref. [62]). TPR is the clearest indicator of threat-type responses and was therefore the focal outcome measure in this research. TPR assesses vascular resistance, and when threatened, resistance increases from baseline[26]. TPR was calculated using the following validated formula: (MAP/CO) × 80 (in which MAP is mean arterial pressure and CO refers to cardiac output; ref. [63]). PEP is a measure of sympathetic arousal and indexes the contractile force of the heart. Shorter PEP intervals indicate greater contractile force and sympathetic activation. Both challenge- and threat-type stress responses are accompanied by decreases in PEP from rest; in some studies, a stronger challenge response has corresponded to an greater decrease in PEP relative to a threat response, signifying greater engagement with the task. Threat versus challenge states differ in PEP values, however, in recovery to baseline, with challenge states corresponding to quicker recovery. Stroke volume is the amount of blood ejected from the heart on each beat (on average per minute). Increases in stroke volume index greater beat-to-beat cardiac efficiency and more blood being pumped through the cardiovascular system, and are often observed in challenge states, as the body spreads more oxygenated blood to the periphery[29]. Decreases in stroke volume, on the other hand, are more frequently observed in threat states (even though threat can also elicit little or no change in stroke volume[64]). Cardiac output, which is stroke volume multiplied by heart rate, is frequently used to assess threat- and challenge-type stress responses as well. As in a past paper[29] we focused on stroke volume rather than cardiac output because the effects of the treatment on PEP (and thus heart rate, a part of the cardiac output formula) could distort effects on cardiac output. For all three measures (TPR, stroke volume and PEP) we computed and analysed reactivity scores by subtracting each person's average levels from the five minutes of the baseline epoch, which occurred before random assignment. Thus, all TPR, PEP and stroke volume results in the paper account for any potential baseline differences that existed before random assignment.

## Study 4

**Sample size determination.** Study 3 showed an ATE for the synergistic mindsets intervention of approximately 0.70 s.d. for TPR reactivity during the first minute of the speech epoch. Assuming an approximately 25% reduction in effect size for a replication study, then to have an 80% likelihood of reliably detecting an ATE of 0.50 s.d. with a one-tailed hypothesis test (because this is a replication study), we calculated that we would need approximately 50 participants per condition. Our stopping rule was to collect data from 200 participants who completed one of the conditions and provided valid TPR data for analysis.

**Participants.** Participants were from the same university pool as study 3 and were recruited using the same protocols and exclusion criteria. A total of 200 students provided valid TPR data (163 females, 37 males; 79 white/Caucasian, 22 Black/African-American, 14 Latinx, 79 Asian/Asian-American, 6 other; Mage = 20.11, s.d. = 1.77, range = 18–32; 32% reported their mothers did not have a college degree).

**Procedure.** Study 4 followed the same procedure as study 3 except for three changes. First, we removed the mathematics epoch to streamline the study for the focal epochs only, so that we could collect data as quickly as possible before a COVID-19 outbreak could shut down data collection. Second, the Qualtrics survey randomized participants to one of four conditions; two were new conditions, and two were the same synergistic mindsets and neutral control conditions that appeared in the other studies (the materials for the two new conditions are posted on the OSF; see the Supplementary Information). Third, we assessed threat and challenge appraisals and well-being at the end of the study.

The first new control condition was a growth-mindset-only condition. This used materials from a previously published growth mindset intervention experiment that was successful at improving the grades of lower-achieving adolescents[65]. The intervention involved reading a scientific article about the brain's potential to grow and learn and answering open-ended questions that encourage students to internalize the information, as described in previous reviews of the literature[66]. It did not discuss stress or encourage stress reappraisals. Replicating previous studies, the growth-mindset-only condition reduced reports of fixed mindset by 0.46 s.d. [−0.64, −0.28], which is within the expected range on the basis of a previous national experiment evaluating a growth mindset intervention (which was 0.33 s.d. (ref. [4])). This condition did not reduce reports of stress-is-debilitating mindsets relative to the neutral control condition; ATE = 0.08 SD [−0.25, 0.41]. Thus, the growth-mindset-only condition faithfully manipulated growth mindset but not stress mindset, as intended.

The second new control condition was a stress-mindset-only condition. This used materials from a previously published stress mindset intervention experiment that was successful at changing stress mindsets and showed mixed effects on stress coping in a longitudinal study[67]. This intervention involved watching videos that explained the concept of stress-is-enhancing mindsets, invited participants to practice reappraising stress and guided them through a vivid imagery reflection exercise to make the stress-is-enhancing mindset message vivid and relatable. As expected, this established stress-mindset-only intervention reduced stress-is-debilitating mindsets by −0.33 s.d. on average [−0.095, −0.56] relative to the neutral control condition, but did not reduce (and perhaps even increased) fixed mindsets; ATE = 0.19 [0.01, 0.40].

**Measures.** The measures for TPR, stroke volume and PEP reactivity were identical to study 3. Two new indices were added for exploratory analyses.

The first exploratory measure assessed self-reports of threat-type (versus challenge-type) appraisals. These are global appraisals of whether people feel like the demands of a stressful situation exceed the resources available to them to cope with the situation (see Fig. 1a). The composite consisted of the unweighted average of items used in previous TSST studies[29] (all items appear in materials posted on the OSF; see the Supplementary Information for links). Several questions measured the perceived demand of the speech task ("The task was very demanding"; "The task was very stressful") and several assessed perceived resources ("I felt that I had the abilities to perform well on the task"; "I believe I performed well on the task"); these were combined into an index corresponding to threat versus challenge appraisals by computing the ratio of perceived demand to perceived resources, following previous research. Next, one question assessed perceived threat ("I felt threatened by the task") and one question assessed perceived

challenge ("I felt that the task challenged me in a positive way"); these too were combined by dividing threat by challenge. Finally, the two ratio scores were combined by taking their unweighted average.

The second additional measure involved items taken from an established measure of well-being: reports of whether people felt that their psychological needs were currently being met[68,69]. Measures assessing threats to psychological needs ("I felt disconnected; rejected; insecure") were reverse-scored and averaged with items assessing satisfaction of psychological needs ("I felt good about myself; liked; powerful"; and "My self-esteem was high") to create an index of positive well-being. Notably, feeling bad about oneself, and reporting low self-esteem, is central to the network of depression symptoms[35]. Therefore, this measure of well-being assesses the presence or lack of immediate post-task internalizing symptoms, and conceptually replicated the results of the field experiment in study 5.

**Pre-registered analysis plan.** The pre-registration called for a focus on TPR reactivity during the most stressful speech epoch. In addition to this primary outcome, we used the pre-registered modelling method to replicate study 3's finding with regard to the effects of the synergistic mindsets treatment on stroke volume (also during the speech epoch) and PEP (during the recovery period). As in study 3, we would have focused on cardiac output rather than stroke volume, but because we again found differences in PEP (a measure of SNS activation), we used the less-contaminated stroke volume measure. Finally, we used the pre-registered BCF method, and same covariates and moderators, to analyse two exploratory outcomes that were not mentioned in the analysis plan: threat appraisals and well-being.

## Study 5

**Sample size determination.** We aimed for a minimum of 100 participants and 1,000 daily diary responses in this field experiment evaluating the synergistic mindsets treatment. We sought to recruit as many as possible before the end of October in the autumn of 2019, because the study was focused on normative stressors at the start of a new school year, and because daily diary data collection could not happen during or after the Thanksgiving break in the United States (which is in late November). The number of students recruited each week was constrained by the research team's capacity to support twice-daily diary surveys and thrice-daily saliva samples in a school environment. The ultimate sample size was determined by the total number of students who could be recruited from the school in the autumn semester of 2019, given these constraints.

**Participants.** Participants were adolescents from economically disadvantaged families (99%); 78% were Black/African-American, 5% were white or Asian, and the remaining students were Hispanic/Latino; 36% were in 9th grade; 34% were in 10th grade; 18% were in 11th grade; and 12% were in 12th grade. Students attended a high-quality urban charter school that showed a high graduation rate (98%) relative to the urban city school district (68%). The teachers at the school were well-trained and motivated, having earned a national distinction for this charter school. This was a meaningful school for a first evaluation study because the synergistic mindsets intervention was not expected to overcome an absence of objective opportunities to learn, but rather to inspire students to take advantage of opportunities for upward mobility.

**Procedure.** Participants were assigned to one of three data collection cohorts on the basis of their academic schedules and available research staff. Cohorts 1, 2 and 3 completed daily diary measures across three consecutive weeks during the autumn term. The intervention was administered on a Thursday, and then students began their weekly daily diary data collection 1–3 weeks later ($M = 14$ days). Intervention materials (see experiment 1) were completed on a tablet computer with headphones in a quiet room at the school. Randomization to conditions occurred at

this time. All data collection was supervised by trained research staff who assisted participants and answered any questions, while being blind to condition assignment and specific hypotheses. Before intervention or control materials, participants completed baseline measures of mindsets (stress mindsets and growth mindsets) along with demographic information.

The week of daily diary data collection began on a Monday and students were surveyed twice each day for five consecutive days through to Friday. Students provided their first self-report at lunch and the second at the conclusion of the school day but before leaving the school's campus. Saliva samples were collected three times per day by adding the morning, before the first class period of the day. Lunchtime samples were collected before students ate. Thus, we targeted 10 total reports for each student and 15 total saliva samples. In addition to occasional non-response, there were two exceptions to these targeted numbers. One cohort had four days of data collection owing to a school-wide event on a Friday, and the first cohort had up to three preliminary days of self-report (not saliva) data collection while the research team was refining procedures. Rather than exclude these additional self-report records, they were included, although the results were the same when excluding them.

The daily diary measures were designed to be brief (around five minutes) and were completed on paper. In the mornings only, students completed brief writing prompts that asked them to reflect on the themes from their respective treatment or control groups. The purpose of the reflections was to collect qualitative data to use in future research and development about how students were using the treatment messages in their daily lives. Students provided their saliva samples either before completing the reflections or simultaneously with them; as noted, at lunch and in the afternoon, students completed their daily stress diaries. Note that although there was a possibility that the morning reflections influenced students' self-reports later in the day, they could not have influenced the saliva samples, because, as noted, salivary samples were collected before or simultaneously with the reflections, and salivary cortisol levels reflect stress responses 30–45 min earlier.

To report daily stressful events, students first checked boxes indicating which of several categories of stressors they experienced that day (for example, friends/social, academics, romantic relationships, daily hassles and so on), then how intense the stressors, combined, were overall ("How negative would you say these experiences were?"; 1 = not negative at all, 5 = extremely negative). Following published standard operating procedures for the diary studies in this laboratory[29], days on which no social-evaluative stressors were listed were coded as a 1 for stressor intensity (the lowest value), to avoid dropping data from those who did not experience a social-evaluative stressor.

Students were compensated US$10 for completing intervention materials, and US$5 for each daily diary entry. Thus, the maximum compensation per participant was US$60. After the conclusion of data collection, students and instructors were debriefed. At the end of the school year, students randomly assigned to the control condition were provided with the mindset intervention.

**Daily negative self-regard.** On each daily survey, students reported daily negative self-regard, an internalizing symptom, operationalized as overall positive or negative feelings about themselves ("Overall, how good or bad did you feel about yourself today?"; 1 = extremely good, 7 = extremely bad). This was a single-item measure owing to the limited respondent time.

**Cortisol.** Acute cortisol responses follow a specific time course (peak levels occur around 30 min after stress onset). However, the diary survey stressors were not calibrated to identify the timing of specific events, so the two sources of information could not be yoked. Indeed, as noted in the main text, there was no association between the intensity of stressors reported and cortisol in the control condition (unlike self-regard and stressor intensity). In addition, levels of cortisol have a diurnal cycle (peak levels at wakening, rapid declines within the first

waking hours and nadir at the end of the day). Waking levels and diurnal slopes can map onto well-being, stress coping and health[70]. Because all sampling was conducted during the school day, waking levels and diurnal cortisol slopes could not be accurately and precisely measured. The lack of time-course specificity and diurnal cycle data means that our reported effect sizes for global cortisol levels are likely to be conservative because noise in the data attenuates effect sizes.

**Academic achievement.** The research team obtained students' transcripts from schools after credits were recorded in the spring of 2020. Credit attainment (that is, whether students passed the course) in core classes (mathematics, science, social studies and English or language arts) were coded. An 'on-track' index[71] was computed for each student (1 = students passed all four of their core classes; 0 = they did not). In addition, following a previous growth mindset intervention study[4], a STEM course on-track indicator was computed (1 = passed mathematics and science; 0 = they did not) as was a non-STEM course on-track indicator (1 = passed social studies and English or language arts; 0 = they did not).

## Study 6

**Sample size determination.** We recruited all students possible from an entire social science class in the spring of 2020, which, we would later learn, was a unique cohort for examining stress during the COVID-19 lockdowns. A minimum of 278 students would be needed to have a greater than 80% chance of detecting a directional effect on anxiety of 0.3 s.d. with a conventional linear model analysis, and more students than this participated.

**Participants, procedure and measures.** Data were collected during the spring semester of 2020. Participants were from the same university as study 2 and the same intervention procedures were followed. (Owing to a difference in data collection procedures relative to study 2, quiz appraisal data could not be collected in study 5). The intervention was delivered at the end of January 2020. In March 2020, students were sent home owing to COVID-19 quarantines. In mid-April 2020, students completed the Generalized Anxiety Disorder-7 (GAD-7)[38] as a part of a class activity focused on psychopathology. The GAD-7 asks "How often have you been bothered by the following over the past 2 weeks?" and offers several symptoms, including "Feeling nervous, anxious, or on edge," "Not being able to stop or control worrying," and "Feeling afraid as if something awful might happen." Each symptom is rated on a scale from 0 ("Not at all") to 3 ("Nearly every day"). The seven items were summed, producing an overall score ranging from 0 to 21, with higher values corresponding to higher levels of general anxiety symptoms.

## Reporting summary

Further information on research design is available in the Nature Research Reporting Summary linked to this paper.

## Data availability

Data are available on the OSF (https://osf.io/3zmqc/).

## Code availability

Syntax files and the multibart package v.0.3, which was used for all Bayesian analyses, are available on the OSF (https://osf.io/3zmqc/).

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

**Acknowledgements** This research was supported by funding provided to the University of Texas at Austin by the National Institutes of Health under award number R01HD084772-01, and grant P2CHD042849, and by the National Science Foundation under grant numbers 1761179 and 2046896. Funding was also provided to the University of Rochester by Google Empathy Lab. We acknowledge the following individuals for contributing to the research: A. Audette, A. Rank, F. Medrano, H. Y. Lee, J. O'Brien, E. Seo, S. Gosling, J. Pennebaker, C. Kirshbaum, C. S. Dweck and T. Manchester. We thank S. Talamus, A. Duckworth and the CLRN for supporting data collection for study 1.

**Author contributions** D.S.Y., C.J.B. and J.P.J. jointly conceived of the studies and led their design. D.S.Y., C.J.B., D.K.C. and M.J. developed the intervention materials. J.P.J. and H.G. led the collection of data for studies 3, 4 and 5. M.J. and H.G. processed and merged those data. D.S.Y., J.J.G., C.J.B., D.K.C. and J.P.J. developed the conceptual framework. D.S.Y. and J.J.G. wrote the first draft of the paper. C.J.B. and J.P.J. authored sections of the paper and provided critical edits and revisions to the rest. P.S. and J.S.M. developed the BCF method used here, conducted analyses for studies 3–6, authored descriptions of the statistical methodology and, with D.S.Y., developed figures and data visualizations for those studies. D.S.Y. analysed the data for studies 1 and 2 using methods developed by P.S. and J.S.M.

**Competing interests** The authors declare no competing interests for this study. D.S.Y. has disseminated growth mindset research to public audiences and has complied with institutional financial disclosure requirements; no financial conflicts of interest have been identified. D.K.C. is employed by Google, which owns technology products designed to support well-being, but does not currently make or sell any product based on the research presented here. None of the study funders accessed the raw data, nor did they influence the data collection, analysis or reporting.

**Additional information**
**Correspondence and requests for materials** should be addressed to David S. Yeager, Christopher J. Bryan, Jared S. Murray, Pedro H. F. Santos or Jeremy P. Jamieson.

**A)** TPR reactivity by mindset subgroup

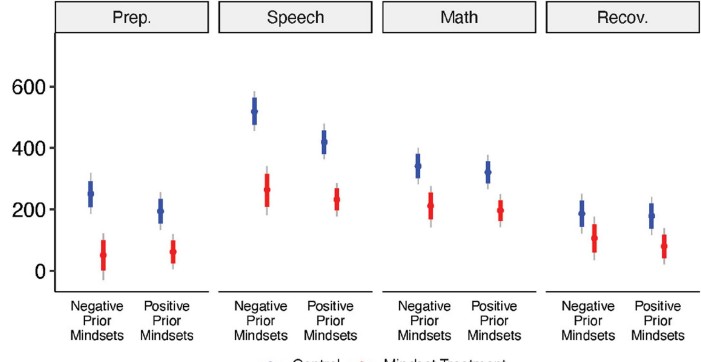

**C)** CATEs by prior mindset subgroup

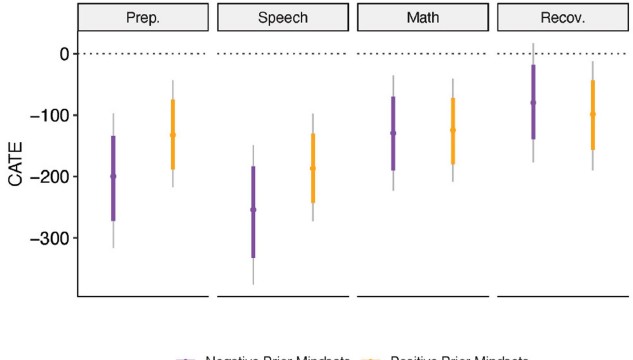

**B)** Moderating effect of prior mindsets on TPR reactivity effects during speech epoch

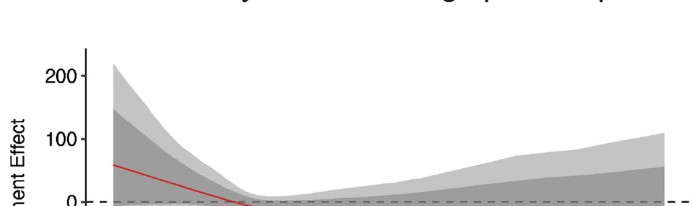

**D)** Treatment x prior mindset interactions for TPR reactivity by epoch

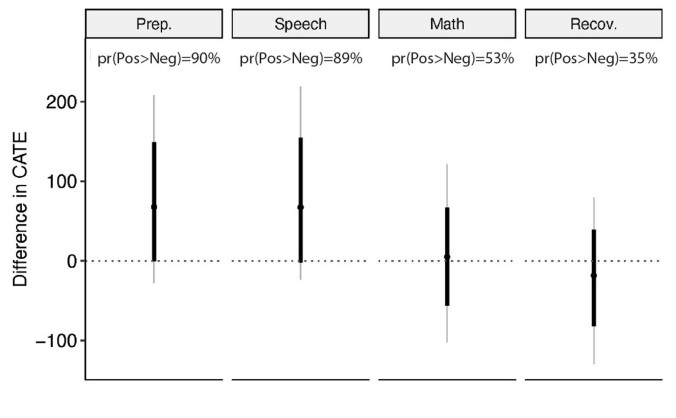

**Extended Data Fig. 1 | In study 3, prior mindsets moderated the treatment effect on TPR during stressful TSST epochs.** In (A), the expected value of TPR reactivity for each epoch and for each prior mindset group, by condition, in (B), an additive summary of the posterior distribution of treatment effects, by negative prior mindset levels, in (C) the conditional average treatment effects (CATEs) for each prior mindset subgroup for each epoch, and in (D) the interaction between treatment and prior mindsets on TPR responses across TSST epochs. Note: TPR = total peripheral resistance (in dyne-sec x cm$^5$). Dots correspond to the expected values (**a**), CATEs (**b**), and average of the posterior distribution of a difference in CATEs (**C**) estimated with the Bayesian algorithm. Thick lines represent the 10th to 90th %iles of the posterior

distribution; grey lines represent the 2.5th to 97.5th %iles. ATE = average treatment effect. In (B), the red line corresponds to the expected partial treatment effect, which corresponds to the offset from the average treatment effect (ATE) at each level of the moderator, holding other potential moderators constant; the dark band is the 10th to 90th percentile of the posterior distribution and the light band is the 2.5th to 97.5th %iles. The prior mindset subgroups used to display treatment effects in (A), (C) and (D) were identified by implementing a hands-off Bayesian decision-making algorithm that maximized the differences among the mindset groups in terms of the outcome, without using information on the magnitudes of the treatment effects (see SI online). Control $n$ = 86, Treatment $n$ = 74.

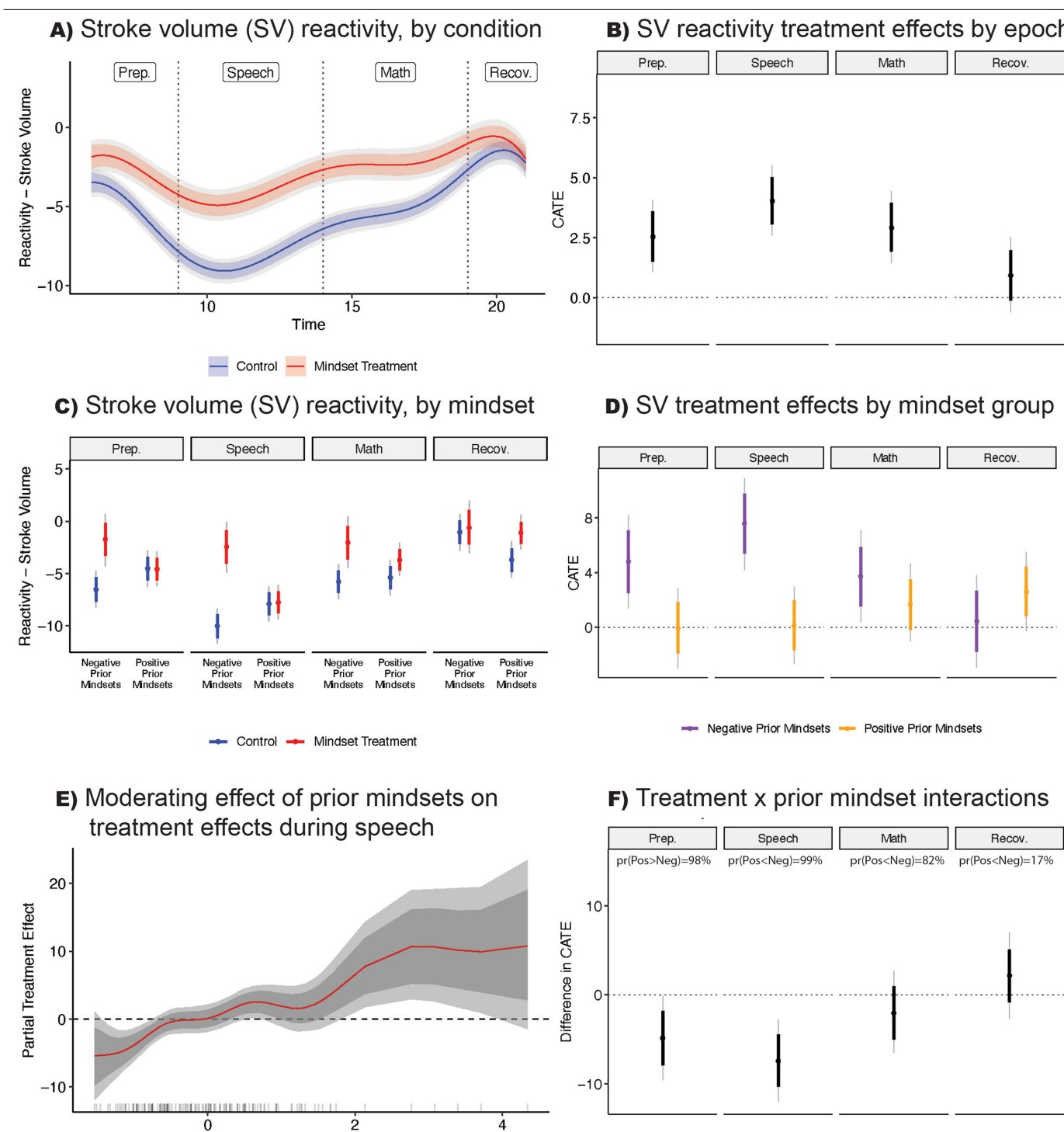

**A) Stroke volume (SV) reactivity, by condition**

**B) SV reactivity treatment effects by epoch**

**C) Stroke volume (SV) reactivity, by mindset**

**D) SV treatment effects by mindset group**

**E) Moderating effect of prior mindsets on treatment effects during speech**

**F) Treatment x prior mindset interactions**

**Extended Data Fig. 2 | In study 3, the synergistic mindsets intervention improved cardiovascular responses to the TSST.** Effects of the intervention on stroke volume (SV)−the amount of blood ejected from the heart during each beat, in *ml*−were tested because challenge (relative to threat) responses increase SV to facilitate actively addressing stressors[25,26,29]. Thus, we anticipated those experiencing challenge-type stress during the stressful TSST epochs should exhibit relatively higher stroke volumes as their bodies distribute oxygenated blood to optimize performance, whereas threatened individuals were expected to have lower stroke volumes during stressful epochs of the TSST as their bodies seek to concentrate blood in the core. SV values reported here are reactivity scores, which means that the average of the 5 min during the baseline epoch were subtracted from each. In (A) the darkest lines correspond to the expected value of the outcome, estimated in the Bayesian model. Dots correspond to the ATEs (B), expected values (C),

CATEs (D), and average of the posterior distribution of a difference in CATEs (F) estimated with the Bayesian algorithm. ATE = average treatment effect. In (E), the red line corresponds to the expected partial treatment effect, which corresponds to the offset from the average treatment effect (ATE) at each level of the moderator, holding other potential moderators constant. In all panels, thick bands represent the 10th to 90th %iles of the posterior distribution; the lightest/grey lines represent the 2.5th to 97.5th %iles. In (B), ATE for Prep = 2.5 ml [1.5, 3.6], Speech = 4.0 ml [3.1, 5.0], Math = 2.9 [1.9, 4.0], Recovery = 0.9 [−.1, 2.0]. In (C), (D) and (F), the prior mindset subgroups used to display the different treatment effects were generated by implementing a hands-off Bayesian decision-making algorithm that maximized the differences among the mindset groups in terms of the outcome, without using information on the magnitudes of the treatment effects. Control *n* = 86, Treatment *n* = 74.

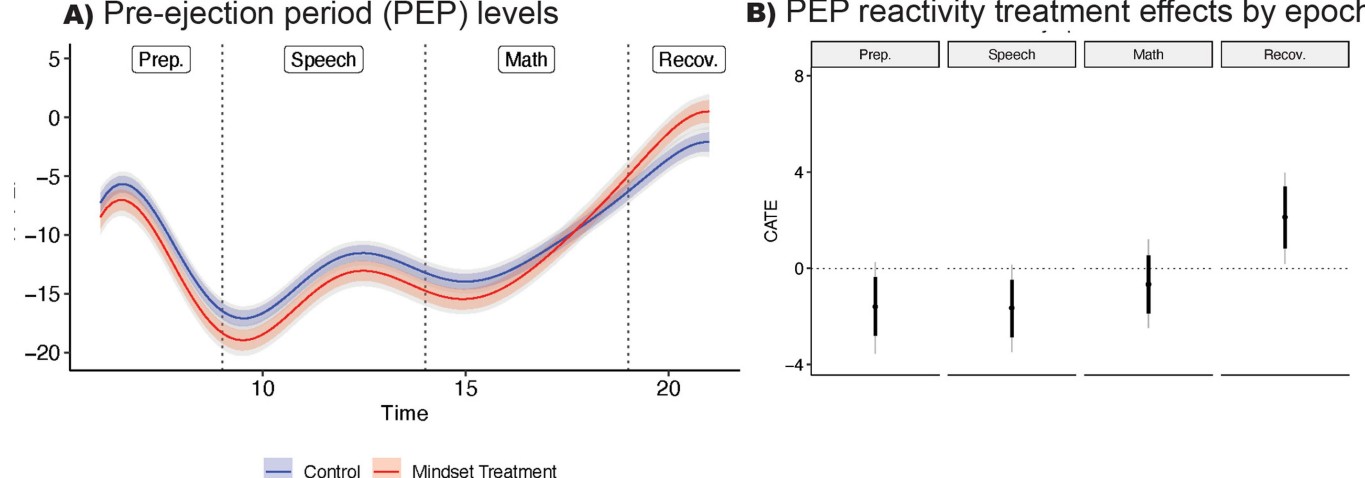

**Extended Data Fig. 3 | In study 3, the effect of the synergistic mindsets intervention on PEP reactivity in milliseconds across TSST epochs.**
Pre-ejection period (PEP)—which assesses the contractile force of the heart by measuring the time from onset of ventricular depolarization to aortic valve opening—was examined to test for effects of the intervention on sympathetic nervous system (SNS) arousal. Challenge responses evoke more rapid onset of SNS arousal during a stressor and more rapid recovery to homeostasis after stress offset. Threat-type responses are associated with sustained vigilance for sources of harm and prolonged stress responses, thus threat is associated with slower recovery to baseline after stress offset[17,64]. Whereas all participants should show PEP decreases (leading to a more rapid heart rate) relative to

baseline during the stressful epochs[29,64] (see Fig. 1), condition differences are expected to emerge during the recovery period, because controls should be slower to return to homeostasis relative to treated individuals. In (A) the darkest lines correspond to the expected value of the outcome, estimated in the Bayesian model. In (B), dots correspond to the ATEs. ATE = average treatment effect. In both panels, thick bands represent the 10th to 90th %iles of the posterior distribution; the lightest/grey lines represent the 2.5th to 97.5th %iles. In (B), a positive treatment effect of 2.13 ms [0.8, 3.4] was found during the recovery epoch, as expected[29]. PEP values reported here are reactivity scores, which means that the average of the 5 min during the baseline epoch were subtracted from each. Control $n$ = 86, Treatment $n$ = 74.

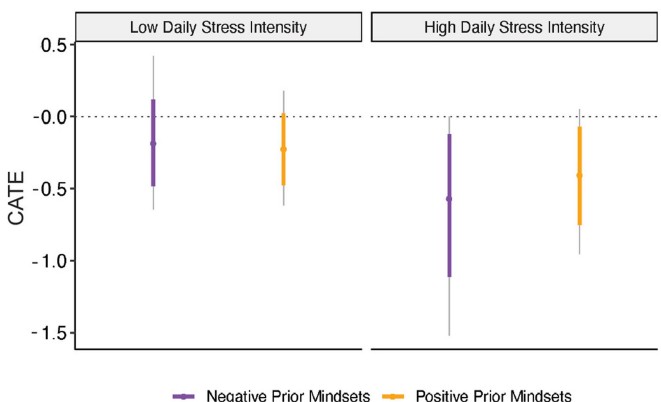

**A)** Negative self-regard, by mindset group

**B)** Treatment effects by mindset group

**Extended Data Fig. 4 | In study 5, the synergistic mindsets intervention reduced daily negative self-regard relative to controls the most among people with negative prior mindsets, on their most highly stressful days.** Note: $n = 119$, $n \le 1,213$ observations. In (A), dots correspond to the average expected value of the outcome, and in (B) dots correspond to the CATEs, estimated by the Bayesian algorithm. CATE = Conditional Average Treatment Effect. In both panels, thick bands represent the 10th to 90th %iles of the posterior distribution; the lightest/grey lines represent the 2.5th to 97.5th %iles. The CATEs are: Low Daily Stress Intensity, Negative Prior Mindsets

CATE = −.19 [−.48, .12], Positive Prior Mindsets CATE = −.23 [−.48, .022]; High Daily Stress Intensity, Negative Prior Mindsets CATE = −.57 [−1.11, −.12], Positive Prior Mindsets CATE = −.41 [−.75, −.07]. Hence, the CATE was 40% for negative prior mindsets participants on high-stress days relative to positive prior mindsets participants. The prior mindset subgroups used to display different treatment effects were generated by implementing a hands-off Bayesian decision-making algorithm that maximized the differences among the mindset groups in terms of the outcome, without using information on the magnitudes of the treatment effects. Control $n = 58$, Treatment $n = 61$.

**A)** Moderating effect of negative mindsets

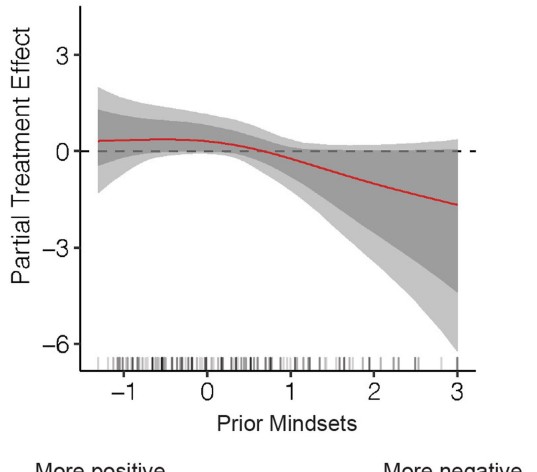

More positive          More negative

**B)** Treatment effects by mindset subgroup

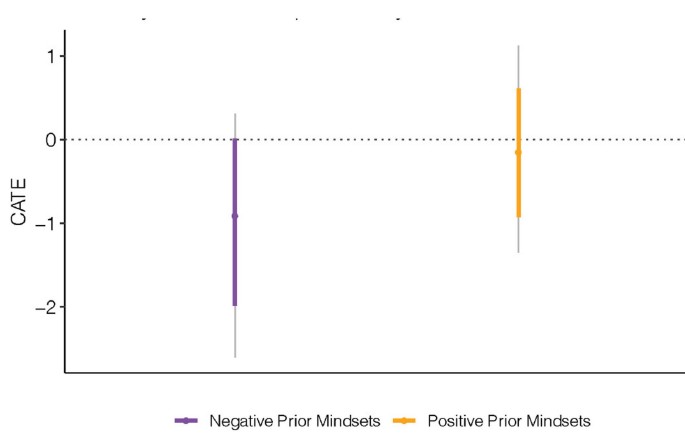

— Negative Prior Mindsets   — Positive Prior Mindsets

**Extended Data Fig. 5 | In study 6, an additive summary of the posterior distribution of treatment effects shows greater reductions in anxiety in response to the treatment among those with negative prior mindsets, and this same result is supported when examining the CATEs for positive and negative prior mindsets.** *Note:* In (A), the red line corresponds to the expected partial treatment effect, which corresponds to the offset from the average treatment effect (ATE) at each level of the moderator, holding other potential moderators constant, estimated in the Bayesian algorithm. In (B) dots correspond to the CATEs, estimated by the Bayesian algorithm. CATE = Conditional Average Treatment Effect. In both panels, thick bands represent the 10th to 90th %iles of the posterior distribution; the lightest/grey lines represent the 2.5th to 97.5th %iles. Control *n* = 172, Treatment *n* = 179.

**Extended Data Table 1 | Negative prior mindsets predicted outcomes in the control condition in five experiments**

| Outcome | Correlations of self-reported negative prior mindsets (fixed mindset and stress-is-debilitating mindset) with self-reported outcomes in the control condition, $r =$ | $df =$ |
|---|:---:|:---:|
| **Study 1** | | |
| Negative event appraisals | **.26** | 1388 |
| Negative response appraisals | **.16** | 1382 |
| **Study 2** | | |
| Quiz#1 appraisals | Prior mindsets not assessed | NA |
| Quiz#2 appraisals | Prior mindsets not assessed | NA |
| **Study 4** | | |
| Threat-type task appraisals | **.24** | 42 |
| Well-being | **-.38** | 42 |
| **Study 5** | | |
| Daily negative self-regard | **.27** | 523 |
| **Study 6** | | |
| Anxiety symptoms | **.38** | 170 |

Negative prior mindsets are a multiplicative term of event- and response-focused mindset measures assessed before the intervention. In study 1, negative event and response appraisals refer to appraisals that the stressor and the response are harmful and uncontrollable. In study 4, threat-type task appraisals refers to the global assessment that task demands during the TSST exceeded perceived resources, and well-being refers to reports of self-esteem, feeling liked, powerful and good about oneself, and fewer reports of feeling insecure. Self-report outcomes are prioritized here because physiological indicators of challenge or threat stress responses rarely correlate with self-reported measures[27].

**Extended Data Table 2 | Treatment effect estimation with traditional linear regression analysis and classical null hypothesis testing reproduces the primary findings from each of the six studies**

| Outcome | Treatment effect (in *SD* units) | *se=* | *t =* | *df =* | *p =* |
|---|---|---|---|---|---|
| **Study 1** | | | | | |
| Negative mindsets | **-0.293** | 0.037 | 7.839 | 2530 | **<.001** |
| Event appraisals | **0.132** | 0.037 | 3.619 | 2539 | **<.001** |
| Response appraisals | **0.207** | 0.038 | 5.425 | 2530 | **<.001** |
| **Study 2** | | | | | |
| Negative mindsets | **-0.489** | 0.077 | 6.325 | 620 | **<.001** |
| Quiz#1 appraisals | **0.410** | 0.074 | 5.551 | 672 | **<.001** |
| Quiz#2 appraisals | **0.223** | 0.076 | 2.949 | 659 | **.006** |
| **Study 3** | | | | | |
| Negative mindsets | **-0.646** | 0.151 | 4.292 | 158 | **<.001** |
| TPR reactivity | **-0.591** | 0.129 | 4.575 | 151 | **<.001** |
| SV reactivity | **0.445** | 0.130 | 3.414 | 154 | **<.001** |
| PEP reactivity | **0.206** | 0.070 | 2.947 | 151 | **.002** |
| **Study 4** | | | | | |
| Negative mindsets | **-0.702** | 0.212 | 3.307 | 195 | **<.001** |
| TPR reactivity | **-0.500** | 0.236 | 2.116 | 200 | **.022** |
| SV reactivity | **0.508** | 0.226 | 2.249 | 195 | **.013** |
| PEP reactivity | **0.421** | 0.122 | 3.439 | 195 | **<.001** |
| Threat-type appraisals | **-0.696** | 0.219 | 3.186 | 192 | **<.001** |
| Well-being | **0.511** | 0.221 | 2.311 | 192 | **.011** |
| **Study 5** | | | | | |
| Negative mindsets | **-0.409** | 0.183 | 2.229 | 113 | **.014** |
| Negative self-regard on high-stress days | **-0.334** | 0.139 | 2.403 | 119 | **.009** |
| Salivary cortisol | **-0.283** | 0.103 | 2.757 | 112 | **.003** |
| Core course pass rates | **0.396 (20 pp)** | 0.184 | 2.153 | 115 | **.017** |
| **Study 6** | | | | | |
| Negative mindsets | **-0.585** | 0.102 | 5.723 | 349 | **<.001** |
| Anxiety: Main effect | **-0.327** | 0.131 | 2.510 | 345 | **.006** |
| Anxiety: Treatment × Negative prior mindsets | **-0.291** | 0.094 | 3.102 | 345 | **.001** |
| Anxiety: Simple effect, +1SD Negative mindset | **-0.625** | 0.183 | 3.417 | 345 | **<.001** |

*p* = one-tailed *P* value, owing to directional hypotheses. No adjustments were made to *P* values. TPR = total peripheral resistance; SV = stroke volume; PEP = pre-ejection period; pp = percentage points. The study 3 and 4 TPR, SV and PEP models, and the study 5 self-regard and cortisol models, were estimated using linear mixed effects modelling; the remaining models were ordinary one-level linear regressions. Study 3 included all active epochs, because the epoch of interest was not pre-registered in that study, and study 4 included only the pre-registered speech epoch. In study 5, high-stress days refers to days with social-evaluative stressors with an intensity rating from 2 to 4 (on a 0 to 4 point scale). In study 5, the results for cortisol correspond to morning cortisol; the ATE for all three times of day (not just morning) was −0.20 *SD*, *t*(112)=2.208, *P*=0.0146. In study 6, the negative prior mindsets variable is the multiplicative term of prior stress and fixed mindsets.

# Reporting Summary

## Statistics

For all statistical analyses, confirm that the following items are present in the figure legend, table legend, main text, or Methods section.

| n/a | Confirmed | |
|---|---|---|
| ☐ | ☒ | The exact sample size (*n*) for each experimental group/condition, given as a discrete number and unit of measurement |
| ☐ | ☒ | A statement on whether measurements were taken from distinct samples or whether the same sample was measured repeatedly |
| ☐ | ☒ | The statistical test(s) used AND whether they are one- or two-sided<br>*Only common tests should be described solely by name; describe more complex techniques in the Methods section.* |
| ☐ | ☒ | A description of all covariates tested |
| ☐ | ☒ | A description of any assumptions or corrections, such as tests of normality and adjustment for multiple comparisons |
| ☐ | ☒ | A full description of the statistical parameters including central tendency (e.g. means) or other basic estimates (e.g. regression coefficient) AND variation (e.g. standard deviation) or associated estimates of uncertainty (e.g. confidence intervals) |
| ☐ | ☒ | For null hypothesis testing, the test statistic (e.g. *F*, *t*, *r*) with confidence intervals, effect sizes, degrees of freedom and *P* value noted<br>*Give P values as exact values whenever suitable.* |
| ☐ | ☒ | For Bayesian analysis, information on the choice of priors and Markov chain Monte Carlo settings |
| ☐ | ☒ | For hierarchical and complex designs, identification of the appropriate level for tests and full reporting of outcomes |
| ☐ | ☒ | Estimates of effect sizes (e.g. Cohen's *d*, Pearson's *r*), indicating how they were calculated |

*Our web collection on statistics for biologists contains articles on many of the points above.*

## Software and code

Policy information about availability of computer code

| Data collection | None |
|---|---|
| Data analysis | The open source software, Multibart 0.3, and syntax for each study, are available at https://osf.io/3zmqc/. |

For manuscripts utilizing custom algorithms or software that are central to the research but not yet described in published literature, software must be made available to editors and reviewers. We strongly encourage code deposition in a community repository (e.g. GitHub). See the Nature Portfolio guidelines for submitting code & software for further information.

## Data

Policy information about availability of data

All manuscripts must include a data availability statement. This statement should provide the following information, where applicable:
- Accession codes, unique identifiers, or web links for publicly available datasets
- A description of any restrictions on data availability
- For clinical datasets or third party data, please ensure that the statement adheres to our policy

| The data are available on osf.io (https://osf.io/3zmqc/). |
|---|

# Field-specific reporting

Please select the one below that is the best fit for your research. If you are not sure, read the appropriate sections before making your selection.

☐ Life sciences　　☒ Behavioural & social sciences　　☐ Ecological, evolutionary & environmental sciences

For a reference copy of the document with all sections, see nature.com/documents/nr-reporting-summary-flat.pdf

# Behavioural & social sciences study design

All studies must disclose on these points even when the disclosure is negative.

| Study description | Six between-subjects randomized, controlled intervention experiments. |
|---|---|
| Research sample | All samples were adolescents in secondary and post-secondary education and were chosen because they were expected to be undergoing social-evaluative stressors that are common to formal educational settings. The samples in each study were not representative of the U.S. population but they were representative, universal samples of their respective schools (i.e. no selection criteria were applied when inviting participants to join the study, except for the medical criteria listed for Studies 3 and 4). Study 1: Participants were from a large, heterogeneous sample of adolescents who were evenly distributed across grades 8 to 12 in 35 U.S. public schools (13 y/o: 16%; 14: 20%; 15: 20%; 16: 21%; 17: 18%; 18: 5%). The schools were sampled from a stratum of large, diverse, suburban and urban public schools in the southeast United States. Forty-nine percent of adolescents identified as male, 49% as female, and 2% as gender non-binary. Participants were racially and ethnically diverse (participants could indicate multiple racial/ethnic identities so numbers exceed 100%): Black: 20%; Latinx: 39%; White: 68%; Asian: 7%. Participants were also socioeconomically diverse: 40% received free or reduced-price lunch, an indicator of low family income. Study 2: Participants were predominately first-year college students attending a selective public university in the United States that drew from a wide range of socioeconomic status groups: 17 years-old: 3%; 18: 49%; 19: 29%; 20: 11%: 21 or older: 8%. Sixty-four percent identified as female and the rest as male; 39% had mothers who did not have a four-year college degree or higher (an indicator of lower socioeconomic status), and 59% identified as lower class, lower middle class, or middle class (vs. upper middle or upper class). Study 3: Participants were prescreened and excluded for physician-diagnosed hypertension, a cardiac pacemaker, BMI > 30, and medications with cardiac side effects. A total of 166 students were recruited from a university social science subject pool (120 females, 46 males; 76 White/Caucasian, 12 Black/African-American, 17 Latinx, 65 Asian/Asian-American, 2 Pacific Islander, 4 Mixed Ethnicity, 7 Other; $M_{age}$ = 19.81, SD = 1.16, range = 18–26; 32% reported their mothers did not have a college degree). Study 4: Participants were from the same university pool as Study 3 and were recruited using the same protocols and exclusion criteria. A total of 200 students provided valid TPR data (163 females, 37 males; 79 White/Caucasian, 22 Black/African-American, 14 Latinx, 79 Asian/Asian-American, 6 Other; $M_{age}$ = 20.11, SD = 1.77, range = 18–32; 32% reported their mothers did not have a college degree). Study 5: Participants were adolescents from economically-disadvantaged families (99%); 78% were Black/African-American, 5% were White or Asian, and the remaining students were Hispanic/Latino; 36% were in 9th grade; 34% were in 10th grade; 18% were in 11th grade; 12% were in 12th grade. Students attended a high-quality urban charter school which showed a high graduation rate (98%) relative to the urban city school district (68%). The teachers at the school were well-trained and motivated, having earned a national distinction for this charter school. Therefore, the synergistic mindsets intervention was not expected to overcome an absence of objective opportunities to learn, but rather to inspire students to take advantage of the opportunities for upward mobility. Study 6: Data were collected during the Spring semester of 2020. Participants were from the same university as Study 2, and the demographics were nearly identical to Study 2, and the same intervention procedures were followed. |
| Sampling strategy | All students used convenience sampling methods, with the exception of Study 1, which used a stratified sampling method within the Character Lab Research Network's pool. As noted above, all samples were universal, in that no restrictions were placed on participation in the study (i.e. participants were not screened out for inability to read or speak English, learning disabilities, etc.). In Studies 3-4, participants were prescreened and not invited to participate in the study for physician-diagnosed hypertension, a cardiac pacemaker, BMI > 30, and medications with cardiac side effects. Here is how sample sizes were determined: Study 1: We requested a "fully-powered" sample from CLRN and the exact sample was determined by CLRN. This sample size was planned to have sufficient power to detect a treatment effect in a field experiment of .10 SD or greater, with .10 SD being the minimum effect size that we would interpret as meaningful for a study focused on immediate post-test self-reports. Study 2: All students in an introductory social science course in Fall 2019 were invited to complete the intervention or control materials in return for a small amount of course credit. Sample size was set by the response rate. Study 3: An a priori power analysis was used to determine sample size. Previous stress research that assessed cardiovascular responses in laboratory-based stress induction paradigms produced medium to large effect sizes (e.g., range: d = .59 to d = 1.44 in Yeager et al., 2016, Jamieson et al., 2012, Oveis et al., 2020). Based on a standard medium effect size, at the low end of this range (d = 0.50), with a two-tailed hypothesis, G*Power indicated that 64 participants per condition (i.e., 128 total participants) would be necessary to achieve a target power level of .80 to test for basic effects of the treatment using frequentist methods. In anticipation of potential data loss, we determined a priori that we would oversample by 20%. Data collection was terminated the week after more than 150 participants had been enrolled in the study and provided valid data. Study 4: Study 3 showed an ATE for the synergistic mindsets intervention of approximately .70 SD for TPR reactivity during the first minute of the speech epoch. In this preregistered replication of Study 3, assuming an approximately 25% reduction in effect size for a replication study, then to have an 80% likelihood of reliably detecting an ATE of .50 SD with a one-tailed hypothesis test (because this is a replication study), we calculated that we would need approximately 50 participants per condition. Our stopping rule was to collect data from 200 participants who completed one of the conditions and provided valid TPR data for analysis. Study 5: We aimed for a minimum of 100 participants and 1,000 daily diary responses in this first-ever field experiment evaluating the synergistic mindsets treatment. We sought to recruit as many as possible before the end of October in the fall of 2019, because the study was focused on normative stressors at the start of a new school year, and because daily diary data collection could not happen during or after the Thanksgiving break in the U.S. (which is in late November). The number of students recruited each week was constrained by the research team's capacity to support twice-daily diary surveys and thrice-daily saliva samples in a school environment. The ultimate sample size was determined by the total number of students who could be recruited from the school in |

| | |
|---|---|
| | the fall semester of 2019, given these constraints. Study 6: We recruited all students possible from an entire social science class in the spring of 2020, which, we would later learn, was a unique cohort for examining stress during the COVID-19 lockdowns. A minimum of 278 students would be needed to have a greater than 80% chance of detecting a directional effect on anxiety of .3 SD with a conventional linear model analysis, and more students than this participated. |
| Data collection | In all studies, experimenters, teachers, and anyone else present were kept blind to study hypotheses and condition assignment (which was determined randomly by the survey software). Data collection occurred via web-based surveys in Qualtrics in all studies, except for Studies 3 and 4, which also involved laboratory measures of cardiovascular responding, and Study 5, which involved paper data collection for daily surveys as well as daily saliva sampling. In Study 1, students completed surveys at scheduled times as a part of their participation in CLRN; about half of students were doing remote learning due to COVID-19, and so they completed the surveys at home, and about half completed them in the computer labs. In Study 2, students completed the intervention and the quiz appraisals on their own computers before class (the intervention) or during class (quiz appraisals); because it is a synchronous online class, students were not in the same room as other participating students except in rare circumstances where students watched the course together. In Studies 3 and 4, only the experimenters and the TSST confederates were in the experimental room with the participants; all were blind to condition assignment. In Study 5, the experimenters were present while participants completed the intervention and the daily surveys; Students completed the interventions individually but with peer participants in the room, due to space limitations at the school; Students completed the daily diaries together with the other students who were participating in a given week; students were not allowed to talk about the study with each other during the data collection sessions. In Study 6, the same procedures as Study 2 were used, except students completed the outcome variable, anxiety symptoms, wherever they were quarantining in April of 2020. |
| Timing | Study 1 was conducted in the Fall of 2020, during school closures due to COVID-19; Studies 2, 3, and 5 were in the Fall of 2019, prior to the pandemic; Study 4 was conducted in the Fall of 2021, between the Delta variant and the Omicron variant outbreaks; Study 6 was conducted in the Spring of 2020 (from Jan to April). |
| Data exclusions | Data exclusion rules followed the lab's standard operating procedures and the preregistrations (for studies 1, 2, and 4) and were not changed across studies. Participants were included if they had condition information and provided valid data on the relevant outcome and, for models including moderators, the relevant moderators. That is, all data were included provided that the key variables were present to estimate the model. No additional data exclusions were carried out for studies 1, 2, 5, and 6. In the two laboratory studies, Studies 3 and 4, participants were excluded if the cardiovascular sensors were detached (e.g. if the participant made a sudden movement that disconnected the sensor) or if there were obvious statistical artifacts, as described in the methods for Study 3; this led to the exclusion of 4 participants. In Study 4, because the preregistered primary outcome is TPR reactivity during the speech epoch, participants needed to have at least one minute's worth of TPR data and at least one minute's worth of TPR during the speech epoch. Data were collected until we met the stopping rule of 200 participants with useable data; no data were analyzed with respect to condition effects until that was reached. Thus, in Studies 3/4, all decisions about the TPR data were made blind to condition assignment. |
| Non-participation | We defined adherence as completion of the last page of the intervention. In the studies where participants were closely supervised by researchers (Studies 3, 4, and 5), adherence was high (97% to 99%). In the studies where the intervention was self-administered with no supervision, adherence was lower but still acceptable: 85%, 88% and 82% for Studies 1, 2, and 6, respectively. Because we conducted intent-to-treat analyses, participants were retained in the analytic sample regardless of intervention completion status. |
| Randomization | Randomization happened via the Qualtrics survey in real time as participants completed the online intervention materials, at the individual level. |

# Reporting for specific materials, systems and methods

We require information from authors about some types of materials, experimental systems and methods used in many studies. Here, indicate whether each material, system or method listed is relevant to your study. If you are not sure if a list item applies to your research, read the appropriate section before selecting a response.

## Materials & experimental systems

| n/a | Involved in the study |
|---|---|
| ☒ | Antibodies |
| ☒ | Eukaryotic cell lines |
| ☒ | Palaeontology and archaeology |
| ☒ | Animals and other organisms |
| ☐ | ☒ Human research participants |
| ☒ | Clinical data |
| ☒ | Dual use research of concern |

## Methods

| n/a | Involved in the study |
|---|---|
| ☒ | ChIP-seq |
| ☒ | Flow cytometry |
| ☒ | MRI-based neuroimaging |

# Human research participants

Policy information about studies involving human research participants

| | |
|---|---|
| Population characteristics | These are listed in the "research sample" portion of the reporting summary above. |
| Recruitment | These are listed in the "research sample" and "sampling strategy" portion of the reporting summary above. Note that the study sites are diverse: Study 1: large urban and suburban school districts; Studies 2 and 6: a large, public university; Studies 3 and 4: A private university; Study 5: An urban public charter school serving students experiencing poverty. Nevertheless, |

the sampled schools were not a random sample of potential schools, and further research will be needed to test the generalizability of the intervention, and to further identify the moderating factors (See Bryan, Tipton, & Yeager, 2021, NHB).

Ethics oversight

Approvals for these studies were obtained from the Institutional Review Boards at the University of Rochester or the University of Texas at Austin. In Studies 2 to 6 active consent were obtained either in writing or through the web-based survey; In Study 1, active student assent was obtained via the CLRN standard operating procedures.

Note that full information on the approval of the study protocol must also be provided in the manuscript.

