## [Peer Review File · Nature]

Manuscript Title: A Synergistic Mindsets Intervention Protects Adolescents from Stress

Reviewer Comments & Author Rebuttals

Reviewer Reports on the Initial Version:

Referee #1 (Remarks to the Author):

This manuscript reports on a series of randomized-controlled evaluations of a novel, single-session, easily scalable intervention to promote adolescents' positive mindsets about stressors (growth mindset and "challenge" vs. "threat" appraisals) and stress responses (stress responses can be helpful), which the authors term a synergistic mindsets intervention. Five sequential studies revealed intervention effects consistent with each step in the authors' theory of change: (1) positive changes in growth mindset and helpful stress appraisals post-intervention (immediately and at a 3-week delay), (2) reductions in total peripheral resistance (TPR; a cardiovascular index of threat reactivity) during a well-validated laboratory-based stress task, (3) reduced daily internalizing symptoms (negative self-regard) roughly 2 weeks post-intervention, particularly on days with high-intensity stressors, (4) reduced cortisol levels throughout the school day roughly 2 weeks post-intervention, and (5) lower generalized anxiety symptoms during the early stages of the COVID-19 lockdown (3 months post-intervention) among individuals with negative mindsets at baseline. The authors additionally found that treatment effects on cardiac stress reactivity and negative self-regard on high-stress days were greater for those with negative mindsets at baseline, as would be expected based on the mechanistic theory of change.

This work is novel in that the intervention combines two features of interventions that have not previously been combined: growth mindset and stress reappraisal. The work is methodologically strong and compelling, including the use of intent-to-treat analyses as a conservative approach for estimating intervention effects and sophisticated statistical modeling using Bayesian Causal Forest regressions. Sample sizes are justified by a priori power analyses where possible and two of the

studies were pre-registered for scientific transparency. The authors' conclusions are well supported by the data. The robustness of the conclusions is supported by focusing on effect sizes and the posterior distribution of treatment effects as opposed to focusing on statistical significance as determined by an arbitrary p-value cutoff. While many effect sizes are relatively small, the findings are impressive in their consistency across studies, providing evidence of their reliability. Moreover, such a minimal intervention would be expected to produce very small effect sizes on distal outcomes such as daily cortisol levels or generalized anxiety symptoms. Given the ease with which this intervention could be implemented at a large scale, the findings have important public health implications.

The authors admit that additional research will be needed to further delineate for whom the intervention is effective, and the conditions under which its impacts on distal outcomes are maintained over time.

Relevant prior work is appropriately cited, the paper is extremely well written, and figures are well utilized.

I recommend this article for publication following minor revisions to address several concerns and some minor comments, listed below.

Concerns:

- 1) I am uncertain why the title and abstract focus on "social stress," whereas the intervention and findings are not limited to adolescents' responses to social stressors.
- 2) I did not see information about the specific covariates or potential moderators that were entered into each of the BCF models.
- 3) Study 4: "Internalizing symptoms" were operationalized as participants' responses to one question on each daily survey: "Overall, how good or bad did you feel about yourself today?" Although this is an internalizing symptom (low self-regard), I feel it is slightly misleading to describe this measure as "internalizing symptoms" which implies a more comprehensive measure of various anxious and depressive symptoms. In Fig 4, this measure is referred to as "negative self-regard" and I feel this is a more accurate descriptor of the construct being measured and should be used throughout the manuscript.

Minor comments:

- 1) Effects on Cognitive Appraisals, line 200: The word "and" appears to be missing in the following parenthetical phrase: "which occurred 1-3 days post-intervention was not mentioned in the intervention content."
- 2) Effects on Physiological Responses, line 240: The authors state that "treated adolescents recovered to baseline more quickly after stress offset (Fig. 3B)." While Fig 3B clearly illustrates that treated adolescents exhibited lower TPR throughout the stress task, I am not seeing evidence in this

figure of a quicker return to baseline TPR during the recovery period in the treatment group. Rather, both groups appear to show a similar TPR slope during the recovery period, and the group differences in levels seems to be due to the difference that was present at the beginning of the recovery period.

3) Effects on Physiological Responses: Did the authors verify that the treatment and control groups exhibited similar levels of TPR at baseline? It may be helpful to add baseline values to Fig 2A.

4) Heterogeneous effects, line 248: It is unclear what the authors mean by “Here and in the following studies we assessed mindsets...via self-reports at baseline...” To which study does “here” refer? Which are the “following” studies?

5) Heterogeneous effects, lines 271-278: References to Fig. 3D appear to actually be referring to the figure labeled 3C.

6) Methods, Line 774: The baseline measure of “negative prior mindsets” is described as a “three-item measure” but only two items are listed (measuring the fixed mindset of intelligence and the stress-is-debilitating mindset).

7) Methods, Study 1, Participants: To better understand the generalizability of the sample, it would be helpful to have information about the number of different public schools that were included in the study, the geographic regions within the U.S. where participating schools are located, and their urbanicity.

8) Methods, Study 2, Participants, line 886: Please correct a grammatical error/typo (“Participants were predominately first-year college students attended a selective...”)

9) Methods, Study 4: Daily cortisol samples were taken upon arrival at school, at lunchtime, and after school. Since eating causes a cortisol response, it would be helpful to know whether the timing of the morning and lunchtime cortisol samples were constrained to be immediately before or after a meal (school breakfast/lunch), or whether this was an uncontrolled source of variance.

10) Methods, Study 4: Were cortisol samples collected by participants themselves, or were they collected by researchers? How were the samples collected?

Referee #2 (Remarks to the Author):

This study examined whether a “synergistic mindsets” intervention can help adolescents improve their stress responses. Two mindsets in particular were examined: one related to a growth mindset and one related to the idea that physical stress responses can mobilize energy and be controlled. Across five randomized controlled trials, the authors found that the mindsets intervention resulted in better cognitive, internalizing, cardiovascular, and neuroendocrine responses to stress. These effects were particularly effective for individuals who held negative mindsets prior to the intervention.

Although interventions related to the growth mindset are well-established in the literature, the novel piece of this work was to combine the growth mindset with the mindset that the physical stress response (e.g., racing heart, sweaty palms) can help individuals meet the challenge in a beneficial way. To my knowledge, this is the first study to do this. That said, without a comparison group that uses just the growth mindset (and not the two mindsets combined), it is not clear to me that the “synergistic mindsets” intervention is substantially better than existing work on growth mindset interventions. That said, the set of studies provides a comprehensive test of the model presented in Figure 1.

The intervention seems focused on skill development or cultivating positive mindsets specifically in the intellectual domain (not the social domain), so I am struggling to understand how that would have been applicable in Study 5 when the stressor was presumably the COVID-19 pandemic, related closures, and social isolation. Although the Summary and Introduction try to integrate social challenges into the overall story (and there is some overlap between intellectual challenges and the social context), I found that piece less compelling given that the intervention seems targeted primarily to academic challenges.

In general, the figures are nicely done. I also found the description for Figure 1 useful but it could be bolstered by supporting references.

The authors should provide evidence for compliance with and adherence to the intervention. Manipulation checks for all studies are given, but I can't help but wonder how many participants finished each study, were adherent to the intervention, etc.

In Study 3 (and other relevant studies), it is not clear to me how many participants had negative versus positive prior mindsets. Do these tend to be evenly distributed throughout the population or, on average, are adolescents more likely to have one type of mindset versus the other?

In Study 4, randomization to condition occurred at the individual level among students who attended a private school. Did the researchers do anything to ensure that students did not talk with others about what their intervention entailed?

In Study 5, the authors indicate that generalized anxiety symptoms were reduced for adolescents who had negative mindsets at the start of the study. Does this mean generalized anxiety was measured at baseline and the reductions reflect a change from baseline to end of study? Or were generalized anxiety symptoms just lower for those adolescents at the end of the study compared to those with initially positive mindsets? If the latter, I would replace "reductions" with something like

"lower levels."

Please provide more information about the measure of prior mindsets. How were the three items for each mindset combined? What is the internal consistency reliability for each mindset?

It would be helpful to see the intervention materials used for the control condition.

The Discussion section was remarkably short and seems to be missing some key pieces. I would have liked to see a comparison of the current findings to previous findings, mention of study limitations, generalizability of findings, and implications for application to other populations. Please also expand the paragraph about the "supports" or "affordances" that are mentioned in the discussion. It is unclear to me what those might be and examples would be helpful.

Referee #3 (Remarks to the Author):

Summary of the key results

This paper reports an interesting set of studies that demonstrate improvements in adolescent stress responses following a 30-minute online 'synergistic mindsets' intervention based on growth mindset and stress-can-be-enhancing mindsets. Five studies involving a large number of adolescents (school students and undergraduates) demonstrated improvements in stress-related cognitions (Studies 1-2), cardiovascular reactivity (Study 3), daily internalizing symptoms and cortisol levels (Study 4), and generalized anxiety symptoms during the 2020 COVID-19 lockdowns (Study 5).

Originality and significance

The results are surprisingly impressive for such a short online intervention, given that virtually no previous universal intervention has had a reliable and robust impact on mental health or stress in adolescents, as is well described in a previous paper written by the first author together with Dahl and Dweck (2018). In this respect, the study and findings are very novel and could have a large impact, especially in educational settings.

Clarity and context: lucidity of abstract/summary, appropriateness of abstract, introduction and conclusions

The summary is clear but the following sentence is not accurate as these are not strictly trials as three out of five were not registered, and the total N is misleading as sample size (as well as demographics and age) varied substantively between groups across studies: 'In five double-blind, randomized, controlled trials (total N = 4,091 adolescents)'. The introduction is quite long and sometimes repetitive. The theory in figure 1 is complex and broad. Can the authors explain how it is falsifiable? The paper is framed as focusing on social stress, but most of the stresses tested were not social in nature.

Appropriate use of statistics and treatment of uncertainties

Only two of the five studies (studies 1 and 2) were preregistered. It appears from the Methods that studies 3-5 were not preregistered and this is a puzzle and a concern as trials must be registered. It

would be preferable to see replication of these three studies in independent samples of adolescents in a new set of preregistered trials. At the very least the language in the paper needs to change, and the word 'trial' (in the summary, for example) removed and trial terminology such as blinding, randomization and control arms avoided. Justification of the lack of preregistration should be explicitly included. This is currently not included in the paragraph in the Methods, except for Study 5 (and this is not a very convincing justification as many other studies during the pandemic were preregistered).

It would be helpful if the authors could include a hypothesis-testing framework and frequentist statistics (e.g. from Extended Data Table 2), as well as Bayes factors, throughout to support their analyses. It appears that the BCF analysis was developed recently by one of the co-authors, and it is not easy to follow if you have never used it before. Expertise in machine learning and Bayesian modeling is required to understand it. The final model needs more detail – what covariates and moderators were included?

I would like to see raw data for each study, not just the lines from models and confidence intervals, which hides a lot, most notably individual data, which needs to be shown. Currently the paper does not tell us anything about individual level predictions but we would expect large individual differences for this kind of intervention. Who did it work for and why? Did anyone get worse post-intervention? Were there individual differences associated with, for example, gender, SES, personality, IQ? Some of the plots look very similar for the two groups (intervention and control e.g. Figure 4) and this made me wonder what exactly we are looking at – are they similar because they are modelled to be this way?

Data & methodology: validity of approach, quality of data, quality of presentation

Studies 1 and 2. Not a criticism, just a comment: the results of the first two studies are not particularly surprising as they are essentially testing the understanding and/or memory of the statements delivered in the intervention.

Study 3. How does this intervention affect cardiovascular and physiological output? What is the mechanism? Further interrogation of the data would be useful here as I don't think looking at prior mindsets – although interesting - tells us much about mechanism per se. Treated negative mindset individuals' vascular resistance became indistinguishable from controls with positive prior mindsets – but how? What is the mechanism here?

Although the intervention was associated with reductions in peripheral resistance, how did participants do on the stress tests (the speeches and the math)? Does the intervention improve performance, or make it worse, or have no effect? Is the reduction in peripheral responses the mediator of these effects? This would be interesting to look at, if possible.

Study 4. We need to see the raw data. The lines look surprisingly similar between the two groups, and I have never seen such clean data with saliva cortisol – presumably this is because this is not data, but models?

Treated negative mindset individuals improved in terms of internalizing symptoms, but is this

because they had further to move? As with study 3, what are the mechanisms by which the intervention affects cortisol, and does this improve performance on the stress test, and is it the mediator? This could all be analyzed and seems important to know.

Intervention. Some of the neuroscience in the intervention is questionable. For example, what is the evidence for the claim: 'Doing challenging things strengthens your brain's connections'? And for the claim: 'When people don't know this information about how the brain works, they tend to avoid frustration and difficulty by not doing things that feel hard.' Stress is not always good for the brain – in fact most animal studies on stress show that stress is damaging to the brain, in the short and longer term. This is a concern for an intervention that might gain traction in schools.

Why were different age groups (school students or undergraduates) included in each study? Was there a theoretical or scientific justification for this? They are quite different populations, biologically and in terms of social context and levels of independence in their lives. And why the very different sample sizes across studies, from 160 to 2,717? This is not well justified in the methods: in some studies, N was determined by power calculations, in others by pragmatics. There were very different demographic groups in each study, with some studies involving social science undergraduate classes (presumably largely middle class and not hugely ethnically diverse?) whereas one study included adolescents from economically-disadvantaged families who were nearly all (95%) from black or indigenous racial/ethnic groups. Why were such different groups included across studies?

More information about covariates is needed. Was IQ measured? An IQ difference between the intervention and control group might explain the differences in some outcomes.

Referee #4 (Remarks to the Author):

A. The authors present evidence from five double-blind RCTs of different populations of adolescents that a brief two-part self-administered online intervention designed to encourage the development of mindsets that (1) intelligence is malleable and (2) exposure to stress can be enhancing under certain circumstances improves stress-related cognitions, cardiovascular reactivity, and psychological well-being.

B. The research extends prior studies of mindset interventions (some by the lead author and his colleagues) by examining a combined intervention to change beliefs about both intelligence and stress. The research is novel and timely.

C. I have three concerns about the paper. First, and most importantly, the authors emphasize the synergistic nature of the intervention, which I take to mean that the two interventions interact to produce an outcome that is greater than either alone. But to test this hypothesis, the design would need to have four experimental groups (not simply one treatment group and a no-treatment control). In addition to the combined treatment and control groups, a proper test of the synergy hypothesis requires a group that receives the intelligence intervention but not the stress intervention, and a group that receives the latter but not the former. It is all well and good to point out that prior evaluations of stress coping interventions have not been greatly effective. But we don't know from the research presented here whether the specific stress intervention used in this study would be just

as effective on its own as in combination with the intelligence intervention. The argument that people who had the most dysfunctional mindsets of both types before the intervention benefited the most from the intervention isn't quite enough to persuade me that there is true synergy here. My second concern is with the description of the treatment and control groups. I realize that details are included in the supplement, but I think that additional description of the intervention itself in the body of the text would be helpful. Along these lines, no description of the control group's experience is provided. What, exactly, were these participants asked to do? Finally, I found the figures impenetrable, and more confusing than clarifying.

D. No concerns about the statistics or treatment of uncertainty.

E. The absence of appropriate comparison groups in which only one part of the two-part intervention is administered limits the validity of the conclusions.

F. The authors should provide evidence that the synergistic intervention is more effective than either component alone. Alternatively, they should temper their synergistic argument and simply describe the intervention as having two components.

G. No concerns about the references.

H. The manuscript is generally clearly written.

Referee #5 (Remarks to the Author):

This paper investigates the effect of a synergistic mindset intervention on reducing adolescents social stress. The authors studied five experiments with different stressors and concluded that the synergistic mindset intervention successfully reduced stress responses in all studies.

The proposed intervention is innovative and can potentially help with adolescent stress management. The main statistical model used in the paper is (targeted smooth) BCF, which is a flexible semiparametric model for evaluating causal effects. However, I have doubts about the appropriateness of this model for your studies. In particular,

1. For most of the studies, the outcomes y_{ij} 's take discrete values (from 1 to 3, 5 or 7). However, the BCF model assumes a normal distribution. Why is the model appropriate for analyzing your data?

2. How many covariates x_{ij} are included in the data analysis? For example, are the covariates gender and race included in the model? The BCF model is designed to incorporate a large number of covariates. If there are only one or two covariates, the BCF model doesn't have much benefit over linear regressions.

3. For each study, are the individuals "similar" in mindset treatment and control groups? That is, are their propensity score distributions balanced?

4. The original BCF or targeted smooth BCF model includes the propensity score in the $\beta()$ function to mitigate treatment effect bias. However, the propensity score seems to be missed in your model. Why did you leave out the propensity score?

In summary, I'd like to see justifications of the appropriateness of the BCF model from the above perspectives.

Referee #1 (Remarks to the Author):

This manuscript reports on a series of randomized-controlled evaluations of a novel, single-session, easily scalable intervention to promote adolescents' positive mindsets about stressors (growth mindset and "challenge" vs. "threat" appraisals) and stress responses (stress responses can be helpful), which the authors term a synergistic mindsets intervention. Five sequential studies revealed intervention effects consistent with each step in the authors' theory of change: (1) positive changes in growth mindset and helpful stress appraisals post-intervention (immediately and at a 3-week delay), (2) reductions in total peripheral resistance (TPR; a cardiovascular index of threat reactivity) during a well-validated laboratory-based stress task, (3) reduced daily internalizing symptoms (negative self-regard) roughly 2 weeks post-intervention, particularly on days with high-intensity stressors, (4) reduced cortisol levels throughout the school day roughly 2 weeks post-intervention, and (5) lower generalized anxiety symptoms during the early stages of the COVID-19 lockdown (3 months post-intervention) among individuals with negative mindsets at baseline. The authors additionally found that treatment effects on cardiac stress reactivity and negative self-regard on high-stress days were greater for those with negative mindsets at baseline, as would be expected based on the mechanistic theory of change.

This work is novel in that the intervention combines two features of interventions that have not previously been combined: growth mindset and stress reappraisal. The work is methodologically strong and compelling, including the use of intent-to-treat analyses as a conservative approach for estimating intervention effects and sophisticated statistical modeling using Bayesian Causal Forest regressions. Sample sizes are justified by a priori power analyses where possible and two of the studies were pre-registered for scientific transparency. The authors' conclusions are well supported by the data. The robustness of the conclusions is supported by focusing on effect sizes and the posterior distribution of treatment effects as opposed to focusing on statistical significance as determined by an arbitrary p-value cutoff. While many effect sizes are relatively small, the findings are impressive in their consistency across studies, providing evidence of their reliability. Moreover, such a minimal intervention would be expected to produce very small effect sizes on distal outcomes such as daily cortisol levels or generalized anxiety symptoms. Given the ease with which this intervention could be implemented at a large scale, the findings have important public health implications.

The authors admit that additional research will be needed to further delineate for whom the intervention is effective, and the conditions under which its impacts on distal outcomes are maintained over time.

Relevant prior work is appropriately cited, the paper is extremely well written, and figures are well utilized.

I recommend this article for publication following minor revisions to address several concerns and some minor comments, listed below.

Concerns:

1) I am uncertain why the title and abstract focus on "social stress," whereas the intervention and findings are not limited to adolescents' responses to social stressors.

We see that it would be better to clarify that "social stress" is shorthand for social-evaluative stress, which refers to experiences in which the self could be negatively judged by others. This includes the classroom

and peer experiences. Now we say “social-evaluative” in the title and paper. The first sentence of the abstract now says this:

Social-evaluative stressors—experiences when the self could be negatively judged by others—pose a major threat to adolescent health via their effects on internalizing disorders, such as anxiety and depression.

2) I did not see information about the specific covariates or potential moderators that were entered into each of the BCF models.

We added a covariate disclosure table to the SI and we uploaded the BCF syntax for each study. (See above). The content from the SI is copied below:

Here we list the covariates that were included in each study. In every case, if a given covariate was not included in the model, it was not measured in that study. Unless otherwise specified, each covariate was chosen because (a) the variable could be related to the outcome, and (b) the variable could show chance differences at baseline, which could lead to spurious results if the model did not adjust for those chance differences. Our philosophy was to include as many covariates as possible when they were measured, and then allow BCF to decide to use them.

Baseline covariate	Justification	Study					
		1	2	3	4	5	6
Fixed mindset*	Baseline value of the variable targeted by the intervention, suspected to influence the outcomes across studies						
Stress mindset*	Baseline value of the variable targeted by the intervention, suspected to influence the outcomes across studies		+				
Perceived social stress*	Established baseline measure of internalizing symptoms;						
Sex*	Girls/women tend to show higher internalizing symptoms						
Age	Students at different grade levels could experience different academic loads that contribute to their social-evaluative stress						
Race/Ethnicity	Students from different racial or ethnic groups could experience different kinds or intensities of social-evaluative stressors						
Self-esteem	Established measure of psychological well-being; expected to be related to outcomes	#					
Test anxiety	Highly test-anxious adolescents could be lower-performing and could be more likely to show negative stress reactivity						
Social class	Lower-SES adolescents could be lower-performing and face environmental stressors that relate to internalizing outcomes and prior mindsets.						
Personality (BFI)	Requested by reviewer						
Time of day / Day of the week	There is hourly and weekly variation in cortisol levels						

Table S5: Covariates Included in BCF Models Across Studies. Note: Grayed-out boxes indicate inclusion of a given covariate. * The “core” moderators are indicated with an *, and they were included in the model whenever

they were measured. All results were identical when only including the “core” variables as covariates and excluding all additional covariates. + Only in Study 2, there was an error in the programming of the baseline stress mindset measure, and so data were incomplete for this measure and not usable for the correlational analyses presented in Extended Data Table 1. # Only for Study 1, a single-item measure of self-efficacy was the pre-registered covariate, rather than the global self-esteem measure, because the scenario was very specifically tied to an academic setting (i.e. doing poorly on a stressful assignment) and we sought to control for any potential chance differences in academic confidence across conditions.

Here is what we now say in the methods section of the paper:

The covariates included in each study are listed in Table S5 in the SI. The core covariates and moderators were: the prior mindset measures (fixed mindset and stress-is-debilitating mindsets), sex, and perceived social stress, as pre-registered (osf.io/tgysd). When available, other covariates were added as well: age, race/ethnicity, self-esteem, test anxiety, social class, and personality. Justifications for each covariate appear in Table S5 in the SI.

We note that the BCF analysis method addresses concerns about covariate inclusion, because the algorithm decides whether a given covariate is relevant, and how to model it.

3) Study 4: “Internalizing symptoms” were operationalized as participants’ responses to one question on each daily survey: “Overall, how good or bad did you feel about yourself today?” Although this is an internalizing symptom (low self-regard), I feel it is slightly misleading to describe this measure as “internalizing symptoms” which implies a more comprehensive measure of various anxious and depressive symptoms. In Fig 4, this measure is referred to as “negative self-regard” and I feel this is a more accurate descriptor of the construct being measured and should be used throughout the manuscript.

We have made this change. In addition, we now cite a paper that demonstrates negative self-regard is a central symptom in the network of depression symptoms (see Mullarkey et al., 2020) putting it squarely in the category of “internalizing symptoms.”

Here is what we now say in Study 5:

Study 5 assessed the effects of the synergistic mindsets intervention on psychological and biological indicators of stress responses that have a longer time-course: *negative self-regard*, an internalizing symptom, and *cortisol*, a hormonal indicator of threat-type stress responses.

Minor comments:

1) Effects on Cognitive Appraisals, line 200: The word “and” appears to be missing in the following parenthetical phrase: “which occurred 1-3 days post-intervention was not mentioned in the intervention content.”

We fixed this.

2) Effects on Physiological Responses, line 240: The authors state that “treated adolescents recovered to baseline more quickly after stress offset (Fig. 3B).” While Fig 3B clearly illustrates that treated adolescents exhibited lower TPR throughout the stress task, I am not seeing evidence in this figure of a quicker return to baseline TPR during the recovery period in the treatment group.

Rather, both groups appear to show a similar TPR slope during the recovery period, and the group differences in levels seems to be due to the difference that was present at the beginning of the recovery period.

We fixed this. In the results to Study 3, we say:

In the treatment condition TPR increases were blunted, and especially during the speech (Fig. 3B). At every epoch of the TSST, and especially during the most-stressful speech epoch, the expected TPR for the mindset treatment group was lower than the control group and the conditional average treatment effect (CATE) was less than zero (see Fig. 3C).

3) Effects on Physiological Responses: Did the authors verify that the treatment and control groups exhibited similar levels of TPR at baseline? It may be helpful to add baseline values to Fig 2A.

All analyses were conducted with “reactivity” scores, which subtracts off the baseline levels, and so even if there had been meaningful differences, the analysis would have adjusted for them.

To answer R1’s question, in Study 3 the group difference at baseline was 161, $t(157) = -.77, p = .44$, which is what is expected due to our study’s use of person-level random assignment, with very little attrition. In the note to Fig. 3, we now state that the groups exhibited similar levels of TPR at baseline. We say:

The differences in TPR for the two groups were similar at baseline (see propensity score comparisons in the SI).

4) Heterogeneous effects, line 248: It is unclear what the authors mean by “Here and in the following studies we assessed mindsets...via self-reports at baseline...” To which study does “here” refer? Which are the “following” studies?

We fixed this by removing the confusing terms.

5) Heterogeneous effects, lines 271-278: References to Fig. 3D appear to actually be referring to the figure labeled 3C.

We fixed this.

6) Methods, Line 774: The baseline measure of “negative prior mindsets” is described as a “three-item measure” but only two items are listed (measuring the fixed mindset of intelligence and the stress-is-debilitating mindset).

The measures for all studies are now posted to the study registration site, and there are direct links to the measures in the disclosure table above.

7) Methods, Study 1, Participants: To better understand the generalizability of the sample, it would be helpful to have information about the number of different public schools that were included in the study, the geographic regions within the U.S. where participating schools are located, and their urbanicity.

We contacted our data collection partner, the Character Lab Research Network (CLRN), and they provided as much detail as permitted without disclosing the identities of the schools. This information now appears in the paper and is copied below:

Participants were from a large, heterogeneous sample of adolescents who were evenly distributed across grades 8 to 12 in 35 U.S. public schools (13 y/o: 16%; 14: 20%; 15: 20%; 16: 21%; 17: 18%; 18: 5%). The schools were sampled from a stratum of large, diverse, suburban and urban public schools in the southeast United States. Forty-nine percent of adolescents identified as male, 49% as female, and 2% as gender non-binary. Participants were also racially and ethnically diverse (participants could indicate multiple racial/ethnic identities so numbers exceed 100%): Black: 20%; Latinx: 39%; White: 68%; Asian: 7%. Participants were also socioeconomically diverse: 40% received free or reduced-price lunch, an indicator of low family income.

8) Methods, Study 2, Participants, line 886: Please correct a grammatical error/typo (“Participants were predominately first-year college students attended a selective...”)

We fixed this.

9) Methods, Study 4: Daily cortisol samples were taken upon arrival at school, at lunchtime, and after school. Since eating causes a cortisol response, it would be helpful to know whether the timing of the morning and lunchtime cortisol samples were constrained to be immediately before or after a meal (school breakfast/lunch), or whether this was an uncontrolled source of variance.

Saliva samples were collected during the lunch period but before students ate, and this is now stated in the paper’s methods. We say:

Lunchtime samples were collected before students ate.

10) Methods, Study 4: Were cortisol samples collected by participants themselves, or were they collected by researchers? How were the samples collected?

Researchers supervised all data collection, which is now stated in the paper. The text from the methods is copied below.

All data collection was supervised by trained research staff who assisted participants and answered any questions, while being blind to condition assignment and specific hypotheses.

Referee #2 (Remarks to the Author):

This study examined whether a “synergistic mindsets” intervention can help adolescents improve their stress responses. Two mindsets in particular were examined: one related to a growth mindset and one related to the idea that physical stress responses can mobilize energy and be controlled. Across five randomized controlled trials, the authors found that the mindsets intervention resulted in better cognitive, internalizing, cardiovascular, and neuroendocrine responses to stress. These effects were particularly effective for individuals who held negative mindsets prior to the intervention.

Although interventions related to the growth mindset are well-established in the literature, the

novel piece of this work was to combine the growth mindset with the mindset that the physical stress response (e.g., racing heart, sweaty palms) can help individuals meet the challenge in a beneficial way. To my knowledge, this is the first study to do this. That said, without a comparison group that uses just the growth mindset (and not the two mindsets combined), it is not clear to me that the “synergistic mindsets” intervention is substantially better than existing work on growth mindset interventions.

We are grateful for the reviewer’s suggestion because this substantially improved the paper. We conducted the requested pre-registered study (Study 4). Synergistic mindsets were more effective than either the stress mindset or growth mindset alone, as noted above

Importantly, manipulation check analyses showed that the two single-mindset conditions changed their respective mindset manipulation checks, suggesting that our study faithfully manipulated the two solitary mindsets. In sum, this is a powerful demonstration of how the two mindsets need to go together.

We note that the null findings for the single mindset treatments in Study 4 are not at odds with the literature. Previous studies found effects of growth mindset and stress mindset alone, but for different outcomes, such as grades in the case of growth mindset. No study has shown, for example, that either mindset on their own reduced threat-type cardiovascular responses to the TSST or overall cortisol levels.

That said, the set of studies provides a comprehensive test of the model presented in Figure 1.

The intervention seems focused on skill development or cultivating positive mindsets specifically in the intellectual domain (not the social domain), so I am struggling to understand how that would have been applicable in Study 5 when the stressor was presumably the COVID-19 pandemic, related closures, and social isolation. Although the Summary and Introduction try to integrate social challenges into the overall story (and there is some overlap between intellectual challenges and the social context), I found that piece less compelling given that the intervention seems targeted primarily to academic challenges.

R2 is correct that the intervention says a lot about skill development situations, but the main purpose of the intervention is to help people deal with the feeling of being judged and evaluated by others for not living up to standards for performance (i.e. social-evaluative stressors). This applies in the classroom and to peer comparisons.

We now state in the paper that the aim of the intervention is not to help students cope with the existential uncertainty of a global pandemic, per se, but rather to help them to address normative social-evaluative stressors that persisted even during the pandemic’s school closures, and that could have been elevated as more students were comparing themselves to each other on social media while in isolation. Here is what we say in the paper:

In Study 6, the environmental stressor was continued academic pressure and social isolation during the early stages of the COVID-19 pandemic in the U.S in the Spring of 2020, as students were forced to exit University housing (see study procedure in Fig. 6). ...

Of note, the intervention need not discuss the profound uncertainty and isolation that young people experienced during COVID-19 lockdowns to help improve overall symptoms. All it would need to do is help adolescents reappraise the normative social-evaluative demands that contributed to mental health problems and persisted during the pandemic.

In general, the figures are nicely done. I also found the description for Figure 1 useful but it could be bolstered by supporting references.

We added references to Fig. 1.

The authors should provide evidence for compliance with and adherence to the intervention. Manipulation checks for all studies are given, but I can't help but wonder how many participants finished each study, were adherent to the intervention, etc.

We have now added this information to the methods. We say:

We defined adherence as completion of the last page of the intervention. In the studies where participants were closely supervised by researchers (Studies 3, 4, and 5), adherence was high (97% to 99%). In the studies where the intervention was self-administered with no supervision, adherence was lower but still acceptable: 85%, 88% and 82% for Studies 1, 2, and 6, respectively. Because we conducted intent-to-treat analyses, participants were retained in the analytic sample regardless of intervention completion status.

In Study 3 (and other relevant studies), it is not clear to me how many participants had negative versus positive prior mindsets. Do these tend to be evenly distributed throughout the population or, on average, are adolescents more likely to have one type of mindset versus the other?

The figures that show the additive summaries of the mindset moderation have a “rug,” which shows how much data there are at each part of the distribution (see, e.g., Fig. 5 and ED Figs 1 and 2). This helps the readers to see the distribution of the mindset variable.

We have also now added the baseline means and standard deviations for each of the studies so that readers can assess this information (see below).

We note, however, that these are continuous variables, so the designations of “having” one mindset or another is relative within each study. To avoid making arbitrary decisions about mindset cut-points, we developed a hands-off algorithm, as we explained previously in the SI (see Fig. S3).

Descriptive Statistics: Baseline Mindset Values

In each study, a composite of baseline mindset values was constructed by taking the unweighted average of all of the items administered in that study, separately for each construct. The means and standard deviations for the baseline fixed and stress mindsets are depicted in Table S6 below. Note that the two high school studies (1 and 5) showed higher fixed mindset values at baseline. Studies 2 and 6 were conducted at the same university, and Studies 3 and 4 were conducted at the same university. Studies 3 and 4, which were laboratory studies, allowed more time for measurement and therefore used a longer (8-item) measure of stress mindsets.

	Study 1	Study 2	Study 3	Study 4	Study 5	Study 6
Fixed mindset						
M =	3.15	2.52	2.93	2.71	2.99	2.7
SD =	1.12	1.25	1.2	1.1	1.06	1.1
Stress-is-debilitating mindset						
M =	4.27	-	2.9	3.56	3.47	4.1
SD =	1.2	-	0.85	0.84	1.21	1.12

Table S6. Comparisons of baseline values of fixed mindsets and stress mindsets across the six studies. Only in Study 2, there was an error in the programming of the baseline stress mindset measure, and so data were incomplete for this measure and not comparable to the other samples.

In Study 4, randomization to condition occurred at the individual level among students who attended a private school. Did the researchers do anything to ensure that students did not talk with others about what their intervention entailed?

Yes we did. In Study 5 we used the standard procedures for student-level random assignment experiments that have been effective in previous mindset interventions conducted in nationally-representative samples (e.g. Yeager et al., 2019, *Nature*). This includes having treatment and control materials that are closely aligned (e.g. using the same formatting, font, and graphic art), having students complete the materials privately (where comparisons are not possible), and discouraging students from discussing the study with one another. Study 5 also had students self-administer the materials in semi-private rooms, overseen by a researcher who simply assisted with the computer/browser but did not discuss the treatment content in any way, which added another layer of experimental control. We cannot guarantee that treated students did not share the treatment content with their peers after leaving the lab. But this kind of contamination would lead to more conservative effect sizes because it would involve treating the control group, and therefore it is not an alternative explanation for our findings.

In Study 5, the authors indicate that generalized anxiety symptoms were reduced for adolescents who had negative mindsets at the start of the study. Does this mean generalized anxiety was measured at baseline and the reductions reflect a change from baseline to end of study? Or were generalized anxiety symptoms just lower for those adolescents at the end of the study compared to those with initially positive mindsets? If the latter, I would replace "reductions" with something like "lower levels."

Thank you for pointing this out. There was no baseline measure of anxiety in Study 6. We meant to say that there is a reduction relative to the counterfactual value at that time (i.e. relative to the neutral control condition). That is, we were using "reduce" in the Rubin-counterfactual sense, not in the longitudinal sense. In Study 6 we say:

Three months after the online intervention, we observed reductions in generalized anxiety symptoms relative to controls, CATE = - .17 *SD* [-.37, .00], among adolescents who reported negative mindsets prior to random assignment (see Fig. 6B)

Please provide more information about the measure of prior mindsets. How were the three items for each mindset combined? What is the internal consistency reliability for each mindset?

The items were combined by taking the unweighted average of the items within each construct. The Cronbach's alphas ranged from .70 to .85. Of note, because of the pre-registrations and standard operating procedures, we would have been required to use the scales even if the alphas had been lower.

It would be helpful to see the intervention materials used for the control condition.

They are now posted here: <https://osf.io/4y6gx/>. We note that these are the same control materials that were used in Yeager et al. (2019), but with an addition of content related to the body's stress system, to control for the possibility that any reflection on stress could help students cope better with it.

The Discussion section was remarkably short and seems to be missing some key pieces. I would have liked to see a comparison of the current findings to previous findings, mention of study limitations, generalizability of findings, and implications for application to other populations. The format for this journal calls for a short discussion section, and so we responded to each of these while doing our best to adhere to space restrictions. Here is how we addressed each one:

- **Comparison to previous findings:** We compare our findings to the dominant appraisal-based approach when we comment in several places on how the present study's results "transfer" across stressful situations.

This intervention overcomes the primary limitation of popular "reappraisal" approach to promoting well-being. ... appraisal-focused approaches suffer from the *transfer problem*: people usually fail to show effects of a reappraisal treatment on untrained stimuli—which are virtually all of the stimuli the individual later encounters^{36–39}. ... The synergistic mindsets intervention solves the transfer problem by targeting cognitive processes that operate at a more general level than situation-specific appraisals: mindsets. Mindsets affect how people think about categories of situations (e.g., academic failures or negative emotions in general)^{15,40,41}, and therefore help people deductively make appraisals about the meanings of situations they have not been trained on. This allows the benefits of mindset interventions to transfer across situations, even novel ones (such as how to deal with ongoing academic and social stressors during a global pandemic).

- **Study limitations:** The paragraph on affordances and customizing the treatment was meant to be a discussion of the key limitations; we have now stated it more explicitly.

A limitation of this set of studies is that the examples in the intervention were tailored for social-evaluative threats in academic contexts. Careful attention to heterogeneity of effects will be required to prepare the synergistic mindsets intervention for wider implementation.⁷⁴ First, the materials would need to be adapted for other stressors/stressful contexts in such as the workplace, romantic relationships, or athletics⁷⁵. Second, mindset interventions in general depend on the supports, or *affordances*, in a context to sustain the self-reinforcing cycles that propagate their effects over time (Fig.1)^{27,76,77}. To illustrate, in past experiments one-time mindset interventions only changed academic outcomes when teachers used language or policies that made the core mindset message locally valid and useful (e.g. when the teachers themselves held a growth mindset and framed mistakes in class as positive learning opportunities rather than signs of inability).⁷⁸ Currently, we do not know which affordances are most critical in academic or other contexts, nor how to make them more abundant.

- **Generalizability and implications for applications to other populations:** We now call for tests of generalizability and heterogeneity at the end of the first paragraph, and we cite our recent paper

on this topic ² (see below). We also note the applications to other settings in the paragraph quoted above.

Because mindset interventions similar to the one tested here can be delivered cost-effectively in national or regional scale-up studies ^{27,72}, the present research represents a critical theoretical step between basic insights about affect regulation and towards the discovery of actionable intervention methods that might be able to produce real, lasting change at scale. An important next step for social scientists will be to assess the generalizability and heterogeneity in these effects with new large-scale trials ⁷¹.

We should note that we have many additional thoughts on these topics, and we are glad that the reviewer raised them. We would be happy to discuss these in greater detail and be even more responsive to R2's comment if the Editor thinks we have the space for it.

Please also expand the paragraph about the “supports” or “affordances” that are mentioned in the discussion. It is unclear to me what those might be and examples would be helpful.

We have added an example of this to the discussion in the paragraph copied above.

Referee #3 (Remarks to the Author):

Summary of the key results

This paper reports an interesting set of studies that demonstrate improvements in adolescent stress responses following a 30-minute online ‘synergistic mindsets’ intervention based on growth mindset and stress-can-be-enhancing mindsets. Five studies involving a large number of adolescents (school students and undergraduates) demonstrated improvements in stress-related cognitions (Studies 1-2), cardiovascular reactivity (Study 3), daily internalizing symptoms and cortisol levels (Study 4), and generalized anxiety symptoms during the 2020 COVID-19 lockdowns (Study 5).

Originality and significance

The results are surprisingly impressive for such a short online intervention, given that virtually no previous universal intervention has had a reliable and robust impact on mental health or stress in adolescents, as is well described in a previous paper written by the first author together with Dahl and Dweck (2018). In this respect, the study and findings are very novel and could have a large impact, especially in educational settings.

Clarity and context: lucidity of abstract/summary, appropriateness of abstract, introduction and conclusions

The summary is clear but the following sentence is not accurate as these are not strictly trials as three out of five were not registered, and the total N is misleading as sample size (as well as demographics and age) varied substantively between groups across studies: ‘In five double-blind, randomized, controlled trials (total N = 4,091 adolescents)’.

We are grateful for this comment. The revised abstract is copied below. We refrained from going into detail about distinguishing between studies that were registered versus pre-registered because there are many dimensions of rigor on which the studies varied and it felt too complicated to go into how non-pre-registered studies restrained degrees of freedom by following standard operating procedures.

Social-evaluative stressors—experiences when the self could be negatively judged by others—pose a major threat to adolescent health via their effects on internalizing disorders, such as anxiety and depression. Preventative interventions to help adolescents improve their responses to social-evaluative stressors, however, have rarely been effective. Here we show that replicable benefits for adolescents’

stress responses can be achieved with a short (~30-minute), scalable *synergistic mindsets* intervention. This intervention, which is a self-administered online training module, synergistically targets both growth mindsets (the idea that people's intelligence can be developed in response to challenge) and stress-can-be-enhancing mindsets (the idea that people's stress responses can fuel optimal performance). In six double-blind, randomized, controlled experiments conducted with youth in grades 8-12 and in undergraduate education, the synergistic mindsets intervention improved stress-related cognitions—the immediate target of the intervention (Studies 1-2, $N = 3,472$)—and a cascade of stress-linked outcomes: cardiovascular reactivity (Studies 3, $N = 160$; Study 4, $N = 200$), cortisol levels (Study 5, $N = 118$ students, $n = 1,213$ observations), psychological well-being (Studies 4-5), course pass rates (Study 5), and anxiety symptoms during the 2020 COVID-19 lockdowns (Study 6, $N = 341$). Evidence that the two mindsets worked synergistically came from heterogeneity analyses showing stronger effects among participants holding both negative mindsets at baseline (Studies 3, 5, and 6) and from a four-cell experiment that showed stronger effects of the synergistic mindsets intervention relative to changing single mindsets alone (Study 4). Confidence in these conclusions is rooted in a conservative, Bayesian machine-learning statistical method for detecting heterogeneous effects.

The introduction is quite long and sometimes repetitive.

We worked to reduce the introduction, but after careful consideration, we believe that it is not possible to properly frame our studies' contributions in fewer than approximately 1,000 words, as there are many concepts to introduce (e.g., adolescent stress and mental health, two different mindsets, synergizing the mindsets, appraisal processes, challenge/threat responses, and brief interventions).

The theory in figure 1 is complex and broad.

We thought hard about this comment too, and considered other organizations. Ultimately, though, we feel that Figure 1 brings together several lines of work and theoretical models into a single model that allows for clear predictions across our complex set of studies. These include the process model of emotion regulation, the biopsychosocial model of challenge and threat, growth and stress mindset models, and recursive process models. It is important to represent each aspect of these models to help the reader understand the sequence of studies—for example, why the studies start with appraisals, then move to acute threat-type stress physiology, and then begin to look at recursive effects over time. Relatedly, Figure 1 is key to understanding the mechanisms for the effects. It explains, for instance, why we would expect TPR to be reduced in the synergistic mindsets condition. And without understanding those acute processes, it can seem surprising, magical even, why we see long-lasting effects on outcomes like anxiety or course-pass-rates. In fact, one major contribution of the paper is to integrate the different models into a single, coherent model. This contribution would be reduced if the figure was made less informative.

Can the authors explain how it is falsifiable?

Figure 1 presents a theoretical model that helps us derive our predictions, but it is not a theory in and of itself (like mindset theory, for instance). The model's predictions could be falsified, however, if they were not borne out in the data.

For example, Figure 1B implies that changing mindsets will be most effective among people who, counterfactually, would have been on a negative path to the upper-right quadrant of Figure 1B. Our heterogeneity analyses demonstrated this (Studies 3, 5, and 6). Further, it implies that changing just one of the two negative mindsets, while leaving the other intact, will be less effective than changing both. We now show this in Study 4.

The paper is framed as focusing on social stress, but most of the stresses tested were not social in nature.

See response to R1 above.

Appropriate use of statistics and treatment of uncertainties

Only two of the five studies (studies 1 and 2) were preregistered. It appears from the Methods that studies 3-5 were not preregistered and this is a puzzle and a concern as trials must be registered. It would be preferable to see replication of these three studies in independent samples of adolescents in a new set of preregistered trials.

We agree that all trials should be registered at a permanent and open repository, complying with NIH requirements. We have done so. Here is the updated text in the methods:

Study registration and efforts to curb researcher degrees of freedom. All studies are registered on the Open Science Framework (Study 1: <https://osf.io/tgysd>; Study 2: <https://osf.io/hb6vs>, Study 3: <https://osf.io/x4a63>; Study 4: <https://osf.io/fkgru>; Study 5: <https://osf.io/9pfha>; Study 6: <https://osf.io/mkqgf>). Detailed descriptions of open science disclosures, links to study materials, analysis plans, and deviations from analysis plans appear in the Supplemental Information. Studies 1, 2, and 4 were registered prior to analyzing the data. Studies 3, 5, and 6 were registered after analyzing the data. As explained in greater detail in the Supplementary Information, researcher degrees of freedom for Studies 3, 5, and 6 were constrained by following published and previously pre-registered standard operating procedures for TSST and daily diary studies⁵⁴, (the focus on TPR, SV, and PEP in Study 3 and the focus on the stressor intensity \times treatment interaction in Study 5), and by following the same analysis steps as the pre-registered studies (e.g., the same core covariates and moderators and conservative BCF modeling approach).

The issue, which we addressed above, is *preregistration* of analysis plans. This issue is less cut and dried.

Of course, we are aware of the benefits of pre-registration and have published some of the first-ever studies using that method for mindset research (e.g. Yeager et al., 2016, *Psych Science*; Yeager et al., 2019, *Nature*). Over the past several years, we have also learned that study registration has myriad purposes, many of which are independent from freezing an analysis plan prior to data collection (see³). We direct the reviewer to a discussion of these issues in the SI or our reply to the Editor above.

Nevertheless, there are cases of scientific discoveries being hampered by pre-registered analysis plans that presumed more knowledge than the researchers had at that time. For example, studies have pre-registered a primary outcome as a key test of a hypothesis but then, after collecting the data, learned that their method was not appropriate for the sample and could not provide a test of it (we discuss a few examples in the introduction to this paper: ⁴).

In general, there are many ways for a study to be rigorous³. Pre-registration of the analysis plan is only one, and so we shy away from “all or nothing” thinking about studies’ registration status.

Instead, when investigating a new phenomenon like in the present paper, our approach was to (a) be transparent about how the study’s design and methods adhered to established principles in the literature; (b) make the data and syntax available; (c) use a conservative Bayesian analysis method that penalizes the model results for the uncertainty in the analysis plan, and (d) conduct direct replications when feasible. We did this, and showed that our findings did, in fact, hold up to a direct replication.

At the very least the language in the paper needs to change, and the word ‘trial’ (in the summary, for example) removed and trial terminology such as blinding, randomization and control arms avoided.

We take this comment to mean that the word “clinical trial” implies a host of study design decisions to certain audiences and that we want to avoid confusing the issues. To respond to this, we make sure to call the studies “experiments” (or studies) and we do not refer to them as clinical trials, to avoid the jargonistic baggage. However, we note that the studies were, in fact, blinded and randomized and used control arms. Those design features are independent from the question of whether we froze the analysis plan on a public site prior to data collection.

Justification of the lack of preregistration should be explicitly included. This is currently not included in the paragraph in the Methods, except for Study 5 (and this is not a very convincing justification as many other studies during the pandemic were preregistered).

This now appears in the SI and in the study registrations for each of the three studies. **The text is copied to the top of this document.** The links are provided in the open science disclosure table in the SI and at the top of this document.

It would be helpful if the authors could include a hypothesis-testing framework and frequentist statistics (e.g. from Extended Data Table 2), as well as Bayes factors, throughout to support their analyses.

Our previous submission included the frequentist / linear model analyses in Extended Data Table 2. We have updated those analyses in light of the reviewers’ comments and the inclusion of a new study and we summarize them in the paper (**text copied above**).

Even so, we have decided not to implement the reviewer’s request to include null hypothesis testing throughout the main paper (either p-value cutoffs or Bayes Factors, which have the same problems). Our emphasis on estimation and uncertainty, rather than “accepting” vs. “rejecting” a null hypothesis, is in line with mainstream statistical reasoning. Much has been written about this, but this statement from Gelman nicely summarizes the view of the field:

The problem is not with p-values but with null-hypothesis significance testing, that parody of falsificationism in which straw-man null hypothesis A is rejected and this is taken as evidence in favor of preferred alternative B (see Gelman, 2014). Whenever this sort of reasoning is being done, the problems discussed above will arise. Confidence intervals, credible intervals, Bayes factors, cross-validation: you name the method, it can and will be twisted, even if inadvertently, to create the appearance of strong evidence where none exists.
(<https://statmodeling.stat.columbia.edu/2016/03/07/29212/>)

Ultimately, conventional null hypothesis testing suggests that our results are incredibly strong; in most cases, they meet even the high bar of $p < .001$. Nevertheless, we are not comfortable over-stating our results because there is still a great deal more to learn about heterogeneity, generalizability, and durability of these effects. The present studies establish the foundation of this phenomenon, but they are the beginning of the scientific process, not a definitive end. (See, e.g., our paper in *NHB*: <https://www.nature.com/articles/s41562-021-01143-3>).

In summary, we would prefer to continue presenting the results in a Bayesian framework because it is more up-front about the uncertainty of our results, and therefore, more likely to lead to more qualified applications of our findings in policy and practice.

It appears that the BCF analysis was developed recently by one of the co-authors, and it is not easy to follow if you have never used it before. Expertise in machine learning and Bayesian modeling is required to understand it.

We are grateful for this comment because it led us to revise the paper and take steps to make our syntax more accessible. We have now posted our syntax and data, which will hopefully make it easier for readers to see the nuts and bolts of how BCF works.

In addition, we note that BCF is based on two ideas that have a longstanding tradition in the field, so it is not as new or risky as it may appear initially. The first is Bayesian Additive Regression Trees (BART), which is the algorithm used to estimate the effect of the covariates on the outcome, and to estimate the interactions between the moderators and the treatment. The original BART papers^{5,6} were published fifteen years ago, cited over 1,400 times, and have been employed by researchers in a variety of disciplines for prediction and causal inference.^{7,8} Social scientists have been using the method since at least 2012^{9,10,11}. BCF builds on the original BART methods in a few ways that are explained clearly in the methods. For example, as mentioned, BCF uses BART prior distributions separately for the covariate function and for the moderator function, which strikes the right balance of controlling for potential chance baseline differences and exploiting predictive power to improve precision – similar to any regression adjustment – while applying judicious regularization to heterogeneous treatment effects. The main update to BCF developed for the present paper—adding a multi-arm implementation for the replication study—has a ready analogue in conventional linear models (e.g., linear mixed effects models and ANOVA) and is described in detail in the SI.

The second major foundation of BCF is Bayesian analysis. Of course, Bayesian inference itself has been in use for centuries. Recently, with advances in computing power, has it become faster and easier to estimate Bayesian models and summarize them by drawing from the posterior distribution (which is the Bayesian equivalent of a table of regression coefficients or *t*-tests). As a result, Bayesian modeling has quickly become standard in many areas. The papers we cite in the manuscript explain how Bayesian analysts summarize these draws from the posterior distribution in great detail¹².

Thus, the BCF method is solid and established, and we hope that our new text, and syntax, help more people learn how to adopt it.

The final model needs more detail – what covariates and moderators were included?

These are reported in the SI, and the information is copied above.

I would like to see raw data for each study, not just the lines from models and confidence intervals, which hides a lot, most notably individual data, which needs to be shown.

Our priority in this manuscript was to maximize clarity about the treatment effects for a reader of this complex set of studies. To that end, we carefully adhered to conventional standards in terms of how to summarize Bayesian results (e.g., draws from the posterior distribution). The convention involves presenting the estimates of interest—the group means, the treatment effects, the differences in treatment effects—and the uncertainty intervals around them—either through bands or the entire posterior

distribution. Note, that does not involve plotting raw data, which would have involved a substantial increase in complexity. Instead, what we have done is we provided the raw data for any reader to access on OSF. Overall, we feel this strikes the right balance for presentation of complex findings and open access for people who are interested in the raw data.

We also remind the reviewer that the noise from the raw data has already been factored into the model's uncertainty intervals, which are depicted in the figures, and so the model results are not "hiding" any of that variation.

Currently the paper does not tell us anything about individual level predictions but we would expect large individual differences for this kind of intervention. Who did it work for and why?

Our understanding of the reviewer's concern here is that we chose to prioritize an understanding of average treatment effects. Crucially, we remind the reviewer that we supplemented this with a robust heterogeneity analysis, looking at carefully-chosen moderators using very conservative statistical methods. What this approach allows us to do is to understand the overall effect of the intervention and begin to understand for whom the intervention is maximally effective.

Going beyond this approach—looking at individuals who differ in myriad ways—could be an interrogation of noise. We think that it is important to be judicious at this stage in the research program to focus on carefully selected moderators for heterogeneity analyses and believe we've achieved the appropriate balance for this stage. Now that we have observed a replicable effect with some heterogeneity, the field can advance and conduct more systematic heterogeneous effects tests, for example in a new national experiment (see, e.g., Yeager et al., 2019, *Nature*).

Did anyone get worse post-intervention?

We take this question to mean whether more people got worse post-intervention in the treatment group relative to the control group. Heterogeneity analyses did not find any subgroup of students who showed a reliable and meaningfully harmful average treatment effect for any outcomes across all studies.

Were there individual differences associated with, for example, gender, SES, personality, IQ?

We interpret this question to be asking about moderators/subgroups (not about simply explaining variation in the outcomes due to these covariates).

To respond, we note that we limited our moderation analyses to four theory-driven moderators in each study: sex, prior fixed mindset, prior stress mindset, and prior perceived stress. When using the BCF method, it is important to be judicious in the selection of moderators because the random forest priors are quite conservative and will "shrink to homogeneity" if too many irrelevant moderators are included in the model. Nevertheless, we have posted the data, and so readers will be welcome to conduct additional exploratory analyses if they choose to.

The following points answer the reviewer's specific queries:

- We did not find reliable evidence of moderation by sex across studies.
- We did not measure IQ in any study.
- In Studies 2 and 6 we had four Big 5 personality variables available to us. In response to the reviewer, we have now included them as covariates in those studies' analyses.

- In some studies, SES was measured. We did not find moderation by SES. We now included SES as a covariate wherever it was measured.

Moreover, we should also note that effects are likely robust across level of SES given that students from exclusively low SES backgrounds exhibited treatment effects in their naturalistic school setting (Study 5).

Some of the plots look very similar for the two groups (intervention and control e.g. Figure 4) and this made me wonder what exactly we are looking at – are they similar because they are modelled to be this way?

We take this question to be asking why the treatment and control groups looked similar to each other in some parts of the analyses (suggesting a weak average treatment effect). Yes, the Bayesian model includes a prior belief centered at a null average treatment effect, which shrinks the data toward a smaller treatment effect. This makes our findings more conservative, but also less likely to be influenced by noise in the data.

Data & methodology: validity of approach, quality of data, quality of presentation Studies 1 and 2. Not a criticism, just a comment: the results of the first two studies are not particularly surprising as they are essentially testing the understanding and/or memory of the statements delivered in the intervention.

While we agree that the self-report outcomes are perhaps less interesting than the physiological or internalizing ones, we think the first two studies provide valuable “proof of concept” data, and are impressive from a certain perspective. In Study 1, for example, these are middle school and high school teenagers who are experiencing a historically stressful pandemic in the Fall of 2020. In the midst of all of those distractions, our treatment encouraged them to see stress as potentially positive – and did so without any specialized training of counselors or mental health support staff. If this was *not* the case, then it would suggest that our findings would be more limited. Thus, although it is nice to show the downstream physiology effects, it is still critical to rigorously test the initial, cognitive appraisal effects, which could have easily been null, especially considering how hard it is to change teenagers’ minds.

Study 3. How does this intervention affect cardiovascular and physiological output? What is the mechanism? Further interrogation of the data would be useful here as I don’t think looking at prior mindsets – although interesting - tells us much about mechanism per se. Treated negative mindset individuals’ vascular resistance became indistinguishable from controls with positive prior mindsets – but how? What is the mechanism here?

Our model of the mechanism is presented in Fig. 1. Based on the theoretical frameworks and data observed across the diverse set of studies, we posited that the intervention changes how people *appraise* stressful events and their stress responses on two dimensions: good/bad for me, and controllable/uncontrollable. Those appraisals reflect the degree to which people perceived that they possessed the resources needed to address the demands the situation presents them with—that is, patterns of threat (resources < demands) versus challenge (resources \geq demands) type appraisals. Next, it is well-established in the BPS literature that challenge and threat appraisals are associated with specific patterns of physiological responding (e.g., Mendes & Park, 2014) derived from activation of the sympathetic-adrenal-medullary (SAM) and hypothalamic–pituitary–adrenal (HPA) axes. To provide some additional detail, both challenge and threat are accompanied by SAM activation. However, whereas challenge is

characterized by increased SV and decreased TPR stemming from SAM stimulation, threat also strongly activates the HPA axis, which counteracts vasodilatory effects of SAM stimulation resulting in increased TPR. While space does not permit detailed psycho-biological pathway information in the main text, we direct readers to engage with more detailed content from BPS literature in the models reviewed and in our references.

Also note that the previous version of the paper included event/response appraisals, but not global challenge/threat appraisals. We added the latter to Study 4 on an exploratory basis. We show that only the synergistic mindsets condition changed global challenge/threat appraisals and TPR during the stressful evaluative speech. Thus, the new data from Study 4 helps address R3's question about mechanisms. The text for the new findings is copied below:

Understanding mechanisms: Extension to stress appraisals. Study 4 also included, on an exploratory basis, two self-report measures that extended the model in Fig. 1. The first was a more direct measure of threat (vs. challenge) appraisals (e.g. ratings of the statements “*I felt threatened by the task*” and “*I felt that the task challenged me in a positive way.*”). Relative to the neutral control condition, the synergistic mindsets intervention reduced global threat-type appraisals by $-.46$ SD $[-.72, -.20]$, posterior probability = .988. The synergistic mindsets ATE was more than three times the magnitude of the single mindset conditions' ATEs (see Fig. 4B and C), posterior probabilities of a difference from the stress only condition = .936, for the growth mindset only condition = .953.

Extension to psychological well-being. The second exploratory analysis showed that the synergistic mindsets improved well-being (i.e. perceiving one felt liked, powerful, satisfied, good about themselves, and high self-esteem, and perceiving they did not feel rejected, insecure, or disconnected). The ATE for synergistic mindsets was $.25$ SD $[.04, .48]$, posterior probability = .941. The synergistic ATE was meaningfully larger than either of the ATEs for the single mindsets, which did not reliably improve well-being (see Fig. 4C), posterior probabilities of a difference from the stress mindset only ATE = .957, for growth mindset only = .884.

Although the intervention was associated with reductions in peripheral resistance, how did participants do on the stress tests (the speeches and the math)? Does the intervention improve performance, or make it worse, or have no effect? Is the reduction in peripheral responses the mediator of these effects? This would be interesting to look at, if possible.

We did not score participants' speech performance. In fact, there is no standard scoring procedure for TSST speeches. We shied away from this analysis in the present study because an unvalidated in-house scoring procedure might be hard to justify.

However, in Study 5, the longitudinal field study, we just recently obtained course pass rates at the end of the year from the school's registrar. To respond to the reviewer's comments added analyses of these data to the paper and show that the intervention increased core course pass rates, consistent with some previous mindset studies (e.g. Yeager et al., 2019, *Nature*). The new paragraph is copied below:

Academic achievement. A previous national experiment showed that a growth mindset intervention increased the rate of passing core classes (Math, Science, English/Language Arts, and Social Studies) among relatively lower-achieving students²⁷. Building on this, in Study 5 data on students' credit attainment were obtained from official school transcripts, which were recorded 7-8 months after the intervention in the Spring of 2020 during the COVID-19 lockdowns. Here, all students were relatively lower-achieving (average grade in core classes = 70%) and were therefore eligible to be included in analysis. This exploratory analysis, which used the conservative BCF method, found that the synergistic

mindsets intervention increased the overall rate of passing the core classes by 14.4 percentage points (pp) [0.4, 29.4]. Notably, treatment effects were driven by improvements in the most demanding STEM courses (Math and Science), which showed just a 47% pass rate in the control condition versus 63% in the synergistic mindsets condition, ATE = 14.5pp [0.4, 31.7].

Smaller and less reliable effects were observed in non-STEM courses (English/Language Arts and Social Studies), which had a much higher pass rate and tend to be less stressful on average (Control = 67%; Mindset Treatment = 73%; ATE = 5.3pp [-4.8, 17.2]). Treatment effects on course pass rates were not moderated by prior negative mindsets. Overall, this exploratory analysis showed that the intervention can meet its goal of both retaining students in the pipeline for more rigorous academic skills while also improving stress coping and well-being.

To answer the reviewer's follow-up question, in a previous study, we did find that TPR effects mediated mindset effects on performance on the TSST using an in-house scoring procedure (see the Supplemental Online Material in Yeager, Lee, & Jamieson, 2016, Study 1). So, there is reason to suspect that the same might be true for the synergistic mindsets intervention. We note, however, that TPR typically operates more effectively as an outcome than as mediator because variation in self-reports typically only weakly correspond to variation in stress physiology, similar to small associations observed among attitudes and behaviors. In addition, mediation analyses have problems with confounding because mediators are measured, not manipulated¹³. Because of the structure of the data and analytic concerns, we opted not to use a mediation approach in the present paper.

Study 4. We need to see the raw data. The lines look surprisingly similar between the two groups, and I have never seen such clean data with saliva cortisol – presumably this is because this is not data, but models?

See reply above. Also, we take this question to be asking why the treatment and control groups looked similar to each other in some parts of the analyses (suggesting a weak average treatment effect). Yes, the Bayesian model includes a prior belief centered at a null average treatment effect, which shrinks the data toward a smaller treatment effect. This makes our findings more conservative, but also less likely to be influenced by noise in the data.

Treated negative mindset individuals improved in terms of internalizing symptoms, but is this because they had further to move? As with study 3, what are the mechanisms by which the intervention affects cortisol, and does this improve performance on the stress test, and is it the mediator? This could all be analyzed and seems important to know.

The cortisol levels were not associated with any self-reports. They were negatively correlated with course pass rates ($r \sim -.20$), but we are not inclined to make a mediation argument about cortisol for reasons listed above.

Intervention. Some of the neuroscience in the intervention is questionable. For example, what is the evidence for the claim: ‘Doing challenging things strengthens your brain's connections’?

Here we are simply describing the basic fact that learning changes the brain's connectivity. There are many examples of this in the literature, including: <https://www.nature.com/articles/427311a>

And for the claim: ‘When people don't know this information about how the brain works, they tend to avoid frustration and difficulty by not doing things that feel hard.’

The following paper shows that students in two large national samples who have not received the growth mindset intervention message avoided challenging assignments and opted for easy assignments that did not make them struggle: <https://psycnet.apa.org/record/2020-84549-001>.

Stress is not always good for the brain – in fact most animal studies on stress show that stress is damaging to the brain, in the short and longer term. This is a concern for an intervention that might gain traction in schools.

In the discussion section, we are clear that stress is not always good. For instance, trauma is bad. The intervention content itself is careful on this point. The intervention states:

One final point: Not all stressful experiences are good

Thank you for your response.

Here's a final point. The lessons of this program apply to normal experiences in school, like stress we feel when we are trying to learn or master a difficult concept or skill. That's "good stress," and you should trust that your body's responses are helping you perform.

But sometimes people experience trauma—stress that is outside their control and harmful. If you experience trauma, you could reach out to a parent, teacher, counselor, or other adult who you trust for advice.

In addition, the first two paragraphs of our paper are clear that the kind of stress we are talking about is the normative, unavoidable stress of transitioning from a child into an adult. This stress is not "good," per se, but it comes from experiences which are normal and can be positive (like learning how to handle many different competing demands). The synergistic mindsets intervention was aimed at helping students optimally navigate and address these stressors, and it is effective at doing so.

Finally, regarding the reviewer's note that this is a concern for using the intervention in schools, we think this is a reason why follow-up RCTs are needed. Here, we present six studies and find that the intervention is quite helpful in a variety of populations for several key outcomes. Although it should continue to be tested, this is evidence at least that the intervention is helpful in real school contexts.

Why were different age groups (school students or undergraduates) included in each study? Was there a theoretical or scientific justification for this? They are quite different populations, biologically and in terms of social context and levels of independence in their lives. ... There were very different demographic groups in each study, with some studies involving social science undergraduate classes (presumably largely middle class and not hugely ethnically diverse?) whereas one study included adolescents from economically-disadvantaged families who were nearly all (95%) from black or indigenous racial/ethnic groups. Why were such different groups included across studies?

A main goal in our studies was to be inclusive of different age groups, geographic regions, social class backgrounds, and racial/ethnic backgrounds, while also building across the dependent measures, from appraisals to stress physiology to outcomes.

The key reason is that looking only at homogeneous samples can lead to findings that don't hold up to scrutiny and replication in more heterogeneous samples (see, e.g., ²). The first sample showed the

generality of the phenomenon in 36 schools. The next sample was a university sample, but it had many students from who were from lower social-class families (i.e. first-generation college students). This sample afforded a controlled, timed, naturalistic stressor (not just a hypothetical scenario). The next two samples were laboratory studies that afforded even more experimental control. These were based at the co-PI's university, out of necessity for the efficiency of data collection. Study 5 was designed to be diverse and different from the lab samples to test generalizability in a lower-SES sample. A final sample returned to the Study 2 setting, with a new cohort, primarily due to the good fortune of partnering with a classroom in which students were already scheduled to report on anxiety during the COVID-19 pandemic.

And why the very different sample sizes across studies, from 160 to 2,717? This is not well justified in the methods: in some studies, N was determined by power calculations, in others by pragmatics.

We revised the sample size justifications. In general, though, we followed our lab's standard operating protocol, which is to first identify a population of interest and then collect as much data from it as possible in the time frame allotted for the study. For Study 1, for instance, we could have conducted a much smaller study and still had sufficient statistical power, but we didn't see any reason to put a limit on our sample size if we could have more data in the same period of time.

Here are all of the justifications:

Study 1

Sample size determination. Sample size was planned to have sufficient power to detect a treatment effect in a field experiment of $.10 SD$ or greater, with $.10 SD$ being the minimum effect size that we would interpret as meaningful for a study focused on immediate post-test self-reports. We worked with our data collection partner, the Character Lab Research Network (CLRN)⁸⁹, to recruit as close to 3,000 participants as possible in a single semester. The final sample size was determined by the logistical constraints of data collection during the COVID-19 pandemic and by CLRN's data availability.

Study 2

Sample size determination. All students in an introductory social science course in Fall 2019 were invited to complete the intervention or control materials in return for a small amount of course credit. Sample size was set by the response rate.

Study 3

Sample size determination. An *a priori* power analysis was used to determine sample size. Previous stress research that assessed cardiovascular responses in laboratory-based stress induction paradigms produced medium to large effect sizes (e.g., range: $d = .59$ to $d = 1.44$ in Yeager et al., 2016, Jamieson et al., 2012, Oveis et al., 2020). Based on a standard medium effect size, at the low end of this range ($d = 0.50$), with a two-tailed hypothesis, G*Power indicated that 64 participants per condition (i.e., 128 total participants) would be necessary to achieve a target power level of $.80$ to test for effects of the treatment using frequentist methods. In anticipation of potential data loss, we determined *a priori* that we would oversample by 20%. Data collection was terminated the week after more than 150 participants had been enrolled in the study and provided valid data.

Study 4

Sample size determination. Study 3 showed an ATE for the synergistic mindsets intervention of approximately $.70 SD$ for TPR reactivity during the first minute of the speech epoch. Assuming an approximately 25% reduction in effect size for a replication study, then to have an 80% likelihood of reliably detecting an ATE of $.50 SD$ with a one-tailed hypothesis test (because this is a replication study), we calculated that we would need approximately 50 participants per condition. Our stopping rule was to

collect data from 200 participants who completed one of the conditions and provided valid TPR data for analysis.

Study 5

Sample size determination. We aimed for a minimum of 100 participants and 1,000 daily diary responses in this first-ever field experiment evaluating the synergistic mindsets treatment. We sought to recruit as many as possible before the end of October in the fall of 2019, because the study was focused on normative stressors at the start of a new school year, and because daily diary data collection could not happen during or after the Thanksgiving break in the U.S. (which is in late November). The number of students recruited each week was constrained by the research team's capacity to support twice-daily diary surveys and thrice-daily saliva samples in a school environment. The ultimate sample size was determined by the total number of students who could be recruited from the school in the fall semester of 2019, given these constraints.

Study 6.

Sample size determination. We recruited all students possible from an entire social science class in the spring of 2020, which, we would later learn, was a unique cohort for examining stress during the COVID-19 lockdowns. A minimum of 278 students would be needed to have a greater than 80% chance of detecting a directional effect on anxiety of $.3 SD$ with a conventional linear model analysis, and more students than this participated.

More information about covariates is needed. Was IQ measured? An IQ difference between the intervention and control group might explain the differences in some outcomes.

IQ was not measured, but (a) we used individual-level random assignment and we show that balance was achieved (see the SI), and (b) we are not aware of a theory that differences in IQ would lead to differences in threat-type stress responses (and we can think of many ways that it could have a difference in either direction, or none at all). Therefore, we think it is unlikely that an undetected difference in IQ across studies could account for our results.

Referee #4 (Remarks to the Author):

A. The authors present evidence from five double-blind RCTs of different populations of adolescents that a brief two-part self-administered online intervention designed to encourage the development of mindsets that (1) intelligence is malleable and (2) exposure to stress can be enhancing under certain circumstances improves stress-related cognitions, cardiovascular reactivity, and psychological well-being.

B. The research extends prior studies of mindset interventions (some by the lead author and his colleagues) by examining a combined intervention to change beliefs about both intelligence and stress. The research is novel and timely.

C. I have three concerns about the paper. First, and most importantly, the authors emphasize the synergistic nature of the intervention, which I take to mean that the two interventions interact to produce an outcome that is greater than either alone. But to test this hypothesis, the design would need to have four experimental groups (not simply one treatment group and a no-treatment control. In addition to the combined treatment and control groups, a proper test of the synergy hypothesis requires a group that receives the intelligence intervention but not the stress intervention, and a group that receives the latter but not the former. It is all well and good to point out that prior evaluations of stress coping interventions have not been greatly effective. But we don't know from the research presented here whether the specific stress intervention used in this

study would be just as effective on its own as in combination with the intelligence intervention. The argument that people who had the most dysfunctional mindsets of both types before the intervention benefited the most from the intervention isn't quite enough to persuade me that there is true synergy here.

We now report this experiment in Study 4 (text copied above).

My second concern is with the description of the treatment and control groups. I realize that details are included in the supplement, but I think that additional description of the intervention itself in the body of the text would be helpful.

While we agree with this comment, it was at odds with the need to have a short introduction. However we would be happy to follow the Editor's lead on this and add content if requested.

Along these lines, no description of the control group's experience is provided. What, exactly, were these participants asked to do?

The control group is described in detail in the methods right after the treatment. Both groups were asked to complete a computer module that was about stress and learning in school. Then the software randomized students to an active control condition or a mindset condition. From students' perspectives, the treatment and control group experience would have been highly similar. Also as noted we posted the control group activity (see link above).

Finally, I found the figures impenetrable, and more confusing than clarifying.

We currently find the figures to be helpful in showcasing our results, but we would be happy to be responsive to this comment in any way requested by the Editor, especially if there are certain aspects of the figures that are particularly impenetrable.

D. No concerns about the statistics or treatment of uncertainty.

E. The absence of appropriate comparison groups in which only one part of the two-part intervention is administered limits the validity of the conclusions.

F. The authors should provide evidence that the synergistic intervention is more effective than either component alone. Alternatively, they should temper their synergistic argument and simply describe the intervention as having two components.

G. No concerns about the references.

H. The manuscript is generally clearly written.

Referee #5 (Remarks to the Author):

This paper investigates the effect of a synergistic mindset intervention on reducing adolescents social stress. The authors studied five experiments with different stressors and concluded that the synergistic mindset intervention successfully reduced stress responses in all studies.

The proposed intervention is innovative and can potentially help with adolescent stress management. The main statistical model used in the paper is (targeted smooth) BCF, which is a flexible semiparametric model for evaluating causal effects. However, I have doubts about the

appropriateness of this model for your studies. In particular,

1. For most of the studies, the outcomes y_{ij} 's take discrete values (from 1 to 3, 5 or 7). However, the BCF model assumes a normal distribution. Why is the model appropriate for analyzing your data?

In most studies, the outcome is a continuous/interval valued variable that takes on many values (e.g. cortisol or TPR), so this is not an issue for most of the DVs in our paper.

The reviewer is correct, however, that in Study 5 the primary outcome is daily negative self-regard, which is a 7-point likert-type scale. This will cause a slight model misspecification for the error terms. The reason why we use the model that assumes a normal distribution stems from the estimand that we wish to estimate: the additive effect of the treatment on the Likert-style outcomes. If we used an ordinal regression model, that would not give us the estimand of interest.

Further, while not specific to BCF, past studies have found that model misspecifications that come from assuming that a 7-point scale is continuous rarely change the substantive inferences drawn from analyses (see e.g.¹⁴). For example, the cited paper states, “*With 6-7 categories, results were similar across methods for many conditions; in these cases, either method is acceptable.*”

We note that although it would have been ideal to have many items measuring daily negative self-regard to yield a continuous measure, this was not feasible. Recall that Study 5 was conducted in a highly socioeconomically disadvantaged community during the school day where time was at a premium. If the survey was longer, it would have reduced response rates and increased attrition. (For example, most students rode the bus home after school; the school administrators were concerned that if the end-of-day survey had been any longer they could have missed the bus, and working parents could not have picked them up).

2. How many covariates x_{ij} are included in the data analysis? For example, are the covariates gender and race included in the model? The BCF model is designed to incorporate a large number of covariates. If there are only one or two covariates, the BCF model doesn't have much benefit over linear regressions.

All covariates are now listed in table S5 in the Supplemental Information (**text copied above**). Each study included more than two covariates.

Further, the BCF model still has many benefits over linear regression even with few covariates. These benefits include Bayesian inference and conservative priors to avoid over-stating our results, along with flexible modeling of non-linear effects of controls and effect moderators, along with flexible modeling of the effect of time in the repeated measures studies.

3. For each study, are the individuals “similar” in mindset treatment and control groups? That is, are their propensity score distributions balanced?

We thank the reviewer for raising this issue. This should not be an issue because we are reporting a randomized experiment, and we also use BCF to model baseline differences using the observed covariates.

Nevertheless, to address this comment we used random forest methods to estimate propensity scores for the treatment and control groups for all studies and plotted distributions of the propensity scores in the

Supplemental Information. Groups were evenly balanced (See Figure S1), which is what we expected, of course, because individual-level random assignment was carried out cleanly by the Qualtrics software and there was little to no attrition in our studies.

Figure S1 in the SI shows that the only studies with seemingly meaningful differences across groups were the studies with smaller cell sizes, as would be expected from random noise. To understand this, we used a random permutation method (in which we randomly shuffled the treatment allocations and then re-ran the propensity score estimation) and found that the magnitudes of the differences in our data were very similar to random noise. For example, here is Study 4, the study with the smallest cell sizes. Panel A shows the propensity score distributions from the observed data. Panel B shows the distributions from randomly shuffled data.

Further, we note the variables used in the propensity score analyses were also included in the covariate function in the BCF model, which allowed us to correct for any chance imbalances. In summary, the studies' propensity scores were balanced to the extent expected by random variation alone, and our machine-learning model accounted for any observed imbalances.

4. The original BCF or targeted smooth BCF model includes the propensity score in the $\beta()$ function to mitigate treatment effect bias. However, the propensity score seems to be missed in your model. Why did you leave out the propensity score?

The original BCF and tsBCF models were developed (and applied in those papers) in the context of observational studies, not experiments. The propensity score adjustment is unnecessary in true random assignment experiments like those presented here, since the propensity score is a known, constant value (for a previous implementation of BCF for a randomized experiment, which likewise did not include the propensity score, see the SI for our 2019 *Nature* paper¹⁵). As noted above, randomization was successfully implemented with little to no attrition in all our studies.

As an exercise, we tried adding estimated propensity scores into our models and found that the results for the ATEs were identical to the third decimal of their posterior standard deviations, as expected based on statistical considerations.

Summary

We have responded to reviewers' comments, and added new data. We find that the results held up to the additional scrutiny. We hope that the paper is now ready for acceptance.

Sincerely,

The Authors

Second round of review:

Editorial note: *Reviewer 3 was unable to re-review the paper, so Reviewer 1 kindly provided comments on the revisions in response to Reviewer 3's concerns.*

Referee #1 (Remarks to the Author):

Authors' responses to my comments (reviewer #1):

The authors satisfactorily addressed all of my comments, with the exception of comment #2 as described below.

In response to comment #2, in which I requested the specific covariates that were entered into each BCF model, I appreciate the addition of a supplementary table listing the covariates that were included in each study. However, I wonder why this information was only included in a supplement rather than in the methods section of the manuscript. The covariates and moderators that were used in each model are important for interpreting the results of the model and should be described in detail the manuscript itself, with clarity about which covariates were included in each model, rather than in a supplement, which many readers will not review.

More importantly, relevant details about the measures used as covariates are still entirely missing. The general list of covariates that the authors added to the methods section does not provide sufficient detail about which covariates were included in which models, and it implies that perceived social stress was included in each model as a "core covariate and moderator" whereas Table S5 makes clear that it was not included in the models for Studies 2 or 3. Currently, only the negative prior mindset scales are described in the manuscript, and these descriptions do not specify how many items were included in each scale. Table S5 lists the construct measured by each covariate in each study, but does not identify the specific measure(s) used to assess these construct. The names of all survey scales (where applicable), how many items each scale included, sample items, and information about their reliability should be included in the methods section of the manuscript. I acknowledge that the full survey materials are posted on the study registration site, but these do not specify which items were included in which survey scales and are additionally unlikely to be accessed by most readers.

Finally, the reasons why certain covariates were not measured or were excluded in each study should be provided, rather than providing a blanket statement that "if a given covariate was not included in the model, it was not measured in that study." For example, why was race/ethnicity not included in the model for Study 5? The sample for Study 5 is described as "95% from Black or indigenous racial/ethnic groups (BIPOC)." BIPOC is a broad, heterogenous category with meaningful variation to be explored within this category, if participants' self-reports of their racial and ethnic identities were obtained.

Authors' responses to comments from reviewer #3:

Reviewer #3 commented that, in the abstract, it is misleading to report the total N across studies as the sample size. In response, the authors reported the sample sizes separately for each study in the abstract, with the exception of Studies 1 and 2 which were combined. For the same reasons originally stated by reviewer #3, it seems that the sample sizes for these studies should also be reported separately.

The authors pushed back on the reviewer's comment that Figure 1 is complex and broad. While I agree that the figure is complex and it takes some time to fully process its contents, I see the authors' point about the necessity of the complexity and I find the figure to be helpful and informative.

Similarly, the authors pushed back on reviewer 3's comment that the theory depicted in Figure 1

may not be falsifiable. I find the authors' rebuttal to be convincing.

In response to the reviewer's concern that only 2 of the 5 trials had been pre-registered, the authors registered all of the studies on Open Science Framework, added an in-depth discussion of steps taken to minimize researcher degrees of freedom in all studies in the Supplemental Information, and conducted a pre-registered replication and expansion of one of the non-pre-registered studies. While registering the studies on OSF after analyses were conducted does not address the reviewer's concern about the lack of pre-registration, it does allow the broader research community the opportunity to independently analyze the data and determine whether different analytic decisions may have produced different results (albeit, after the results are published and disseminated). The confirmatory findings from the pre-registered replication study are certainly encouraging and offer some confidence in the robustness of the results of at least this one study. I leave it up to the editor to determine whether this response, and the justification for not pre-registering three of the five initial studies, sufficiently address the reviewer's concerns about pre-registration.

I agree with the authors' assertion that the words "randomization" and "control" are appropriate terms to describe the experimental procedures used and do not need to be avoided solely because the experiments were not pre-registered. However, for reasons separate from those stated by reviewer #3, I do wonder if the use of the term "double-blind" is appropriate. Although participants were not aware that there were different experimental conditions, they were of course aware of the content of the training that they received and this awareness could influence outcome measures, particularly self-reports of appraisals that are directly targeted by the intervention content (which are the outcomes measured in studies 1 and 2). Participants are therefore not blind to their treatment, even if they are unaware that they are being assigned to a treatment or control condition. Many psychological experiments follow a similar protocol of randomizing participants to experimental conditions without informing participants of this manipulation, and these are not typically published as "double-blind" trials.

Regarding the reviewer's request for the inclusion of a hypothesis-testing framework and reporting of frequentist statistics through the paper, I agree with the authors' decision to continue to include the results of more classical frequentist / linear models in the Extended Data Table and to solely report the results of the BCF analyses in the paper, particularly as results from the two approaches do not substantively differ and the Bayesian framework is shifting away from the misleading "all-or-none" thinking of null hypothesis-testing.

Although I agree with the reviewer's comment that the BCF method is not easy to follow and requires expertise to understand it, I do not think this is a reason to avoid using an innovative method that the authors assert is superior to classical hypothesis-testing and linear modeling approaches. This paper may introduce many readers to a methodology that they may wish to learn and apply in their own work. I appreciate the authors' sharing of their syntax so that interested readers can examine and learn from it.

I am satisfied by the authors' justification for the decision not to plot raw data points in their figures.

In response to the reviewer's question about whether some plots for the treatment and control groups "are similar because they are modelled to be this way," the authors note that the groups may appear similar because the model shrinks the data towards a smaller treatment effect. However, I doubt the size of the treatment effect is what the reviewer's comment was referring to. Rather, I believe the reviewer was referring to the similar shape of the daily negative self-regard values and daily cortisol curves for the treatment and control groups in the current Figure 5 panels B and C (previously Figure 4). Importantly, nowhere in Figure 5 is it specified that model-based estimates are plotted in panels B and C. Rather, the vertical axis titles (e.g., "daily negative self-regard") imply that unadjusted mean values and percentiles are plotted. The axis titles or figure

notes should be modified to make clear that model-based estimates are plotted.

I believe that the authors' responses satisfactorily addressed all of reviewer 3's comments that are not explicitly addressed here.

Additional comments:

Table 1 states that the stressor in Study 6 is "Ongoing academic demands during COVID-19 quarantines." I understand that this is the stressor that is directly targeted by the synergistic mindsets intervention (whereas broader pandemic-related stress was not targeted), however it seems strange to single this out as the only or even primary stressor in this study since it was not manipulated or even measured, and presumably there were a host of other stressors (threats to personal safety and the health of loved ones, social isolation, uncertainty about the future) that may have been more salient than academic stressors for many students during the onset of the pandemic. In particular, individual differences in the outcome of "internalizing symptoms" (which is specifically a Generalized Anxiety Disorder measure, and should be clearly described as such in Table 1) seem likely to be driven at least as much by reactions to broader pandemic-related stressors as they are to be driven by the stress of ongoing academic demands during the pandemic. Rather than implying that the primary stressor of the pandemic is "ongoing academic demands," the authors may instead want to interpret this study as a potential generalization of the mindsets encouraged by the intervention beyond solely academic / "social-evaluative" stressors. Indeed, the authors appear to endorse this interpretation on page 22 line 463 where they state: "Study 6 provided a strong test of the hypothesis that the young people transferred the lessons from the intervention to cope with novel stressors," yet they walk back this implication two sentences later when they say "All it would need to do is help adolescents reappraise the normative social-evaluative demands that contributed to mental health problems and persisted during the pandemic."

Regarding the content of the intervention, I wonder whether there is any explicit acknowledgement that, sometimes, it is healthy to recognize one's own limits and make a decision to reduce the stressors in one's life. Accepting reasonable challenges and managing moderate stress to persist in those challenging situations can be helpful, but does the intervention run the risk of encouraging people to push themselves too hard to persist in highly challenging situations despite continuing to experience high levels of stress, resulting in greater internalizing symptoms in the long term? For example, a pre-med student who is greatly struggling with managing their stress may be right to decide that this level of stress is not healthy and switch to a major that will be less stressful. Does the intervention speak at all to recognizing the line between potentially manageable stress and severe/recurrent stress that is unhealthy for you (even if it is a "plausibly beneficial" stressor for others)? This level of stress is not "traumatic" or "uncontrollable," but it can still be chronic and detrimental to one's long-term health. I understand that this may be too nuanced of a message to include in such a brief intervention and may lead to confusion, but I do wonder about potentially detrimental long-term effects for some individuals. I appreciate the paragraph on p. 25 lines 523-532 noting that the intervention only applies to "plausibly beneficial or growth-positive stressors," but is any guidance provided to students on identifying which stressors are "growth-positive" for them, or at what point they should switch from stress coping to stressor reduction?

Methods, Study 5, p. 58, lines 1266-1270: The authors state that "Students attended a high-quality urban charter school which showed a high graduation rate (98%) relative to the urban city school district (68%). Therefore, this was a population that was expected to face social, economic, and academic stressors, but with supportive and competent teachers, and who could therefore make use of a stress optimization intervention." This statement is worrying as it begs the question: Do the authors expect that students in urban schools with lower graduation rates would not benefit from a stress optimization intervention? This has important implications for the generalizability of the intervention to the populations that may stand to benefit the most from an

intervention that enhances their growth mindset and stress coping skills. It also comes off as dismissive to imply that students from lower-performing urban schools could not “make use of” a stress optimization intervention, or that students from these schools have unsupportive and incompetent teachers. Similarly, on p. 17, lines 351-354, the authors state: “This population was chosen because students facing the combination of socioeconomic disadvantages and high academic standards are likely to face chronic, daily stressors which have the potential to elicit threat-type stress responses.” This sentence implies that students facing the combination of socioeconomic disadvantage and low academic standards are not likely to face chronic, daily stressors, which is certainly untrue as socioeconomic disadvantage is itself associated with chronic stressors. Rather, I think the authors mean that the study sample is likely to face greater academic stressors, although even this assertion seems unwarranted as students in lower-performing schools may still feel academically stressed (just because the school is low-performing, this does not mean the students in the school do not strive to succeed academically). Moreover, the references cited in this sentence do not support this assertion. In general, I do not see why the authors draw so much attention to the performance level of this school as potentially signifying that its students are more stressed or could “make better use of” a stress optimization intervention, and I think this entire argument is unnecessary and should be avoided.

p. 18, line 375: The clause “daily affect reports on intensely stressful days” appears to be misworded, as affect reports were obtained in reference to specific stressors that happened to occur on the days on which students completed the daily diaries, which were not necessarily “intensely stressful days.”

p. 60, line 1298: The authors state: “Note that although there was a possibility that the daily reflections influenced students’ self-reports moments later, they could not have influenced the saliva samples provided moments later, because, as noted, salivary cortisol levels reflect stress responses 30-45 minutes earlier.” The assertion that the daily reflections “could not have influenced” salivary cortisol levels moments later is not quite accurate because, although it is true that salivary cortisol levels do not typically peak until 30+ minutes after stressor onset, salivary cortisol levels do begin to increase shortly after stressor onset. It would be more accurate to state that the reflections are unlikely to have a large impact on salivary cortisol levels moments later. More importantly, the fact that participants completed daily reflections on the training content immediately prior to reporting their negative self-regard is an important detail that is left out of the results section and Figure 5A, which provides an overview of the procedures for Study 5. The impact of the intervention on daily negative self-regard seems much less impressive when one notes that, in the treatment group, self-regard was reported immediately after completing a 5-minute reflection on the growth/stress-is-enhancing mindsets presented in the training. Without these preceding intervention reflection activities, negative self-regard would presumably be less strongly influenced by the intervention completed weeks prior. Therefore, I believe this detail should be mentioned in the results section and included in Figure 5A rather than being buried in the methods. This does beg the question as to why the decision was made to have students complete these reflections prior to collecting self-reported affect and saliva samples, rather than the other way around. The rationale for this decision is not provided in the manuscript.

p. 17, line 363: The authors state “The daily surveys measured the intensity of evaluative stressors.” Were only “evaluative” stressors measured, or did participants report on any stressors they experienced that day? If the latter is true, the wording should be changed throughout the reporting of results from Study 5. Similarly, on p. 60, line 1307, the authors state: “days on which no social-evaluative stressors were listed were coded as a “1” for stressor intensity.” Did the coders make a judgment about whether a stressful event was a “social-evaluative” stressor versus another kind of stressor, and exclude those stressors that were not “social-evaluative”? For example, “daily hassles” would not necessarily be social-evaluative stressors. Again, the term “social-evaluative” or “evaluative” should not be used if all reported stressors were counted.

On pages 19-20, the authors slip into using the broad term “internalizing symptoms” instead of the

more precise term "negative self-regard." I believe the term "negative self-regard" should be used consistently since this is what was measured, and even if it is a core internalizing symptom it is not synonymous with "internalizing symptoms."

Editorial notes:

There are various typographical errors in the supplementary information that should be addressed prior to final publication (e.g., p. 12 line 281: "osur" for "our").

Abstract, line 31: "Studies 3" should be "Study 3."

p. 61, line 1331: The word "of" should be removed in "...because of noise in the data attenuates effect sizes."

Referee #2 (Remarks to the Author):

The authors have been responsive to my original comments. In particular, I think including another study that compared the synergistic mindsets condition with the single-mindset conditions (i.e., growth mindset only and stress mindset only) is a valuable addition. I recommend that this manuscript be published.

Referee #4 (Remarks to the Author):

I am pleased with this very responsive revision and have no further concerns about the manuscript.

Referee #5 (Remarks to the Author):

The authors have adequately addressed the questions in my original review.

Author rebuttals to the second round of review

Referee #1 (Remarks to the Author):

Authors' responses to my comments (reviewer #1):

The authors satisfactorily addressed all of my comments, with the exception of comment #2 as described below.

In response to comment #2, in which I requested the specific covariates that were entered into each BCF model, I appreciate the addition of a supplementary table listing the covariates that were included in each study. However, I wonder why this information was only included in a supplement rather than in the methods section of the manuscript. The covariates and moderators that were used in each model are important for interpreting the results of the model and should be described in detail the manuscript itself, with clarity about which covariates were included in each model, rather than in a supplement, which many readers will not review.

We list the covariates on page 39. Because of the Bayesian model we are using, the choice of covariates is not really an issue. It is a machine-learning algorithm that decides how best to use the covariates—if at all.

More importantly, relevant details about the measures used as covariates are still entirely missing. The general list of covariates that the authors added to the methods section does not provide sufficient detail about which covariates were included in which models, and it implies that perceived social stress was included in each model as a “core covariate and moderator” whereas Table S5 makes clear that it was not included in the models for Studies 2 or 3. Currently, only the negative prior mindset scales are described in the manuscript, and these descriptions do not specify how many items were included in each scale. Table S5 lists the construct measured by each covariate in each study, but does not identify the specific measure(s) used to assess these construct. The names of all survey scales (where applicable), how many items each scale included, sample items, and information about their reliability should be included in the methods section of the manuscript. I acknowledge that the full survey materials are posted on the study registration site, but these do not specify which items were included in which survey scales and are additionally unlikely to be accessed by most readers.

We now list the exact items for the prior mindsets, the PSS, text anxiety, and generalized anxiety in the SI. As the reviewer notes, the data and materials are accessible online.

Finally, the reasons why certain covariates were not measured or were excluded in each study should be provided, rather than providing a blanket statement that “if a

given covariate was not included in the model, it was not measured in that study.” For example, why was race/ethnicity not included in the model for Study 5? The sample for Study 5 is described as “95% from Black or indigenous racial/ethnic groups (BIPOC).” BIPOC is a broad, heterogeneous category with meaningful variation to be explored within this category, if participants’ self-reports of their racial and ethnic identities were obtained.

We now listed the specific racial groups. Also, we note that we have already shown earlier in the SI that there was no failure of random assignment by this variable.

Authors’ responses to comments from reviewer #3:

Reviewer #3 commented that, in the abstract, it is misleading to report the total N across studies as the sample size. In response, the authors reported the sample sizes separately for each study in the abstract, with the exception of Studies 1 and 2 which were combined. For the same reasons originally stated by reviewer #3, it seems that the sample sizes for these studies should also be reported separately.

We have done this

The authors pushed back on the reviewer’s comment that Figure 1 is complex and broad. While I agree that the figure is complex and it takes some time to fully process its contents, I see the authors’ point about the necessity of the complexity and I find the figure to be helpful and informative.

Similarly, the authors pushed back on reviewer 3’s comment that the theory depicted in Figure 1 may not be falsifiable. I find the authors’ rebuttal to be convincing.

In response to the reviewer’s concern that only 2 of the 5 trials had been pre-registered, the authors registered all of the studies on Open Science Framework, added an in-depth discussion of steps taken to minimize researcher degrees of freedom in all studies in the Supplemental Information, and conducted a pre-registered replication and expansion of one of the non-pre-registered studies. While registering the studies on OSF after analyses were conducted does not address the reviewer’s concern about the lack of pre-registration, it does allow the broader research community the opportunity to independently analyze the data and determine whether different analytic decisions may have produced different results (albeit, after the results are published and disseminated). The confirmatory findings from the pre-registered replication study are certainly encouraging and offer some confidence in the robustness of the results of at least this one study. I leave it up to the editor

to determine whether this response, and the justification for not pre-registering three of the five initial studies, sufficiently address the reviewer’s concerns about pre-registration.

I agree with the authors' assertion that the words "randomization" and "control" are appropriate terms to describe the experimental procedures used and do not need to be avoided solely because the experiments were not pre-registered. However, for reasons separate from those stated by reviewer #3, I do wonder if the use of the term "double-blind" is appropriate. Although participants were not aware that there were different experimental conditions, they were of course aware of the content of the training that they received and this awareness could influence outcome measures, particularly self-reports of appraisals that are directly targeted by the intervention content (which are the outcomes measured in studies 1 and 2). Participants are therefore not blind to their treatment, even if they are unaware that they are being assigned to a treatment or control condition. Many psychological experiments follow a similar protocol of randomizing participants to experimental conditions without informing participants of this manipulation, and these are not typically published as "double-blind" trials.

We disagree with the reviewer. In a double-blind trial (including in medical trials) all participants think they're getting a treatment. That is what accounts for any potential placebo effects.

Regarding the reviewer's request for the inclusion of a hypothesis-testing framework and reporting of frequentist statistics through the paper, I agree with the authors' decision to continue to include the results of more classical frequentist / linear models in the Extended Data Table and to solely report the results of the BCF analyses in the paper, particularly as results from the two approaches do not substantively differ and the Bayesian framework is shifting away from the misleading "all-or-none" thinking of null hypothesis-testing.

Although I agree with the reviewer's comment that the BCF method is not easy to follow and requires expertise to understand it, I do not think this is a reason to avoid using an innovative method that the authors assert is superior to classical hypothesis-testing and linear modeling approaches. This paper may introduce many readers to a methodology that they may wish to learn and apply in their own work. I appreciate the authors' sharing of their syntax so that interested readers can examine and learn from it.

I am satisfied by the authors' justification for the decision not to plot raw data points in their figures.

In response to the reviewer's question about whether some plots for the treatment and control groups "are similar because they are modelled to be this way," the authors note that the groups may appear similar because the model shrinks the data towards a smaller treatment effect. However, I doubt the size of the treatment effect is what the reviewer's comment was referring to. Rather, I believe the reviewer was referring to the similar shape of the daily negative self-regard values and daily cortisol curves for the treatment and control groups in the current Figure 5 panels B and C (previously Figure 4).

Ok

Importantly, nowhere in Figure 5 is it specified that model-based estimates are plotted in panels B and C. Rather, the vertical axis titles (e.g., “daily negative self-regard”) imply that unadjusted mean values and percentiles are plotted. The axis titles or figure notes should be modified to make clear that model-based estimates are plotted.

We note in the figure note that the lines are the expected values from the model.

I believe that the authors’ responses satisfactorily addressed all of reviewer 3’s comments that are not explicitly addressed here.

Additional comments:

Table 1 states that the stressor in Study 6 is “Ongoing academic demands during COVID-19 quarantines.” I understand that this is the stressor that is directly targeted by the synergistic mindsets intervention (whereas broader pandemic-related stress was not targeted), however it seems strange to single this out as the only or even primary stressor in this study since it was not manipulated or even measured, and presumably there were a host of other stressors (threats to personal safety and the health of loved ones, social isolation, uncertainty about the future) that may have been more salient than academic stressors for many students during the onset of the pandemic. In particular, individual differences in the outcome of “internalizing symptoms” (which is specifically a Generalized Anxiety Disorder measure, and should be clearly described as such in Table 1) seem likely to be driven at least as much by reactions to broader pandemic-related stressors as they are to be driven by the stress of ongoing academic demands during the pandemic. Rather than implying that the primary stressor of the pandemic is “ongoing academic demands,” the authors may instead want to interpret this study as a potential generalization of the mindsets encouraged by the intervention beyond solely academic / “social-evaluative” stressors. Indeed, the authors appear to endorse this interpretation on page 22 line 463 where they state: “Study 6 provided a strong test of the hypothesis that the young people transferred the lessons from the intervention to cope with novel stressors,” yet they walk back this implication two sentences later when they say “All it would need to do is help adolescents reappraise the normative social-evaluative demands that contributed to mental health problems and persisted during the pandemic.”

We have revised this.

Regarding the content of the intervention, I wonder whether there is any explicit acknowledgement that, sometimes, it is healthy to recognize one’s own limits and

make a decision to reduce the stressors in one's life. Accepting reasonable challenges and managing moderate stress to persist in those challenging situations can be helpful, but does the intervention run the risk of encouraging people to push themselves too hard to persist in highly challenging situations despite continuing to experience high levels of stress, resulting in greater internalizing symptoms in the long term? For example, a pre-med student who is greatly struggling with managing their stress may be right to decide that this level of stress is not healthy and switch to a major that will be less stressful. Does the intervention speak at all to recognizing the line between potentially manageable stress and severe/recurrent stress that is unhealthy for you (even if it is a "plausibly beneficial" stressor for others)? This level of stress is not "traumatic" or "uncontrollable," but it can still be chronic and detrimental to one's long-term health. I understand that this may be too nuanced of a message to include in such a brief intervention and may lead to confusion, but I do wonder about potentially detrimental long-term effects for some individuals. I appreciate the paragraph on p. 25 lines 523-532 noting that the intervention only applies to "plausibly beneficial or growth-positive stressors," but is any guidance provided to students on identifying which stressors are "growth-positive" for them, or at what point they should switch from stress coping to stressor reduction?

Yes, in the intervention we have the text copied below.

One final point: Not all stressful experiences are good

Thank you for your response.

Here's a final point. The lessons of this program apply to normal experiences in school, like stress we feel when we are trying to learn or master a difficult concept or skill. That's "good stress," and you should trust that your body's responses are helping you perform.

But sometimes people experience trauma—stress that is outside their control and harmful. If you experience trauma, you could reach out to a parent, teacher, counselor, or other adult who you trust for advice.

Methods, Study 5, p. 58, lines 1266-1270: The authors state that "Students attended a high-quality urban charter school which showed a high graduation rate (98%) relative to the urban city school district (68%). Therefore, this was a population that was expected to face social, economic, and academic stressors, but with supportive and competent teachers, and who could therefore make use of a stress optimization intervention." This statement is worrying as it begs the question: Do the authors expect that students in urban schools with lower graduation rates would not benefit from a stress optimization intervention? This has important implications for the generalizability of the intervention to the populations that may stand to benefit the most from an intervention that enhances their growth mindset and stress coping skills. It also comes off as dismissive to imply that students from lower-performing

urban schools could not “make use of” a stress optimization intervention, or that students from these schools have unsupportive and incompetent teachers.

Similarly, on p. 17, lines 351-354, the authors state: “This population was chosen because students facing the combination of socioeconomic disadvantages and high academic standards are likely to face chronic, daily stressors which have the potential to elicit threat-type stress responses.” This sentence implies that students facing the combination of socioeconomic disadvantage and low academic standards are not likely to face chronic, daily stressors, which is certainly untrue as socioeconomic disadvantage is itself associated with chronic stressors. Rather, I think the authors mean that the study sample is likely to face greater academic stressors, although even this assertion seems unwarranted as students in lower-performing schools may still feel academically stressed (just because the school is low-performing, this does not mean the students in the school do not strive to succeed academically).

We have revised this. All we were saying is that we think students will have social-evaluative stressors, and therefore they can make use of the treatment. This is important to say because in future replications an investigator might administer the treatment to populations that are not undergoing stressors, and it would not be expected to work.

Moreover, the references cited in this sentence do not support this assertion. In general, I do not see why the authors draw so much attention to the performance level of this school as potentially signifying that its students are more stressed or could “make better use of” a stress optimization intervention, and I think this entire argument is unnecessary and should be avoided.

We are trying to provide a guide to future replication studies to let them know our thinking. We are not making a claim, but we are simply trying to leave a record of our speculations.

p. 18, line 375: The clause “daily affect reports on intensely stressful days” appears to be mis-worded, as affect reports were obtained in reference to specific stressors that happened to occur on the days on which students completed the daily diaries, which were not necessarily “intensely stressful days.”

We have fixed this.

p. 60, line 1298: The authors state: “Note that although there was a possibility that the daily reflections influenced students’ self-reports moments later, they could not have influenced the saliva samples provided moments later, because, as noted, salivary

cortisol levels reflect stress responses 30-45 minutes earlier.” The assertion that the daily reflections “could not have influenced” salivary cortisol levels moments later is not quite accurate because, although it is true that salivary cortisol levels do not typically peak until 30+ minutes after stressor onset, salivary cortisol levels do begin to increase shortly after stressor onset. It would be more accurate to state that the reflections are unlikely to have a large impact on salivary cortisol levels moments later.

Students provided saliva samples before the daily reflections or simultaneously with them, so it was not biologically plausible for the daily surveys to influence the cortisol levels.

More importantly, the fact that participants completed daily reflections on the training content immediately prior to reporting their negative self-regard is an important detail that is left out of the results section and Figure 5A, which provides an overview of the procedures for Study 5. The impact of the intervention on daily negative self-regard seems much less impressive when one notes that, in the treatment group, self-regard was reported immediately after completing a 5-minute reflection on the growth/stress-is-enhancing mindsets presented in the training. Without these preceding intervention reflection activities, negative self-regard would presumably be less strongly influenced by the intervention completed weeks prior. Therefore, I believe this detail should be mentioned in the results section and included in Figure 5A rather than being buried in the methods. This does beg the question as to why the decision was made to have students complete these reflections prior to collecting self-reported affect and saliva samples, rather than the other way around. The rationale for this decision is not provided in the manuscript.

The daily reflections were captured as qualitative data, as we state on page 52. This was the first field experiment we conducted and we wanted to know how students were using the treatment messages, if at all, so that we could improve on and expand the intervention in the future. In the main text we now say that the effects on achievement at the end of the year rule out the possibility that participants only appeared to be coping better later in the day because they had completed the daily writing exercises on that morning, since the achievement results were recorded months later.

p. 17, line 363: The authors state “The daily surveys measured the intensity of evaluative stressors.” Were only “evaluative” stressors measured, or did participants report on any stressors they experienced that day? If the latter is true, the wording should be changed throughout the reporting of results from Study 5. Similarly, on p. 60, line 1307, the authors state: “days on which no social-evaluative stressors were listed were coded as a “1” for stressor intensity.” Did the coders make a judgment about whether a stressful event was a “social-evaluative” stressor versus another kind of stressor, and exclude those stressors that were not “social-evaluative”? For example, “daily hassles” would not necessarily be social-evaluative stressors. Again,

the term “social-evaluative” or “evaluative” should not be used if all reported stressors were counted.

As noted in the SI, we followed procedures laid out by Yeager et al. (2016), *Psych Science*. We measured several different categories of stressors, including non-social-evaluative ones. These categories were generated from reliable coding in the 2016 paper. Only the categories that were pre-determined to be social-evaluative, in that paper and in the pre-registration for the replication of the 2016 paper (cited in the SI), were designated as such in the present paper. To summarize, we followed a pre-set procedure and it is accurately described in the paper.

On pages 19-20, the authors slip into using the broad term “internalizing symptoms” instead of the more precise term “negative self-regard.” I believe the term “negative self-regard” should be used consistently since this is what was measured, and even if it is a core internalizing symptom it is not synonymous with “internalizing symptoms.”

We fixed this.

Editorial notes:

There are various typological errors in the supplementary information that should be addressed prior to final publication (e.g., p. 12 line 281: “osur” for “our”).

We fixed this.

Abstract, line 31: “Studies 3” should be “Study 3.”

We fixed this.

p. 61, line 1331: The word “of” should be removed in “...because of noise in the data attenuates effect sizes.”

We fixed this.